# Impacts of a double-moment bulk cloud microphysics scheme (NDW6-G23) on aerosol fields in NICAM.19 with a global 14-km grid resolution

Daisuke Goto[1], Tatsuya Seiki[2], Kentaroh Suzuki[3], Hisashi Yashiro[1], Toshihiko Takemura[4]

[1]National Institute for Environmental Studies, Tsukuba, 305-8506, Japan
[2]Japan Agency for Marine-Earth Science and Technology, Yokohama, 236-0001, Japan
[3]Atmosphere and Ocean Research Institute, The University of Tokyo, Kashiwa, 277-8568, Japan
[4]Research Institute for Applied Mechanics, Kyushu University, Fukuoka, 816-8580, Japan

*Correspondence to*: Daisuke Goto (goto.daisuke@nies.go.jp)

**Abstract.** In accordance with progression in current capabilities towards high-resolution approaches, applying a convective-permitting resolution to global aerosol models helps comprehend how complex cloud-precipitation systems interact with aerosols. This study investigates the impacts of a double-moment bulk cloud microphysics scheme, i.e., NICAM Double-moment bulk Water 6 developed in this study (NDW6-G23), on the spatiotemporal distribution of aerosols in the Non-hydrostatic Icosahedral Atmospheric Model as part of the version 19 series (NICAM.19) with 14 km grid spacing. The mass concentrations and optical thickness of the NICAM-simulated aerosols are generally comparable to those obtained from in situ measurements. However, for some aerosol species, especially dust and sulfate, the differences between the NDW6 and NSW6 experiments were larger than those between experiments with different horizontal resolutions (14 km and 56 km grid spacing), as shown in a previous study. The simulated aerosol burdens using NDW6 are generally lower than those using NSW6; the net instantaneous radiative forcing due to aerosol-radiation interaction (IRFari) is estimated to be -1.36 Wm$^{-2}$ (NDW6) and -1.62 Wm$^{-2}$ (NSW6) in the global annual mean values at the top of the atmosphere (TOA). The net effective radiative forcing due to anthropogenic aerosol-radiation interaction (ERFari) is estimated to be -0.19 Wm$^{-2}$ (NDW6) and -0.23 Wm$^{-2}$ (NSW6) in the global annual mean values at the TOA. This difference among the experiments using different cloud microphysics modules, i.e., 0.26 Wm$^{-2}$ or 16% difference in IRFari values and 0.04 Wm$^{-2}$ or 16% difference in ERFari values, is attributed to a different ratio of column precipitation to the sum of the column precipitation and column liquid cloud water, which strongly determines the magnitude of wet deposition in the simulated aerosols. Since the simulated ratios in the NDW6 experiment are larger than those of the NSW6 result, the scavenging effect of the simulated aerosols in the NDW6 experiment is larger than that in the NSW6 experiment. A large difference between the experiments is also found in the aerosol indirect effect (AIE), i.e., the net effective radiative forcing due to aerosol-cloud interaction (ERFaci) from the present to preindustrial days, which is estimated to be -1.28 Wm$^{-2}$ (NDW6) and -0.73 Wm$^{-2}$ (NSW6) in global annual mean values. The magnitude of the ERFaci value in the NDW6 experiment is larger than that in the NSW6 result due to the differences in both the Twomey effect and the

susceptibility of the simulated cloud water to the simulated aerosols between NDW6 and NSW6. Therefore, this study shows the importance of the impacts of the cloud microphysics module on aerosol distributions through both aerosol wet deposition
and AIE.

## 1 Introduction

The aerosol-cloud interaction (ACI) is one of the largest sources of uncertainty in near-term climate projections (Szopa et al., 2021). The radiative forcing related to the ACI is estimated to range from -1.45 Wm$^{-2}$ to -0.25 Wm$^{-2}$, which is the largest
among the various forcing agents  (Forster et al., 2021). The major process of the ACI is aerosol activation to act as cloud condensation nuclei (CCN) and its subsequent modification of cloud properties through perturbations to cloud droplet number concentration (Twomey, 1977) and to cloud lifetime via water conversion from cloud to precipitation (Albrecht, 1989). On the other hand, in terms of the aerosol itself, wet deposition through rainout and washout often dominates the sink process and determines the spatiotemporal distribution. Because most aerosols are hygroscopic, they are removed from the atmosphere
mainly by rainout or in-cloud scavenging (e.g., Henzing et al., 2006). In the rainout process, activated or formed aerosols in individual cloud droplets fall to the ground surface by precipitation. The modeling of rainout strongly affects the spatiotemporal variation and distribution of hygroscopic aerosols such as sulfate, organic aerosols, and sea salt (Textor et al., 2006; Myhre et al., 2013; Gliß et al., 2021). Even for less hydroscopic aerosols such as dust and black carbon (BC), the wet deposition process is important to determine their atmospheric lifetime (Koffi et al., 2016; Sand et al., 2021). Thus, aerosols and clouds are tightly
connected to each other, and hence, an evaluation of both the cloud module and aerosol physics module is required to improve the ACI in climate models. One of the methods to improve cloud simulations is the use of convection-permitting resolution, which explicitly represents cloud systems with a detailed cloud microphysics scheme (Satoh et al., 2019; Stevens et al., 2019). In very high-resolution models with a horizontal grid size of O (10 km) or less, clouds and precipitation are more realistically represented compared to conventional global models with a grid size of O (100 km) (e.g., Stevens et al., 2019). These results
suggest that convective cloud systems are better represented with a finer model resolution for which cumulus parameterizations are avoided (e.g., Vergara-Temprado et al., 2020). However, most global models with convection-permitting resolution do not treat aerosols explicitly or do not deeply evaluate aerosol distributions because of very expensive computational costs (Satoh et al., 2019; Stevens et al., 2019; Coppola et al., 2020).
One of the global models with convection-permitting resolution is the Non-hydrostatic Icosahedral Atmospheric Model
(NICAM: Tomita and Satoh, 2004; Satoh et al., 2008; Satoh et al., 2014; Kodama et al., 2021) coupled to an aerosol physics module (Suzuki et al., 2008; Dai et al., 2014; Goto, 2014), and the ACI in global cloud resolving simulations has been examined for a decade or more (Suzuki et al., 2008; Sato et al., 2018; Goto et al., 2020). High-resolution simulations of aerosols have various advantages for reproducing the distribution of the observed aerosols (Goto et al., 2015, 2020) and better representing the ACI effect by more realistically simulating the relationship between changes in cloud liquid water path (LWP) and aerosols

(Sato et al., 2018). Especially in the Arctic, the simulated aerosols in the high-resolution model are closer to the observations than those in the low-resolution model (Ma et al., 2014; Sato et al., 2016; Goto et al., 2020). With further improvements in computing resources, online aerosol calculations in such high-resolution models are highly promising next steps for understanding the interaction between aerosols, clouds, and precipitation. On the other hand, some issues remain even in global high-resolution simulations using NICAM (Goto et al., 2020). For example, the difference in the simulated aerosol optical

thickness (AOT) with high- and low-resolution models is small and estimated to be 3% of the global average, whereas the difference in the simulated aerosol mass concentrations at the surface is large and estimated to be 20% near the source areas. Over remote oceans such as the Southern Ocean, the simulated AOT sometimes exceeds 0.3 in monthly averages, which apparently shows the overestimation of the simulated AOT compared to the satellite observations. The simulated AOTs include a relatively large bias of 20% compared to the surface-observed results. Past research (Goto et al., 2020) indicated that biases

could be partially resolved by improving wet deposition through improved cloud-precipitation processes.

The main objective of this study is to clarify the impacts of cloud microphysics modules on aerosol distribution. Therefore, this study uses two different types of cloud microphysics schemes in the NICAM. For the evaluation, the simulated aerosols, clouds, precipitation, and radiation are compared with the observations. In addition, the global budgets for the simulated aerosols are compared to other models for reference.

Section 2 describes the model and the observations used in this study. Section 3 shows the results of the simulated clouds (section 3.1), precipitation (section 3.1) and aerosols (sections 3.2 and 3.3) in the numerical experiments using both the NICAM double-moment bulk cloud microphysics module (NDW6) and the NICAM single-moment bulk module with 6 water categories (NSW6). They are evaluated by a reference obtained from the NICAM with 14 km and 56 km grid spacing in Goto et al. (2020). Section 4 shows and discusses the impacts of aerosols on radiation through aerosol-radiation interactions (ARIs)

and ACIs by comparing them with references obtained from both models and satellites. Finally, the summary is shown in section 5.

## 2 Model descriptions and method

### 2.1 Atmospheric model

The NICAM is a Non-hydrostatic atmospheric model (Tomita and Satoh, 2004; Satoh et al., 2008; Satoh et al., 2014) that can

be run with a coarse resolution of 50 km to 200 km (e.g., Dai et al., 2014; Kodama et al., 2021). It is also a global model with convection-permitting resolution (Satoh et al., 2019) that greatly helps the understanding of atmospheric phenomena related to clouds and precipitation by resolving the interaction among multiple convective systems (Satoh et al., 2014). The horizontal grid sizes in the NICAM generally range from O(1 km) to O(10 km) and are often set at 14 km for a useful and effective balance between model complexity and computing resources (Kodama et al., 2014; Kodama et al., 2021; Seiki et al., 2022).

NICAM aerosol simulations with 14 km grid sizes were performed for the entire year (Sato et al., 2018; Goto et al., 2020). This study improves previous aerosol simulations (Goto et al., 2020) by using an upgraded version of the NICAM (replacing

the version 16 series with the version 19 series, hereafter referred to as NICAM.19) and the sophisticated cloud microphysics module NDW6 (the original version named NDW6-SN14 was incorporated into the NICAM by Seiki and Nakajima, 2014, the updated version named NDW6-S15 was incorporated into the version in NICAM.19 by Seiki et al., 2014, 2015, and the

current version named NDW6-G23 considers the interaction between NDW6-S15 and an aerosol module is introduced to NICAM.19 in this study. The details of the NDW6 update are described in Seiki et al., 2022).

NICAM.19 is an official version of the NICAM that was released at the end of 2019. After the official release, minor updates in NICAM.19 were continuously released. One of the updates of NICAM.19 from NICAM.16 is the vertically high resolution in the standard experiment. The number of vertical layers in NICAM.19 is 78 (15 layers below 2-km height), which is finer

than the 38 (10 layers below 2-km height) in NICAM.16. The layer heights at the bottom and top are 33 m and 50 km, respectively, in NICAM.19, whereas they are 81 m and 37 km, respectively, in NICAM.16. The increased vertical levels force the timestep to change from 60 seconds in NICAM.16 to 30 seconds in NICAM.19. Various bugs in NICAM.16 are eliminated in NICAM.19, and the aerosol module in NICAM.19 is also updated (explained in section 2.2).

This study uses the double-moment bulk cloud microphysics scheme NDW6, which is newly coupled to the aerosol physics

module in this study. For comparison, the original single-moment bulk cloud microphysics scheme (NSW6: Tomita, 2008; Kodama et al., 2012; Roh and Satoh, 2014) is also used. NSW6 predicts the mass mixing ratios of 6 water substances, i.e., water vapor, cloud water, rain, cloud ice, snow, and graupel. Therefore, the cloud droplet number concentration (CDNC) is assumed to be the same as CCN, which was calculated by coupling with the aerosol physics model using the CCN parameterization proposed by Abdul-Razzak and Ghan (2000). This parameterization is a function of the parameterized updraft

velocity with turbulent kinetic energy (Lohmann et al., 1999), aerosol sizes, and aerosol chemical composition. The CDNC is then used for autoconversion and accretion in rain formation. In this way, the ACI for both stratiform- and convective-cloud systems is incorporated in the cloud microphysics scheme. On the other hand, NDW6 predicts both the mass mixing ratios and the number concentrations of water substances. Prior to this study, NDW6 was not coupled with aerosol physics models, and CCN number concentrations at a background level were assumed to be constant globally (Seiki and Nakajima, 2014). In

accordance with the nucleation procedure, the background CCN value set at NDW6-SN14 and NDW6-S15 is replaced with predicted CCN values from the aerosol physics model using the CCN parameterization (Abdul-Razzak and Ghan 2000). In addition, a source term of CDNC value is assumed to be updated to a CCN value only when the CCN value exceeds the CDNC value in a grid box. The CDNC is updated with source (aerosol activation) and sink (autoconversion, accretion, and evaporation for water clouds) in NDW6 (Seiki and Nakajima, 2014). The balance of source and sink tendencies determines the CDNC in

NDW6. In this way, NSW6 and NDW6 coupled with the aerosol physics model are affected by the global distribution of aerosols.

Note that autoconversion and accretion, which mainly determine the strength of aerosol lifetime effects (Albrecht, 1989), are different between NSW6 and NDW6. NDW6 uses the parameterization proposed by Seifert and Beheng (2006), and NSW6 uses the parameterization proposed by Khairoutdinov and Kogan (2000). In addition, since NDW6 predicts the CDNC, the

CDNC and aerosols are individually transported by advection and removed by reduction terms. In contrast, NSW6 assumes

that a change in CCN directly connects with a change in diagnosed CDNC. These differences influence the representation of ACI.

Most relevant cloud parameters used to evaluate the ACI, e.g., LWP, cloud optical thickness (COT), and cloud fraction (CF), are output in every timestep, but in this study, cloud droplet effective radius (CDR) and cloud albedo (CA) were calculated using monthly mean parameters as postprocessing after the model integration. The CF is defined as the cloud occurrence frequency because NICAM with NDW6 and NSW6 does not consider partial-grid clouds. Clouds in a grid exist when the mixing ratios of the sum of cloud water and rain exceed $10^{-5}$ (kg m$^{-3}$), which can be detected by satellites (Goto et al., 2019). In this study, the CDR was calculated using monthly mean cloud water mass and number concentrations. However, only when the simulated CDR at the top of warm clouds were evaluated by a satellite, the simulated CDR with LWP > 1 g m$^{-2}$ and cloud top temperature >273.15 K were extracted. Unfortunately, the calculations were performed for only 1-year because of limitations of available computer resources. The CA is assumed by the following formulation (Platnick and Twomey, 1994) using monthly mean COT ($\tau_c$) for water clouds.

$$CA = \frac{\tau_c(1-g)}{1.5+\tau_c(1-g)} \qquad (1)$$

where g is the asymmetry factor and set at 0.85.

Other physical processes in this study are identical to those set in Goto et al. (2020). The advection module is per Miura (2007) and Niwa et al. (2011). The diffusion module is the level-2 Mellor-Yamada-Nakanishi-Niino (MYNN) scheme (Mellor and Yamada, 1972; Nakanishi and Niino, 2004; Noda et al., 2010). As in previous studies using the NICAM (e.g., Satoh et al., 2010; Kodama et al., 2021), no parameterization schemes for deep and shallow convection are used in this study. The land surface module is the Minimal Advanced Treatments of Surface Interaction and Runoff (MATSIRO) (Takata et al., 2003). The radiation module is the Model Simulation Radiation Transfer code (MSTRN-X) (Sekiguchi and Nakajima, 2008). The aerosol module is the Spectral Radiation-Transport Model for Aerosol Species (SPRINTARS) (Takemura et al., 2005; Suzuki et al., 2008), which is explained in section 2.2.

**2.2 Aerosol module**

The mass mixing ratios of the major tropospheric aerosols (dust, sea salt, carbonaceous aerosols including organic matter or OM and BC, and sulfate) and the precursors of sulfate ($SO_2$ and dimethyl sulfide or DMS) are explicitly calculated in the SPRINTARS-based aerosol module. The details of the aerosol module coupled to the NICAM are also described elsewhere (Dai et al., 2014; Goto et al., 2015, 2019, 2020; Goto and Uchida, 2022), but the main three updates in this study are explained as follows. First, when the CCN number concentration is higher than the CDNC calculated online in the aerosol module, the value of water supersaturation is positive, and the atmospheric pressure is above 300 hPa, the CCN number concentration becomes an input of source tendency for CDNC. The vertical fluxes of the simulated hydrometeors in the cloud microphysics module are used in the wet deposition for aerosols. Second, the assumption of sulfate in clouds is modified. In this study, the sulfate formed in the clouds by aqueous-phase oxidation at the current timestep is not scavenged by the rainout process at the

same timestep because the cloud water used in aqueous-phase oxidation is an output at the current timestep. The model timestep is 30 seconds, so this assumption is reasonable in this simulation. Because the model timestep was more than 1 minute in previous studies (Goto et al., 2020), the original model assumes that the sulfate formed in clouds by aqueous-phase oxidation is scavenged by the rainout process at the same timestep. This is one of the uncertainties of the modeling, and the assumption has an impact on the simulated sulfate, as shown later. Third, the treatment of dust aerosols is modified according to the latest version of SPRINTARS coupled to MIROC (Takemura et al., 2009; Tatebe et al., 2019). Dust particles in a wide range of sizes (from 0.13 μm to 8.02 μm in mode radii) are divided into bins, and the number of bins is reduced from 10 to 6. In addition, the dependence on the leaf area index (LAI) is a newly introduced function of the dust emissions in the aerosol module. The dust emission is a function of the cube of the wind speed at a height of 10 m, absorbed photosynthesis radiation depending on the LAI, soil moisture, and snow cover by using empirical coefficients that depend on 7 regions in the world (Takemura et al., 2009). The empirical coefficients, i.e., threshold values of soil moisture and emission strength, are newly tuned in this study. Except for these updates, the treatment and tuning parameters for the aerosol processes in this study are identical to those in Goto et al. (2020).

The removal processes, i.e., wet deposition, dry deposition, and gravitational settling, for aerosols are not different from those used in previous studies (Goto et al., 2020; Goto and Uchida, 2022). However, the wet deposition fluxes simulated by the NICAM in this study are directly modulated by the change in the cloud microphysics modules and autoconversion from clouds to precipitation because the wet deposition flux is strongly related to clouds and precipitation outside the aerosol module (Goto and Uchida, 2022).

For carbonaceous aerosols, SPRINTARS assumes both external and internal mixtures of organic matter (OM) and BC. Pure OM is generated from terpenes as a product of SOA, whereas pure BC is directly emitted from one-half of the amount in anthropogenic sources. SPRINTARS assumes that pure BC is not aged in the atmosphere. The BC and OM components emitted from other emission sources are internally mixed as two types of internal mixtures of OM and BC with BC to OM ratios of 0.3 and 0.15, respectively. BC, OM and sulfate are assumed to have lognormal particle size distributions with mode radii of 0.1 μm for the internal mixture of BC and OM, 0.08 μm for pure OM, 0.054 μm for pure BC and 0.0695 μm for sulfate. For sea salt, there are 4 categories of tracers, with mode radii of 0.178 μm, 0.562 μm, 1.78 μm, and 5.62 μm, that do not age or coagulate with each other in SPRINTARS. The internal mixture of BC and OM, pure OM, sulfate, and sea salt is hydrophilic, whereas dust and pure BC are hydrophobic. Such physical properties for aerosols in this study are identical to those used in Goto et al. (2020).

The optical properties of the aerosols and the calculation methods for ACI in this study are also identical to those used in Goto et al. (2020). The AOT at a wavelength of 550 nm is calculated online by the mass concentrations and optical properties for the aerosols and a look-up table prescribed by the Mie theory (Sekiguchi and Nakajima, 2008). To evaluate the radiative forcing of the ARI and ACI, the instantaneous radiative forcing of the ARI (IRFari) and effective radiative forcing for the ACI (ERFaci) are calculated by a general method (e.g., Shindell et al., 2013). The IRFari due to each aerosol species is calculated online by the difference in the radiative fluxes with/without the aerosol species in the radiation module (Goto et al., 2020). The

ERFari due to anthropogenic aerosols is calculated as the difference in the IRFari between the preindustrial and present conditions of aerosols. The ERFaci due to anthropogenic aerosols only is calculated by the difference in the cloud radiative fluxes between the preindustrial and present conditions of aerosols according to the method proposed by Ghan (2013). The impacts of anthropogenic aerosols on radiative forcing are estimated by the difference between the standard experiment and the extra experiment under preindustrial conditions. In the extra experiment, everything is the same as those in the standard experiment, except that the anthropogenic emission fluxes of BC, OC and $SO_2$ are set to zero. The uncertainty of this assumption is mentioned in section 2.3.

## 2.3 Experimental conditions

All experiments with both NDW6 and NSW6 are carried out for 6-years after the 1-month spin-up calculation. The simulation results are climatological runs, because the model does not nudge meteorological fields such as wind and temperatures but nudges the sea surface temperature (SST) and sea ice by the results of the NICAM from Kodama et al. (2015). The initial conditions for the model spin-up are obtained from the end of the 1-year aerosol simulations coupled to NSW6 without nudging the meteorological fields under the present era.

The emission fluxes used in this study are the Hemispheric Transport of Air Pollution (HTAP)-v2.2 (Janssen-Maenhout et al., 2015) for BC, organic carbon (OC) and $SO_2$ from anthropogenic sources in 2010 and the Global Fire Emission Database (GFED) version 4 (van der Werf et al., 2017) for BC, OC and $SO_2$ from biomass burning in climatological average from 2005 to 2014. The ratio of OC to OM is set at 1.6 for anthropogenic activities and 2.6 for biomass burning (Tsigaridis et al., 2014). Secondary organic aerosols (SOAs) are assumed to form particles, which are calculated by multiplying the emission fluxes of isoprene and terpenes provided by the Global Emissions Initiative (GEIA) (Guenther et al., 1990) using constant factors. $SO_2$ is emitted from volcanic eruptions (Diehl et al., 2012) and is also formed from DMS, which is interactively emitted in the aerosol module (Bates et al., 1987). Sulfate is formed from $SO_2$ oxidation with a 3-dimensional distribution of monthly oxidants (ozone, $H_2O_2$ and OH) provided by a chemical transport model (CHASER) coupled to MIROC (Sudo et al., 2002). Emission fluxes for dust (Takemura et al., 2009) and sea salt (Monahan et al., 1986) are interactively calculated in the model using mainly the wind speed at a height of 10 m.

In the preindustrial experiments, the anthropogenic emission fluxes of BC, OC and $SO_2$ are assumed to be zero in this study. Hoesly et al. (2018) estimated that the globally averaged emissions of anthropogenic sources in 1850 were 2.1% of the 2010 emissions for sulfate, 12.0% for BC, and 22.7% for OC. The residential sector has the largest contribution to the total anthropogenic emissions in the preindustrial era. Takemura (2020) calculated the IRFari due to anthropogenic sulfate under the conditions of 0% and 30% of the present emissions and found that the difference in the IRFari was within 0.03 Wm$^{-2}$. Therefore, differences in the assumptions for the preindustrial era between this study and other studies, such as IPCC-AR6 (Szopa et al., 2021), will result in a difference in the IRFari due to anthropogenic sources of at most 0.05 Wm$^{-2}$. Takemura (2020) also calculated ERFari and ERFaci due to anthropogenic sulfate under the conditions of 0% and 30% of the present emissions and found that the difference in ERFari plus ERFaci was within 0.2 Wm$^{-2}$. These are possible uncertainties in the

estimated radiative forcings due to anthropogenic sources in this study, but these magnitudes are smaller than the difference between NDW6 and NSW6 in this study, as shown in section 4.

## 2.4 Observations

The NICAM-simulated cloud, precipitation, and radiation fluxes at the top of the atmosphere (TOA) are evaluated by satellite
products. The satellite-based product of precipitation is provided by version 2.2 of the Global Precipitation Climatology Project (GPCP) with monthly 2.5°×2.5° grids (Adler et al., 2003). The satellite-based product of the LWP is provided by the Multisensor Advanced Climatology (MAC) Total Liquid Water Path L3 with monthly 1°×1° grids (Elsaesser et al., 2017). The ratio of the column precipitation to the sum of the column precipitation and cloud liquid water is calculated by CloudSat products of cloud liquid water and precipitation liquid water in 2C-RAIN-PROFILE (Lebsock and L'ecuyer, 2011). According
to Lebsock and L'ecuyer (2011), this product is more reliable than other Cloudsat products, such as 2C-RAIN-COLUMN, but this product is retrieved over only the ocean, and CloudSat cannot properly detect signals below a height of 1 km (Christensen et al., 2013; Huang et al., 2012; Liu, 2002). The COT and CDR at the warm-topped clouds are retrieved from the Moderate Resolution Imaging Spectroradiometer (MODIS) for all types of clouds (Platnick et al., 2015). The CF at a low level is estimated from datasets under the International Cloud Climatology Project (ISCCP; Rossow and Schiffer, 1999). The satellite-
based radiation fluxes, i.e., outgoing shortwave and longwave radiative flux (hereafter referred to as OSR and OLR) and shortwave and longwave cloud radiative forcing (hereafter referred to SWCRF and LWCRF), are provided by the Clouds and the Earth's Radiant Energy System (CERES) experiment onboard Terra and Aqua, as CERES_EBAF_Ed4.1, with 1°×1° grids (Loeb et al., 2009). For the comparisons in this study, these datasets are averaged for the 3 years from 2012-2014, except for approximately 6 yearly averages (June 2006 to April 2011) in 2C-RAIN-PROFILE and 5 yearly averages (2006-2010) in CDR.
The NICAM-simulated aerosols are evaluated by in situ measurements and satellite aerosol products. The climatological observations used in the evaluation of the simulated aerosol mass concentrations are provided by the Interagency Monitoring of Protected Visual Environments (IMPROVE; Malm et al., 1994) program in the United States, the European Monitoring and Evaluation Programme (EMEP) in Europe, the Acid Deposition Monitoring Network in East Asia (EANET) in Asia, and the China Meteorological Administration Atmosphere Watch Network (CAWNET; Zhang et al., 2012) in China. The
climatological observations used in the evaluation of the simulated AOT are provided by the Aerosol Robotic Network (AERONET; Holben et al., 1998), SKYNET radiometer network (Nakajima et al., 2020), and China Aerosol Remote Sensing Network (CARSNET; Che et al., 2015). The same datasets were prepared and used in Goto et al. (2020), which shows the location map and description in Table 1 and Figure 1. In the global aerosol validation, the level 3 AOT product of Collection 6 MODIS onboard the polar-orbiting satellite Terra (MOD08_L3) by Platnick et al. (2015) is used. The AOT is retrieved from
the deep blue (Hsu et al., 2013) and dark target (Levy et al., 2013) methods. The uncertainty of the retrieved AOT from both methods is similar to each other (Sayer et al., 2014) and estimated to be ± (0.05 + 0.15*AOT) (Levy et al., 2013). However, satellite-retrieved AOTs are still divergent among different sensors (Petrenko and Ichoku, 2013; Alfaro-Contreras et al., 2017;

Wei et al., 2019; Sogacheva et al., 2020), so the level 3 AOT product from collection F15_0031 (V22 level 3) of the Multiangle Imaging Spectroradiometer (MISR) onboard Terra by Kahn et al. (2010) is also used in this study. While MODIS has 36 bands from 0.41 µm to 14 µm, a single view and a broad swath of 2330 km, MISR has four bands (0.45 µm, 0.56 µm, 0.67 µm, and 0.87 µm) with nine cameras with the narrowest swath at 380 km. The uncertainty of MISR-retrieved AOT is estimated to be 0.05 or 0.2*AOT (Kahn et al., 2010). Wei et al. (2019) showed that the MODIS-retrieved AOT is the closest to AERONET, and the MISR-retrieved AOT is the second closest to AERONET among various satellite AOT products. Alfaro-Contreras et al. (2017) showed that the bias of the AOT between MODIS and MISR is found over the Southern Ocean, where the MISR-retrieved AOT is larger than the MODIS-retrieved AOT due to cloud contamination (Toth et al., 2013). Petrenko and Ichoku (2013) showed the large uncertainty of the MODIS-retrieved AOT over high albedo areas such as desert, snow, and ice surfaces. In East Asia, the MISR-retrieved AOT is lower than the AERONET-retrieved AOT, but the MODIS-retrieved AOT is higher than the AERONET-retrieved AOT (Kahn et al., 2010). The 3-dimensional distribution of the aerosol extinction coefficients obtained from the Cloud-Aerosol Lidar with Orthogonal Polarization (CALIOP)/Cloud-Aerosol Lidar and Infrared Pathfinder Satellite Observations (CALIPSO) version 3 provided by the NASA Langley Research Center (LaRC) are used in a 1° × 1° grid under clear-sky conditions (Winker et al., 2013). The CALIOP (version 3)-retrieved AOTs are sometimes compared with the MODIS (Collection 6)-retrieved AOTs in previous studies (Kim et al., 2018; Liu et al., 2018; Proestakis et al., 2018). Kim et al. (2018) show that the differences in the CALIOP (version 3)-retrieved AOT and MODIS-retrieved AOT are estimated to be -0.010 over ocean and +0.069 over land due to the inconsistency of the footprint resolution. Compared to the AERONET-retrieved AOT, the CALIOP-retrieved AOT is lower by 0.064. Therefore, over land, the CALIOP-retrieved AOT is underestimated, and the MODIS-retrieved AOT is overestimated. Liu et al. (2018) also showed that the CALIOP-retrieved AOT for polluted days in China is more reliable than the MODIS-retrieved AOT. Therefore, the difference in the retrieved AOT between MODIS, MISR, and CALIOP can be considered as the uncertainty of the satellite retrievals for AOT. These satellite datasets are averaged for the 3 years from 2012-2014.

## 2.5 Reference datasets

Our previous model results provided in Goto et al. (2020) using NICAM.16 at a global 14-km high resolution (hereafter referred to as the HRM) and a global 56-km low resolution (hereafter referred to as the LRM) are used as references to compare the NICAM results. As mentioned in section 2.1, the number of vertical layers is set at 38, and the timestep is 1 minute in both the HRM and LRM. The integration periods in both the HRM and LRM are 3 years as climatological runs. The emission inventories, i.e., 2010 for anthropogenic sources, climatological average in 2005-2014 for biomass burning, and natural sources in the present era, and the nudged SST and sea ice in this study are identical to those in both the HRM and LRM, but the initial conditions in this study are different from those in both the HRM and LRM, which use the model results at the end of December after a 1.5-month spin-up. The initial conditions for the model spin-up are prepared by the reanalysis datasets of the National Centers for Environmental Prediction (NCEP) Final (FNL) (Kalnay et al., 1996) in November 2011. In the cloud microphysics and autoconversion modules, NDW6 coupled to Seifert and Beheng (2006) and NSW6 coupled to Khairoutdinov and Kogan

(2000) are used in this study, whereas NSW6 coupled to Berry (1967) is used in both the HRM and LRM. The improvement in the aerosol module described in section 2.2 is also different from that in the HRM and LRM. The results of the HRM and LRM are useful for evaluating the current model results because the observations are limited in some parameters, such as aerosol global budgets and radiative forcings.

In addition to the results in Goto et al. (2020) as references for a comparison of global aerosol budgets and aerosol optical properties, results obtained from the AeroCom Phase-III project (Gliß et al., 2021) are used in this study. AeroCom Phase-III includes 14 global models and can be the best reference to evaluate global aerosol simulations. For references of the IRFari, the Max Planck Aerosol Climatology version 2 (MACv2 by Kinne, 2019) provides global maps for aerosol optical and radiative properties by calculating an offline radiative transfer model with the ensemble mean among the AeroCom global models and

the in-situ measurements of AERONET. Another reference for IRFari is the mean value from more than 10 studies based on the observations in Thorsen et al. (2021). The IRFari in Thorsen et al. (2021) is only estimated in the shortwave at the TOA.

## 3 Results and discussion

### 3.1 Precipitation and clouds

For simplicity, the simulated results in the numerical experiment with the NDW6 (or NSW6) cloud microphysics module are

expressed hereafter as "the NDW6(NSW6)-simulated results". First, the NICAM-simulated (i.e., both NDW6- and NSW6-simulated) precipitation and clouds are evaluated using satellite data. Figure 1 shows the zonal and horizontal distributions of the annual, January and July averages of precipitation. Table 1 includes the global and annual mean values of precipitation, which are estimated to be 3.01 mm day$^{-1}$ (NDW6), 2.78 mm day$^{-1}$ (NSW6), and 2.68 mm day$^{-1}$ (GPCP). These differences among NDW6, NSW6, and GPCP are also found in January and July. The main reason for these differences is the

overestimation of NICAM-simulated precipitation over the tropics. This tendency can be found in previous studies using other high-resolution models with finer horizontal resolutions (e.g., Stevens et al., 2019; Wedi et al., 2020).

Figure 2 shows the zonal and horizontal distributions of the annual, January and July averages of the LWP over only the oceans, whereas Figure 3 shows these differences among NDW6, NSW6, and MAC. The global and annual mean LWP values over only the oceans (60°S-60°N) are estimated to be 95.8 g m$^{-2}$ (NDW6), 104.4 g m$^{-2}$ (NSW6), and 119.6 g m$^{-2}$ (MAC). The zonal

and annual distributions of the NDW6-simulated LWP near the polar regions (> 45°S and > 45°N) are more comparable to the MAC results than to the NSW6 results. This feature is explained by the better reproducibility of supercooled liquid water in low-level mixed-phase clouds (Roh et al., 2020; Seiki and Roh, 2020; Noda et al., 2021). In the tropics where the LWP is larger than the other areas, the NDW6-simulated LWP is lower and not closer to the MAC results than the NSW6-simulated LWP. Notably, the MAC results contain regional biases of up to 25%, especially in the tropics (Elsaesser et al., 2017), but

even with the largest errors, the NDW6- and NSW6-simulated LWPs in the tropics are still underestimated compared to the MAC results. In the horizontal distribution over the eastern Pacific Ocean and Southern Atlantic Ocean at lower latitudes (30°S-0), the NDW6-simulated LWP is lower than the NSW6 results but comparable to the MAC results. However, over the

western Pacific Ocean and Indian Ocean at the lower latitudes, both NDW6- and NSW6-simulated LWPs are lower than the MAC results. Therefore, the overestimation of the NSW6-simulated LWP in the eastern Pacific Ocean and Southern Atlantic

Ocean effectively balanced the underestimation in the western Pacific Ocean and Indian Ocean, which led to zonal LWP values that were closer to the MAC results. This situation also occurs in the northern hemisphere at lower latitudes (30°N-0). Therefore, in the lower latitudes (30°S-30°N), the zonal averages of the NSW6-simulated LWP look closer to the MAC results, but this is attributed to the compensation errors in the regional distribution. As a result, the global and annual mean values of the NSW6-simulated LWP appear closer to the MAC results.

Table 1 includes other cloud information (COT, CF at the low level, and CDR at warm-topped clouds). Both NDW6- and NSW6-simulated COTs in annual, January and July global mean values are underestimated compared to the MODIS results. This tendency is similar to the results of the LWP. In the spatial distribution, the NDW6-simulated COT has a lower bias over midlatitude to polar regions, whereas the NSW6-simulated COT has a lower bias in other areas (not shown). For low-altitude CF, the differences between the NDW6- and NSW6-simulated results are very small, and both results are underestimated

compared to the ISCCP results. Therefore, the difference in the cloud microphysics module has almost no impact on the CF. For CDR at warm-topped clouds, both NDW6- and NSW6-simulated results in annual, January and July global mean values are underestimated compared to the MODIS results.

In summary, the global and annual mean values of the NDW6 simulation include biases of +12% in precipitation, -20% in the LWP, -45% in the COT, -28% in the CF at low levels, and -29% in the CDR at warm-topped clouds. The biases in the NSW6

simulation have the same sign, but their magnitudes are slightly different (+4% in the precipitation, -13% in the LWP, -35% in the COT, -27% in the CF at low levels, and -42% in the CDR at warm-topped clouds). These mean values are useful for discussing differences among global climate models in terms of the global budget, but they generally include compensation errors in space, as explained above. Therefore, the results of precipitation in both NDW6 and NSW6 are comparable to the observations, but those of LWP in NDW6 are different from those in NSW6. The NDW6-simulated LWPs are generally closer

to the observations, except for the tropics.

## 3.2 Mass loading of aerosols

NICAM-simulated aerosols are evaluated by statistical metrics, including the Pearson correlation coefficient (PCC), normalized mean bias (NMB), and root-mean-square error (RMSE), defined as (A1), (A2) and (A3) in Appendix A. Figure 4 shows scatterplots of the surface mass concentrations of the NICAM-simulated and observed aerosols. For OM, the calculated

statistical metrics in NDW6 are 0.847 (PCC), 3.40 µg m$^{-3}$ (RMSE), and -30.4% (NMB), and the difference between NDW6 and NSW6 is very small. For BC, the calculated statistical metrics in NDW6 are 0.904 (PCC), 1.05 µg m$^{-3}$ (RMSE), and -53.4% (NMB). The difference in the simulated BC between NDW6 and NSW6 is also very small. For sulfate, the calculated statistical metrics in NDW6 are 0.807 (PCC), 3.97 µg m$^{-3}$ (RMSE), and -10.4% (NMB), whereas those in NSW6 are 0.853 (PCC), 3.67 µg m$^{-3}$ (RMSE), and -3.7% (NMB).

Figure 5 and Table A2 indicate global and annual mean values of column burden, emission, and atmospheric lifetime, which
       are calculated by the ratio of column burden to total deposition amount. The column burdens of the NDW6- and NSW6-
       simulated dust range within the uncertainty of the recent models participating in the AeroCom Phase-III project (Gliß et al.,
       2021). The amount of dust emissions and the dust lifetime in all NICAM simulations range within the uncertainty obtained
       from the AeroCom models. The difference in the dust column burden between NDW6 and NSW6 is 23%, which is mainly

caused by the 10% difference in emissions between NDW6 and NSW6 due to the difference in the simulated wind. Since the
       dust emission is approximately proportional to the cubic wind speed at a height of 10 m, only a 3.2% difference in the wind
       speed in the case of a 10 ms$^{-1}$ average causes a 10% difference in the dust emission strength.

       For sea salt, the differences in the column burden, emission, and lifetime among the NICAM simulations are not as large and
       range within the uncertainty of the references. However, the emission flux of the NDW6-simulated sea salt is higher than that

of the NSW6-simulated sea salt, whereas the column burden of the NDW6-simulated sea salt is lower than that of the NSW6-
       simulated sea salt. This is mainly caused by the difference in wet deposition (see Appendix Table A2). The difference in wet
       deposition is strongly affected by the difference in the ratio of column precipitation to the sum of the column precipitation and
       column liquid cloud water (RPCW) between NDW6 and NSW6, as shown in Figure 6. The NDW6-simulated RPCW is larger
       than that of the NSW6 result, which is easy to see from the results of Figures 1 and 2. Because the NSW6-simulated clouds

are larger in most regions except for in the tropics, the NDW6-simulated RPCW is much closer to the CloudSat-retrieved
       RPCW. In the western Pacific Ocean over the tropics where the simulated aerosols are low, the NSW6 results are closer to the
       CloudSat results. An increase in the RPCW leads to an increase in the aerosols that are dissolved into raindrops and are
       removed from the atmosphere. Therefore, NDW6-simulated clouds and precipitation cause more wet deposition of simulated
       aerosols compared to the NSW6 results.

Emissions of OM and BC are given from the database, so the differences in the column burden and lifetime are mainly
       discussed. The column burdens of the NDW6-simulated OM and BC, including water-soluble BC (WSBC) and water-insoluble
       BC (WIBC), are always lower than the NSW6 results. The lifetimes of the NDW6-simulated OM and BC are always shorter
       than those of the NSW6 results. The differences in the column burden as well as the lifetimes of OM and BC between NDW6
       and NSW6 are at most 15%. All the results simulated by the NICAM are within the uncertainty of the AeroCom models but

are relatively lower than the medians and averages among the AeroCom models. The BC lifetimes are 5.4 days (NDW6) and
       6.3 days (NSW6). They range from 2.9 days to 8.7 days (median 5.5 days) in the AeroCom models (Gliß et al., 2021).

       Sulfate is a secondary component and is formed from $SO_2$ oxidation in the atmosphere and within clouds. Its complexity results
       in different features from other primary species. The column burden of sulfate is 0.45 TgS (NDW6) and 0.52 TgS (NSW6).
       The results range from 0.22 TgS to 0.98 TgS (0.60 TgS at the median) in the AeroCom models (Gliß et al., 2021). The lifetimes

of sulfate are 2.9 days (NDW6) and 3.3 days (NSW6). They range from 1.8 days to 7.0 days (median 4.9 days) in the AeroCom
       models (Gliß et al., 2021). To understand the difference in the column burden and lifetime of sulfate between different schemes
       and resolutions, $SO_2$, as a precursor of sulfate, becomes an important factor. The column burden of NDW6-simulated $SO_2$ is
       0.28 TgS, which is 19% lower than the NSW6 result. Therefore, the difference in the column burden of sulfate between NDW6

and NSW6 is mainly caused by the difference in the column burden of $SO_2$ because the difference is very small in the wet deposition of sulfate between NDW6 and NSW6 (Table A2). The difference in the column burden of $SO_2$ between NDW6 and NSW6 is caused by the chemical loss in the aqueous phase (0.5 TgS yr$^{-1}$ or +1%) and gas phase (-1.3 TgS yr$^{-1}$ or -10%) and wet deposition (0.5 TgS yr$^{-1}$ or +23%), as shown in Table A2. The differences between NSW6 and HRM are mentioned in Appendix A.

**3.3 Aerosol optical properties**

Figure 7 shows a global comparison of annual, January, and July averages of both NDW6- and NSW6-simulated AOTs with ground-based measurements (AERONET, SKYNET, and CARSNET). The model performance of both NDW6- and NSW6-simulated AOT is very good, with a high correlation (the PCC value is 0.662 to 0.807 in NDW6 and 0.721 to 0.837 in NSW6), moderate uncertainty (the RMSE value is 0.13 to 0.23 in NDW6 and 0.12 to 0.16 in NSW6), and moderate bias (the NMB value is -24.1% to +27.5% in NDW6 and -8.9% to -5.0% in NSW6). These values are much better than those reported in Goto et al. (2020) (e.g., PCC values of 0.471 to 0.589, RMSE values of 0.21 to 0.23, and NMB values of -44.1% to -5.4%), as shown in Appendix B.

Figure 8 shows horizontal distributions of the annual averages in AOT in both NICAM simulations under all-sky and clear-sky conditions and satellite observations of MODIS and MISR onboard TERRA. Generally, both NDW6- and NSW6-simulated AOTs are comparable to the satellite results. As shown in Figure 7, the NDW6-simulated AOT is lower than the NSW6 result. The AOT under all-sky conditions tends to be larger than the AOT under clear-sky conditions, mainly because the relative humidity (RH) under all-sky conditions is generally higher than the RH under clear-sky conditions (Dai et al., 2015). Over the outflow regions of North Africa over the Atlantic Ocean, both the NDW6- and NSW6-simulated AOTs are generally comparable to the satellite results. Over East China, Russia, and Central Asia, there are relatively large differences among the NICAM-simulated, MODIS-retrieved and MISR-retrieved AOTs. As explained in section 2.2, over land such as East China, near the Arctic such as Russia, and in desert areas such as Central Asia, the MODIS-retrieved AOTs tend to be higher than the MISR-retrieved AOTs (Kahn et al., 2010; Shi et al., 2011; Petrenko and Ichoko, 2013). Over the Southern Ocean, where the MISR-retrieved AOT includes cloud contamination (Toth et al., 2013; Alfaro-Contreras et al., 2017), both the NDW6-simulated and NSW6-simulated AOTs are lower than the MISR-retrieved results and comparable to the MODIS-retrieved results. The simulated AOT compositions are also compared with the references of the AeroCom models in Appendix C.

Figure 9 indicates the vertical profiles of the aerosol extinction coefficients in regional and annual averages. Since the CALIOP-retrieved results above a 5 km height include some bias (Watson-Parris et al., 2018), the discussion is focused on the results below a 5 km height. Large differences between NDW6 and NSW6 are found in South Asia (India and Southeast Asia), Africa (the coast of North Africa, North Africa, the coast of Central Africa, and South Africa), and South America, where the NDW6-simulated aerosols are lower than the NSW6-simulated results. In South Asia (Figures 9d and 9i), the vertical profiles

of both NDW6- and NSW6-simulated aerosol extinction coefficients are comparable to the CALIOP-retrieved results with peak heights of 0.5-1 km. In East China (Figure 9e), the vertical profiles of both NDW6- and NSW6-simulated aerosol extinction coefficients are different from those obtained from CALIOP, which has low aerosols below 2 km height. These

CALIOP (version 3) retrieval results may include biases because CALIOP (version 4) improved this underestimation in East China (Kim et al., 2018). Along the coast of North Africa, both NDW6- and NSW6-simulated aerosols are comparable to the CALIOP-retrieved results (Figure 9g), although in the dust source area in North Africa (Figure 9h), they are overestimated compared to the CALIOP-retrieved results. This may be one problem of CALIOP retrievals over desert areas where the assumed lidar ratio of pure dust is low (Schuster et al., 2012). In the biomass burning areas (the coast of central Africa, South

America, and South Africa), as shown in Figures 9(j), 9(k), and 9(l), the heights at which the extinction coefficient decays (called 'decay height') in the CALIOP results are much more reliable than the vertical profiles of the CALIOP-retrieved extinction coefficient because CALIOP cannot detect the signal below the optically thick layers (Ma et al., 2013). The decay heights of the NICAM-simulated extinction coefficients are lower along the coasts of central Africa and South Africa and higher in South America compared to the CALIOP results. This large bias of the vertical profile indicates a problem of the

vertical transport of aerosols originating from biomass burning in the NICAM, which may not be solved by the improvement of the cloud microphysics module and finer resolution of the model grids. The differences between NSW6 and HRM are mentioned in Appendix B.

## 4 Radiative forcing

This section discusses the impacts of aerosols on radiation through ARI and ACI by comparing them with references obtained

from both models and satellites. These comparisons verify the usefulness of the NICAM aerosol model coupled with both the NDW6 and NSW6 modules for climate simulations.

### 4.1 Aerosol-Radiation Interaction (ARI)

Figure 10 shows the shortwave and longwave instantaneous radiative forcing of the ARI (IRFari) at the TOA and the surface in the NICAM and references (HRM and LRM in Goto et al., 2020; MACv2 in Kinne, 2019; observational estimates in Thorsen

et al., 2021). The magnitudes of the IRFari values among all the NICAM-simulated dust values under both all-sky and clear-sky conditions at the TOA are larger than the reference results (Kinne, 2019). For example, the shortwave IRFari dust values at the TOA under all-sky conditions are calculated to be -0.46 $Wm^{-2}$ (NDW6), -0.57 $Wm^{-2}$ (NSW6), and -0.24 $Wm^{-2}$ (Kinne, 2019). This is partly caused by the weaker absorption of AOT and the higher dust AOT in this study compared to the median value of the AeroCom models, as shown in Figure C1. In contrast, at the surface, the magnitudes of both shortwave and

longwave IRFari values among all the NICAM-simulated dust values under both all-sky and clear-sky conditions are smaller than the results in Kinne (2019). This is consistent with too little shortwave absorption, but this is inconsistent with the results of the larger column burden and AOT of dust in this study compared to those of the AeroCom models in Figures 5 and C1. The comparison with the results of Kinne (2019) may imply a much higher mass extinction coefficient of the dust or bias of

the simulated dust size distribution, as noted by Kok et al. (2017), who concluded that the simulated dust in current global models is too fine. For other absorption components, i.e., POM+WSBC and WIBC, the NSW6-simulated IRFari values are higher than the NDW6 results. Under all-sky conditions, both NDW6- and NSW6-simulated IRFari values due to POM+WSBC and WIBC are positive because of an increase in absorption in the presence of clouds. At the surface, the difference in the IRFari values among all the NICAM simulations has the same tendency as that obtained from the difference in the column burden or AOT. For SOA, as the other component of carbonaceous aerosols and nonlight-absorbing matter, the difference in the IRFari values among all NICAM simulations generally has the same tendency as that obtained from the difference in carbonaceous aerosols. For other nonlight-absorbing components, i.e., sea salt and sulfate, the difference in the IRFari values between the TOA and the surface is very small. At the TOA and the surface, the magnitudes of the NDW6-simulated IRFari values in both shortwave and longwave under both the all-sky and clear-sky conditions are lower than those of the NSW6 results. This is consistent with the results of the column burden (Figure 5) and AOT (Figure C1). The shortwave IRFari values due to sea salt under all-sky conditions are estimated to be -0.56 Wm$^{-2}$ (NDW6), -0.65 Wm$^{-2}$ (NSW6), and -0.72 Wm$^{-2}$ (Kinne, 2019). If the estimation by Kinne (2019) is assumed to be real, the NICAM-simulated AOT of sea salt is underestimated by 10-20%, probably because the NICAM underestimates the column burden of sea salt, which may be due to its short lifetime relative to the values from Kinne (2019) (Figure 5c). This may suggest that the NICAM-simulated sea salt is scavenged more by wet deposition, possibly due to high precipitation in the NICAM (Figure 1). For sulfate, the shortwave IRFari values under all-sky conditions are estimated to be -0.51 Wm$^{-2}$ (NDW6), -0.60 Wm$^{-2}$ (NSW6), and -0.83 Wm$^{-2}$ (Kinne, 2019). This is consistent with the results of lower values of both the column burden and AOT of sulfate among the reference models (Figures 5b and C), which is caused by the lower lifetime of sulfate among the AeroCom models (Figure 5c).

Overall, the IRFari values due to all aerosols under all-sky conditions are estimated to be -1.57 Wm$^{-2}$ (NDW6) and -1.86 Wm$^{-2}$ (NSW6), -1.92 Wm$^{-2}$ (from -3.1 Wm$^{-2}$ to -0.61 Wm$^{-2}$ in Thorsen et al., 2021), and -1.10 Wm$^{-2}$ (Kinne, 2019). The magnitude of the IRFari by Kinne (2019) is lower than the other estimates because the light-absorbing effect is higher in this reference than in the others. The NSW6-simulated shortwave IRFari value is close to the reference value obtained from observational estimates in Thorsen et al. (2021), whereas the NDW6-simulated shortwave IRFari value is lower than the median value of Thorsen et al. (2021) by approximately 0.4 Wm$^{-2}$. The differences in the IRFari values between NDW6 and NSW6 are 0.29 Wm$^{-2}$ (shortwave), 0.03 Wm$^{-2}$ (longwave), and 0.26 Wm$^{-2}$ (sum of shortwave and longwave), which are approximately 16% (shortwave), 14% (longwave), and 16% (sum) of the total IRFari value in NDW6. For anthropogenic aerosols, the shortwave IRFari values under all-sky conditions are estimated to be -0.38 Wm$^{-2}$ (NDW6), -0.45 Wm$^{-2}$ (NSW6), and -0.63 Wm$^{-2}$ (from -0.11 Wm$^{-2}$ to -1.00 Wm$^{-2}$ in Thorsen et al., 2021). The difference in IRFari values between NDW6 and NSW6 is 0.07 Wm$^{-2}$ (shortwave), 0.00 Wm$^{-2}$ (longwave), and 0.06 Wm$^{-2}$ (sum of shortwave and longwave), which is approximately 15% (shortwave), 3% (longwave), and 16% (sum) of the total IRFari value in NDW6. The magnitudes of both NDW6- and NSW6-simulated shortwave IRFari values range within the uncertainty but are lower than the median of Thorson et al. (2021). This difference in the total IRFari between NDW6 and NSW6 is caused by the difference in the simulated dust, sea salt and sulfate, as shown in section 3. The difference between NICAM and the reference is mainly attributed to the lower value of the column

burden of the simulated sulfate. In conclusion, the magnitudes of both the NDW6- and NSW6-simulated IRFari values are within the uncertainty of the references, even if the uncertainty is caused by the assumption in the preindustrial days, as

mentioned in section 2.3. The difference in the IRFari values between NDW6 and NSW6 is up to 20%. In addition, the difference in the IRFari values between NDW6 and NSW6 is larger than the difference between the HRM and LRM in Goto et al. (2020), as mentioned in Appendix D. Therefore, the model development of the cloud microphysics module is important. Figure 11(a) and Table 2 show the global and annual mean ERFari values due to anthropogenic aerosols. The NICAM-simulated ERFari values (-0.19 $Wm^{-2}$ in NDW6 and -0.23 $Wm^{-2}$ in NSW6) are comparable to the value of -0.3±0.3 $Wm^{-2}$

shown in IPCC-AR6 (Forster et al., 2021). The NICAM-simulated shortwave ERFari values (-0.22 $Wm^{-2}$ in NDW6 and -0.26 $Wm^{-2}$ in NSW6) are slightly underestimated compared to the lower limit of the references (-0.25 $Wm^{-2}$) provided by the observational estimates in Thorsen et al. (2021), given the uncertainties for preindustrial emissions in this study. The difference between NDW6 and NSW6 is calculated to be 0.04 $Wm^{-2}$ (approximately 16%). Under clear-sky conditions, the NICAM-simulated values (-0.52 $Wm^{-2}$ in NDW6 and -0.60 $Wm^{-2}$ in NSW6) in shortwave are smaller in magnitude than the lower limit of the references (-0.67 $Wm^{-2}$) in Thorsen et al. (2021). The difference between NDW6 and NSW6 is calculated to be 0.08

$Wm^{-2}$ (approximately 13%). Even in the net, i.e., sum of shortwave and longwave, the NICAM-simulated values are -0.47 $Wm^{-2}$ in NDW6 and -0.55 $Wm^{-2}$ in NSW6, and the calculated difference between NDW6 and NSW6 is 0.08 $Wm^{-2}$ (approximately 13%). In summary, the difference in the ERFari values between NDW6 and NSW6 is up to 15% or 0.08 $Wm^{-2}$.

## 4.2 Aerosol-Cloud Interaction (ACI)


Before evaluating the simulated radiative forcings due to ACI, the simulated cloud radiative forcing (CRF) and total radiation fluxes are compared and evaluated. As shown in Table 1, the global averages of the SWCRF for January are estimated to be -48.4 $Wm^{-2}$ (NDW6), -49.3 $Wm^{-2}$ (NSW6), and -50.4 $Wm^{-2}$ (CERES), whereas the global averages of the SWCRF for July are estimated to be -41.8 $Wm^{-2}$ (NDW6), -48.7 $Wm^{-2}$ (NSW6), and -44.5 $Wm^{-2}$ (CERES). The difference in the SWCRF between

NDW6 and NSW6 is 0.9 $Wm^{-2}$ in January and 6.9 $Wm^{-2}$ in July. The difference in the SWCRF between NICAM and CERES in January is 2.0 $Wm^{-2}$ (NDW6) and 1.1 $Wm^{-2}$ (NSW6), whereas the difference in the SWCRF between NICAM and CERES in July is 2.7 $Wm^{-2}$ (NDW6) and -4.2 $Wm^{-2}$ (NSW6). At 30°S-30°N latitudes and in annual averages, the NDW6-simulated SWCRF values are underestimated compared to the CERES results, whereas the NSW6-simulated SWCRF values are overestimated and more comparable to the CERES results. At other latitudes and in annual averages, the NDW6-simulated

SWCRF is comparable to the CERES results, whereas the NSW6-simulated SWCRF values are underestimated compared to the CERES result. The global and annual averages of SWCRF are estimated to be -42.5 $Wm^{-2}$ (NDW6), -45.9 $Wm^{-2}$ (NSW6), and -45.7 $Wm^{-2}$ (CERES). The NSW6-estimated SWCRF value is highly comparable to the CERES result but includes large compensation errors in the regional distribution. The details of the spatiotemporal characteristics are discussed in Appendix E. The NDW6-estimated SWCRF values are concluded to be better than the NSW6 results, but the underestimation of the NDW6-

simulated SWCRF is mainly caused by the underestimation of the simulated LWP due to the underestimation of the simulated

CDR shown in Table 1. The impacts of this negative biases in the simulated SWCRF and LWP on the aerosol simulations are still unclear due to complex interactions between aerosols, clouds, and precipitation. For OSR, OLR, and LWCRF, the validation using CERES results is also shown in Appendix E. The underestimation of the simulated LWCRF is caused by the underestimation of the simulated high-level clouds, but the impacts of this negative biases in the simulated LWCRF on the aerosol simulations are unclear due to ignorance of the interaction between aerosols and ice crystals (as ice nuclei) in this model.

Given the verification of the NICAM-simulated CRF above, the simulated ACI due to anthropogenic aerosols is discussed by comparing the results between NDW6 and NSW6 for simulations with aerosol and precursor gas emissions for the preindustrial (PI), mentioned in section 2.3, and the present day (PD). Figure 12 shows the global maps of changes in the simulated CCN at 1-km heights, CDNC at 1-km heights only for NDW6, CDR at 1-km heights, LWP, CA, CF at 1-km height and net ERFaci between PD and PI. Figure 13 also shows the average values of the selected regions. These figures show that the global average of the NDW6-calculated $\partial$CCN at a 1-km height is estimated to be 16.70 cm$^{-3}$ ($\partial$CCN), whereas that in NSW6 is estimated to be 19.59 cm$^{-3}$ ($\partial$CCN). The NDW6-calculated $\partial$CCN values are lower than the NDW6 results. In $\partial$CDNC, the NDW6-estimated values are +0.70 cm$^{-3}$ (global), +4.22 cm$^{-3}$ (the United States), +4.58 cm$^{-3}$ (Europe), +3.57 cm$^{-3}$ (East Asia), and +0.34 cm$^{-3}$ (India). However, the CDNC used in NSW6 is equal to the CCN concentrations due to the ignorance of sink processes in the CDNC in NSW6, as mentioned in section 2.1, so the difference in $\partial$CDNC between NDW6 and NSW6 is very large. The NDW6-estimated $\partial$CDR is -0.62 μm (global), -2.34 μm (the United States), -2.48 μm (Europe), -2.42 μm (East Asia), and -2.03 μm (India), whereas the NSW6-estimated $\partial$CDR is -0.31 μm (global), -1.06 μm (the United States), -1.04 μm (Europe), -1.19 μm (East Asia), and -0.68 μm (India). As shown in Figure 12, the NDW6- and NSW6-estimated $\partial$CDR values are negative near the industrial regions where the $\partial$CCN is large. For example, in the United States, the NSW6-simulated $\partial$CDNC (=$\partial$CCN) is approximately 60 cm$^{-3}$ and the NSW6-simulated $\partial$CDR is approximately -1.1 μm, whereas the NDW6-simulated $\partial$CDNC is approximately 4 cm$^{-3}$ and the NDW6-simulated $\partial$CDR is approximately -2.3 μm. The difference in the $\partial$CDNC-$\partial$CDR relationship between NDW6 and NSW6 is caused by the difference in the baseline of CDNC and CDR. The NDW6-simulated CDNC under both the PD and PI aerosol conditions is much lower than the NSW6-simulated results, whereas NDW6-simulated CDR under both the PD and PI aerosol conditions is larger than the NSW6-simulated results.

To evaluate the Twomey effect in NDW6 and NSW6, the global averages of differences in the mixing ratios, number concentrations, and CDR for liquid clouds between the PD and PI aerosol conditions are plotted in Figure 14. The changes of liquid cloud water mixing ratio ($\partial$Qc) in both NDW6 and NSW6 are positive at most heights, so Qc increases as aerosols increase. This is consistent with the results of $\partial$LWP shown in Figures 12 and 13(e). The largest value of $\partial$Qc in both NDW6 and NSW6 occurs at a height of approximately 1.5 km. Above a height of 3 km, $\partial$Qc in NDW6 is positive, whereas $\partial$Qc in NSW6 is close to zero or negative. This difference in $\partial$Qc between NDW6 and NSW6 is possibly caused by the differences in the simulated supercooled liquid water in mixed-phase clouds, as mentioned in section 3.1. For $\partial$CDNC, the largest values in NDW6 occur at a height of 1.2 km, which is slightly lower than the height where the largest value of $\partial$Qc occurs. This reflects the vertical structure of typical clouds in NDW6. In contrast, the vertical profile of $\partial$CDNC in NSW6 is different from that of

$\partial$Qc because NSW6 cannot predict CDNC and adopts $\partial$CCN. Specifically, above a height of 3 km, $\partial$Qc is close to zero, but $\partial$CDR is not zero because $\partial$CDNC has a positive value. Even though the magnitude of $\partial$CDR in NSW6 is lower than that in NDW6, this represents possible overestimation of the Twomey effect in NSW6.

As mentioned above, the NDW6-calculated $\partial$LWP values are higher than the NSW6 results by three times in global averages. The NDW6-estimated values are +2.12 g m$^{-2}$ (global), +7.52 g m$^{-2}$ (the United States), +15.45 g m$^{-2}$ (Europe), +8.77 g m$^{-2}$

(East Asia), and +3.36 g m$^{-2}$ (India), whereas the NSW6-estimated values are +0.65 g m$^{-2}$ (global), +4.96 g m$^{-2}$ (the United States), +2.52 g m$^{-2}$ (Europe), +2.62 g m$^{-2}$ (East Asia), and -0.44 g m$^{-2}$ (India). The positive values in $\partial$LWP in both NDW6 and NSW6 could be caused by a decrease in auto-conversion due to the increase in CDNC. However, magnitudes of $\partial$LWP differ between NDW6 and NSW6, which is the largest in Europe among others, whereas the NDW6- and NSW6-simulated $\partial$CCN are close to each other in most regions. This appears to indicate that the cloud water susceptibility, defined as the

difference in $\partial$LWP against $\partial$CCN from PD to PI conditions, is larger in NDW6 than in NSW6. Such a different susceptibility could be interpreted in terms of different complexities of hydrometeors interactions between NSW6 and NDW6, particularly whether or not the CDNC and rain drop number concentration (RDNC) are predicted. This generates different variabilities of CDNC and RDNC between the two schemes, possibly leading to the different susceptibilities. Nevertheless, more detailed analysis will be required in future studies to explore microphysical processes responsible for these different behaviors between

the two schemes.

The horizontal distribution of changes in the simulated ERFaci is generally consistent with changes in the simulated $\partial$LWP (Figure 12). In both NDW6 and NSW6, by decreasing the simulated $\partial$CDR, increasing the simulated $\partial$LWP from PI to PD, and increasing the simulated $\partial$CA and $\partial$CF at 1-km height, the negative values of the simulated ERFaci in industrial regions, such as the United States, Europe, and East Asia, increase in magnitude. The global annual averages of the net ERFaci value

are estimated to be -1.28 Wm$^{-2}$ (NDW6) and -0.73 Wm$^{-2}$ (NSW6). Both NDW6- and NSW6-estimated ERFaci values range within the results in IPCC-AR6 (Forster et al., 2021), i.e., -0.84 Wm$^{-2}$ (-1.45 Wm$^{-2}$ to -0.25 Wm$^{-2}$), and the Radiative Forcing Model Intercomparison Project (RFMIP) (Smith et al., 2020), i.e., -0.81±0.30 Wm$^{-2}$. The magnitude of the ERFaci value in NDW6 is larger than that in NSW6 by 0.55 Wm$^{-2}$ (approximately 43% of the ERFaci value in NDW6), whereas the NDW6-simulated aerosol loadings are smaller than the NSW6 results, as shown in the previous sections. Figure 13 shows that the

negative NDW6-estimated ERFaci values are larger than the NSW6-estimated ERFaci values by 2.33 Wm$^{-2}$ (US), 3.22 Wm$^{-2}$ (Europe), 1.10 Wm$^{-2}$ (East Asia), and 0.89 Wm$^{-2}$ (India). Therefore, it was suggested that the ERFaci due to both the Twomey and cloud lifetime effects in NDW6 was larger than that in NSW6, although the NSW6-simulated ERFaci certainly includes some bias due to the overestimation of the Twomey effect.

Other possible reasons for the differences in the ERFaci between NDW6 and NSW6 are discussed. Carslaw et al. (2013) and

Wilcox et al. (2015) pointed out that the different baselines of aerosol fields can provide small differences in ERFaci between two simulations. As mentioned in the previous sections for aerosols, the NDW6-simulated aerosols are generally lower than the NSW6 results, for example IRFari is approximately 15% lower. However, the baseline of CCN at 1-km height between

NDW6 and NSW6 under the PI conditions is not very different, so the difference in the baseline of aerosols between NDW6 and NSW6 does not cause the difference in ERFaci between the two simulations.

The difference in the autoconversion from clouds to precipitation between NDW6 and NSW6 can be a reason for the difference in ERFaci between NDW6 and NSW6. Using a global aerosol model, MIROC, coupled to a double-moment bulk cloud microphysics scheme with coarse resolution of $1.4° \times 1.4°$, the difference in ERFaci between Khairoutdinov and Kogan (2000) and Seifert and Beheng (2006) is estimated to be 0.15 Wm$^{-2}$ (Michibata and Suzuki, 2020). This magnitude of ERFaci difference potentially caused by the two different autoconversion schemes cannot explain the difference in ERFaci between

NDW6 and NSW6 of this study.

## 5 Summary

To estimate the impacts of cloud microphysics modules on aerosols and their radiative forcing, 6-year simulations of aerosols are performed using two different types of cloud microphysics schemes, i.e., the double-moment bulk cloud microphysics module (NDW6) and the single-moment bulk module with 6 water categories (NSW6), in the NICAM at a 14 km grid spacing.

The previous study by Goto et al. (2020) also simulated aerosols at a 14 km grid spacing. The NICAM used in this study was updated from our previous study of Goto et al. (2020), which also simulated aerosols at a 14 km grid spacing in terms of the cloud microphysics module (from NSW6 to NDW6), the vertical resolution (from 38 layers to 78 layers) and some aerosol modules (sulfate and dust).

The model performance of the surface aerosol mass concentrations and AOT are evaluated with in situ measurements by

statistical metrics of correlation (PCC), bias (NMB), and uncertainty (RMSE). The model performances of both NDW6-simulated surface mass and NSW6-simulated surface mass as well as AOT are very good, with moderate to high correlation, low to moderate uncertainty, and low to moderate bias. The differences between NDW6 and NSW6 are small, but they are greatly improved from the previous study of Goto et al. (2020). For example, the PCCs between the simulated and observed AOTs in annual averages are 0.807 (NDW6) and 0.837 (NSW6), which are much higher than 0.471 (HRM) and 0.356 (LRM)

in Goto et al. (2020). The reason for these improvements in this study is not only the update from Goto et al. (2020) but also the increase in available computational resources (using the supercomputer Fugaku in this study), resulting in approximately 12 times faster computation time than the supercomputer K in Goto et al. (2020).

The NDW6-simulated aerosol distributions are generally lower than the NSW6 results. For example, the global and annual mean values of the simulated AOT under all-sky conditions are estimated to be 0.127 (NDW6) and 0.153 (NSW6), which

range within the model uncertainty of the AeroCom models. These differences among the NICAM experiments with different cloud microphysics modules, i.e., NDW6 and NSW6, are caused by a different ratio of column precipitation to the sum of the column precipitation and column liquid cloud water or RPCW, which strongly determines the wet deposition in the aerosols. Since the NDW6-simulated LWP is generally lower than the NSW6 result and the NDW6-simulated precipitation is generally comparable to the NSW6 result, the scavenging effect of the aerosols in NDW6 is larger than that in NSW6. The NDW6-

simulated RPCW, precipitation and LWP are generally closer to the satellite-retrieved results compared to the NSW6 result,

although their global and annual mean values in NDW6 are sometimes no closer to the observation than the NSW6 results due to compensation errors in space.

The differences in the dust emissions, dust column burden and $SO_2$, AOT, and IRFari values for total aerosols between NDW6 and NSW6 are larger than those in the other aerosol budgets and components. For example, the net IRFari values due to all aerosols under all-sky conditions are estimated to be -1.36 $Wm^{-2}$ (NDW6), -1.62 $Wm^{-2}$ (NSW6), and -1.92 $Wm^{-2}$ (from -3.1 $Wm^{-2}$ to -0.61 $Wm^{-2}$ in Thorsen et al., 2021). The difference in IRFari values between NDW6 and NSW6 is 0.26 $Wm^{-2}$, which is at most 20% of the total IRFari value in NDW6. The ERFari values due to anthropogenic aerosols under all-sky conditions are estimated to be -0.19 $Wm^{-2}$ (NDW6), -0.23 $Wm^{-2}$ (NSW6), and -0.25 $Wm^{-2}$ (-0.45 $Wm^{-2}$ to -0.05 $Wm^{-2}$) shown in IPCC-AR6 (Forster et al., 2021). The difference between the NICAM and the reference may be larger, given the uncertainties in the preindustrial emissions in this study. The difference in NDW6 and NSW6 is probably caused by the difference in the simulated dust and sulfate in the present study, as shown in section 3. The difference in the dust between NDW6 and NSW6 is mainly caused by the difference in the emission fluxes due to the difference in the simulated wind, whereas those in the sulfate are mainly caused by the differences in the wet deposition of $SO_2$. A large difference among the experiments is also found in the interaction between aerosols and clouds, ERFaci, in which the global annual mean values in the net are estimated to be -1.28 $Wm^{-2}$ (NDW6) and -0.73 $Wm^{-2}$ (NSW6). The difference in the net ERFaci values between NDW6 and NSW6 is 0.55 $Wm^{-2}$ (approximately 43% of the ERFaci value in NDW6). This difference is larger than that in ERFari, which is approximately 20% or 0.04 $Wm^{-2}$. The regional differences in ERFaci between NDW6 and NSW6 are found to be large in the industrial areas, where the NDW6-simulated ERFaci values are negatively larger than the NSW6-simulated results. As discussed in section 4.2, it was suggested the increase in changes in ERFaci due to both the Twomey and cloud lifetime effects in NDW6 is larger than that in NSW6, although the NSW6-simulated ERFaci certainly includes some bias due to the overestimation of the Twomey effect. The different susceptibility between NDW6 and NSW6 could be interpreted in terms of different complexities of hydrometeors interactions between NSW6 and NDW6, particularly whether the CDNC and RDNC are predicted. This generates different variabilities of CDNC and RDNC between the two schemes, possibly leading to the different susceptibilities. Nevertheless, more detailed analysis will be required in future studies to explore microphysical processes responsible for these different behaviors between the two schemes. Other possible reason for the differences in the ERFaci between NDW6 and NSW6 is the different baselines of aerosol fields, as suggested by Carslaw et al. (2013) and Wilcox et al. (2015), but this is minor because the baseline of CCN at 1-km height between NDW6 and NSW6 under the PI conditions is not very different. Another possible reason is the difference in the autoconversion between NDW6 and NSW6, and the difference in ERFaci between Khairoutdinov and Kogan (2000) and Seifert and Beheng (2006) is estimated to be 0.15 $Wm^{-2}$ by using a general circulation model MIROC (Michibata and Suzuki, 2020). However, this magnitude of ERFaci difference potentially caused by the two different autoconversion schemes cannot explain the difference in ERFaci between NDW6 and NSW6 of this study. As mentioned in section 2.3, the assumption of the preindustrial conditions for aerosols can cause possible differences in the aerosol radiative forcing due to the anthropogenic sources between this study and other studies, such as IPCC-AR6 (Szopa et al., 2021). This study assumes that the anthropogenic emission fluxes of BC, OC and $SO_2$ are zero in the

preindustrial conditions, whereas other studies often use them in 1750 or 1850 provided by Hoesly et al. (2018). Using the results of MIROC by Takemura (2020), the possible difference in the aerosol radiative forcing due to the anthropogenic source will be at most 0.05 Wm$^{-2}$ (IRFari) and 0.2 Wm$^{-2}$ (ERFari plus ERFaci).

In climate simulations, the simulated radiation fluxes, i.e., SWCRF and OSR, are important. Although there are compensation errors in the regional distribution, the differences in the global and annual averages between NICAM (NDW6 and NSW6) and

CERES are estimated to be at most 3.2 Wm$^{-2}$ (SWCRF and OSR). For longwave radiation, which is not the specific focus of this study, the differences in the global and annual averages between NICAM (NDW6 and NSW6) and CERES are estimated to be at most 6.4 Wm$^{-2}$ (LWCRF) and at most 3.4 Wm$^{-2}$ (OLR). These biases in the radiation fluxes between NICAM and CERES are acceptable for a climate model, but the negative biases of SWCRF and LWCRF may cause the underestimation of aerosols and the overestimation of the net ERFaci, respectively.

In conclusion, the NICAM at a 14 km grid spacing with both the NDW6 and NSW6 cloud microphysics modules for 6 years generally successfully reproduces the observed aerosols. The NDW6-simulated RPCW, precipitation and LWP are generally closer to the satellite-retrieved results compared to the NSW6 result. The cloud microphysics representation of NDW6 is more elaborate than that of NSW6, and it found that the NSW6-simulated CDR is overestimated due to the inability to predict CDNC in NSW6. Therefore, the use of NDW6 is recommended in environmental and climate simulations. At the same time, because

the ERFaci in NDW6 needs validation, in the future it will be necessary to perform additional experiments targeting specific cases of volcano in Iceland shown in Malavelle et al. (2017) to deeply evaluate the model results in NDW6.

**Appendix A: Comparisons of the aerosol mass loading between NSW6 in this study and the results in Goto et al. (2020)**

As shown in sections 3.2 and 3.3, the evaluation is performed using statistical metrics: the Pearson correlation coefficient (PCC), normalized mean bias (NMB), and root-mean-square error (RMSE). These metrics using the concentration (C) of the

observation (*obs*) and the simulation (*sim*) and the sampling number (N) are defined as follows:

$$PCC = \frac{\sum(C_{obs} - \overline{C_{obs}})(C_{sim} - \overline{C_{sim}})}{\sqrt{\sum(C_{obs} - \overline{C_{obs}})^2 \sum(C_{sim} - \overline{C_{sim}})^2}} \quad\quad\quad (A1)$$

$$NMB = \frac{\sum(C_{sim} - C_{obs})}{\sum(C_{obs})} \times 100[\%] \quad\quad\quad (A2)$$

$$RMSE = \sqrt{\frac{\sum(C_{sim} - C_{obs})^2}{N}} \quad\quad\quad (A3)$$

For the mass loading of OM, the statistical metrics obtained in this study are greatly improved compared to the previous study

using NICAM.16 (e.g., PCC of 0.819, RMSE of 5.03, and NMB of -54.8 µg m$^{-3}$ from Goto et al., 2020), as shown in Table A1. For BC in Table A1, the statistical metrics in this study are not improved from Goto et al. (2020). For sulfate in Table A1, the values of the PCC and RMSE in this study are close to those in Goto et al. (2020), but the NMB in this study is much lower. Therefore, the model performance for surface aerosol mass concentrations in this study is apparently improved by modifying the sulfur module, as described in section 2.2.

Global and annual mean values of column burden, emission, and atmospheric lifetime are also compared to the results of Goto et al. (2020), as shown in Table A2. The difference in the dust emission and its column burden among different cloud microphysics modules, i.e., between NDW6 and NSW6, is larger than that among different horizontal resolutions, i.e., between the HRM (14 km) and LRM (56 km). The dust lifetime in this study is shorter than that in Goto et al. (2020). The difference is probably caused by the difference in the dust emission scheme, as described in section 2.2. In support of this, the global

climate model MIROC uses the same dust emission scheme as in this study and has a shorter lifetime among the AeroCom models (Gliß et al., 2021). The decrease in the number of bins from 10 to 6 may reduce the variability of the particle size distribution, move the center of the particle size distribution to coarser sizes, lead to an increase in the amount of deposition, and then reduce the lifetime. For sea salt, the difference in the emissions between NSW6 in this study and the HRM in Goto et al. (2020) is approximately 11%. This causes the difference in the column burden, although the difference in the lifetime

between this study and Goto et al. (2020) is small. For OM and BC, the differences in NSW6 and HRM are small and within the AeroCom models (Gliß et al., 2021).

For sulfate, the column burden is 0.52 TgS (NSW6), 0.38 TgS (HRM) and 0.32 TgS (LRM), which are within the uncertainty among AeroCom models (Gliß et al., 2021). The lifetimes of sulfate are 3.3 days (NSW6), 2.4 days (HRM) and 2.1 days (LRM), which are also within the variability among the AeroCom models (Gliß et al., 2021). The difference in the sulfate

column burden between NSW6 and the HRM is larger than the difference among cloud microphysics modules and different horizontal resolutions. This is attributed to a change in the assumption that sulfate forms in clouds by aqueous-phase oxidation, as described in section 2.2. The difference in the column burden of the simulated sulfate among cloud microphysics modules is comparable to that among different horizontal resolutions. For $SO_2$, the column burden of the simulated $SO_2$ is 0.33 TgS (NSW6), 0.33 TgS (HRM), and 0.31 TgS (LRM). Therefore, the difference in the horizontal distribution affects $SO_2$ oxidation

and then the column burden of sulfate, whereas the difference in the cloud microphysics module does not affect the chemical budget of $SO_2$ oxidation.

**Appendix B: Comparisons in the AOT between NSW6 in this study and the results in Goto et al. (2020)**

The scatterplots of the simulated AOT and the ground-based observed AOT are shown in Figure 7 in this study and Figure 6 in Goto et al. (2020). The statistical metrics in this study (both NDW6 and NSW6) are much better than those reported in Goto

et al. (2020). The PCC value improved from 0.471 (HRM) to 0.837 (NSW6), the RMSE value improved from 0.21 (HRM) to 0.12 (NSW6), and the NMB value improved from -20.2% to -5.4%. In the horizontal distribution of the AOT shown in Figure 8, especially over the Southern Ocean, the NSW6-simulated AOT is greatly improved from the results of Figure 5 in Goto et al. (2020). This means that an AOT larger than 0.3 was frequently observed in past NICAM simulations. Over heavy AOT areas, for example, in the Sichuan Basin located in Southwest China and the Indo-Gangetic Plain, the NSW6-simulated AOT

is generally more comparable to the satellite-retrieved AOTs compared to the results in the HRM in Goto et al. (2020). The improvement in these areas is caused by the increased number of vertical layers below 2 km height from 10 to 15 in this study, as described in section 2.1. The higher vertical resolution causes better performance of the simulated aerosols in the boundary

layer by suppressing the artificial dispersion and diffusion of the aerosols near the surface on the basin. This change can be found in not only the AOT but also the column burden and the surface mass concentrations of the aerosols.

At the end of section 3.3, the vertical profiles of the simulated aerosol extinction coefficients are evaluated by the CALIOP retrieval results (Figure 9). Due to the increase in the vertical layer from Goto et al. (2020) to the present study, any improvements in the vertical profiles of the simulated aerosols between NSW6 and the HRM are expected to be large, but drastic changes are not found in most regions. Large differences in the vertical profile are found in East China (Figure 9e), the coast of East Asia (Figure 9f), and Southeast Asia (Figure 9i). Along the coast of East Asia, for example, the decay height of

the CALIOP-retrieved extinction coefficients is approximately 0.5 km, whereas the decay height of the NSW6-simulated extinction coefficients is approximately 1 km and that of the HRM-simulated extinction coefficients is zero. In this area, the NSW6-simulated results are improved over the HRM-simulated results. In dusty regions such as the coast of northern Africa (Figure 9g) and northern Africa (Figure 10h), the differences in the aerosol profiles between NSW6 and the HRM are small, even though the dust schemes used in NSW6 are different from those in the HRM, as described in section 2.2. Along the coast

of central Africa and over South America, both NSW6- and HRM-simulated results still include biases in the vertical profiles. As mentioned in section 3.3, this bias indicates a problem of the vertical transport of aerosols originating from biomass burning in the NICAM, which may not be solved by the finer vertical and horizontal resolutions.

**Appendix C: Comparisons of the AOT components between the NICAM and AeroCom models**

The simulated AOT compositions are compared with the references of the AeroCom models (Gliß et al., 2021) in Figure C1.

All the NICAM-simulated dust AOTs are larger than those obtained from the AeroCom models. Since the column loadings of both the NDW6- and NSW6-simulated dust ranges are within the uncertainty among the AeroCom models shown in Figure 5, the dust treatment updates, especially the reduction in the number of bins for dust, may result in a higher mass extinction coefficient of the dust. The uncertainty of the dust AOTs among the AeroCom models is lower than that of other AOTs, probably because the AeroCom models in dusty areas must be tuned by modifying dust emissions to become closer to the

satellite results. All the NICAM-simulated sea salt AOTs range within the uncertainty of the AeroCom models, but the mass extinction coefficient of the sea salt tends to be higher than that of the AeroCom models since the column burden of the NICAM-simulated sea salt is lower than that of the AeroCom models. For carbon and sulfate, all the NICAM-simulated AOTs range within the uncertainty of the AeroCom models. For total species, the NICAM-simulated AOTs, except under NDW6 and clear-sky conditions, are larger than the upper range among the AeroCom models. The NDW6-simulated AOTs and the

NSW6-simulated AOTs under only clear-sky conditions are close to the median values of the AeroCom models but lower than the averages obtained from the ground-based AERONET measurements and satellites in Figure 3 of Gliß et al. (2021). The NDW6- and NSW6-simulated absorption AOTs also range within the uncertainty of the AeroCom models (Sand et al., 2021) but are lower than the median value of the AeroCom models.

**Appendix D: Comparisons of the shortwave IRFari between NSW6 in this study and the results in Goto et al. (2020)**

As shown in Figure 10, the NSW6-estimated IRFari values are compared with the HRM-estimated values and references. Generally, the differences in the IRFari values between NSW6 and HRM are similar to those in the column burden. After the improvement of increased vertical levels and updated aerosol modules, the IRFari values at TOA are changed by +0.27 Wm$^{-2}$ (dust), -0.13 Wm$^{-2}$ (seasalt), -0.01 Wm$^{-2}$ (WSBC+POM, i.e., BC-containing particles), +0.04 Wm$^{-2}$ (SOA, i.e., pure OM), +0.03 Wm$^{-2}$ (WIBC, i.e., pure BC), -0.14 Wm$^{-2}$ (sulfate), -0.11 Wm$^{-2}$ (only anthropogenic aerosols), and +0.08 Wm$^{-2}$ (all

aerosols). At the surface, the IRFari values are changed by +0.45 Wm$^{-2}$ (dust), -0.14 Wm$^{-2}$ (seasalt), -0.08 Wm$^{-2}$ (BC-containing particles), +0.03 Wm$^{-2}$ (SOA), -0.04 Wm$^{-2}$ (pure BC), -0.12 Wm$^{-2}$ (sulfate), +0.08 Wm$^{-2}$ (only anthropogenic aerosols), and -0.25 Wm$^{-2}$ (all aerosols). The differences in the IRFari magnitude between NSW6 and HRM are generally higher than those among the different cloud microphysics modules (NDW6 and NSW6) and the different horizontal resolutions (HRM and LRM).

For IRFari, due to changes in aerosols from the preindustrial era and the present era, the differences in NSW6 and HRM is +0.03 Wm$^{-2}$ (all-sky) and -0.03 Wm$^{-2}$ (clear-sky), respectively (see Figure 11). Under all-sky conditions, the difference in the anthropogenic IRFari values between NSW6 and HRM is the largest, whereas under clear-sky conditions, the difference in the values between NDW6 and NSW6 is the largest among these differences. This result is mainly caused by the increase in the simulated sulfate in the present study, as shown in Appendix A.

**Appendix E: Evaluation of total radiative fluxes in NICAM**

Figure E1 illustrates the horizontal distribution of the NDW6- and NSW6-simulated and CERES-retrieved SWCRF as annual, January, and July averages. Over the Southern Ocean in January and the North Pacific and Atlantic Oceans in July, the magnitudes of the SWCRF values are larger than those in other areas and seasons. In these areas, the NDW6-simulated SWCRF values over the Southern Ocean are closer to the CERES results, whereas the NSW6 results are lower than the CERES results.

These results are very consistent with the LWP results, as shown in Figure 2. Over southern China, the eastern US, Europe, central Africa, and the coast of Mexico in January and central Africa, the central Pacific Ocean, and central Asia in July, the NDW6-simulated SWCRF values are closer to the CERES results, whereas in Australia in January and in South Asia in July, the NSW6-simulated SWCRF values are closer to the CERES results. Conversely, in these areas, the magnitudes of the NDW6-simulated SWCRF values are underestimated compared to the CERES results, which is caused by the underestimation of the

NDW6-simulated clouds. Neither the NDW6- nor the NSW6-simulated SWCRF values are generally comparable to the CERES results in coastal central Africa in January, the outflow in northern Africa in January, the central Pacific Ocean in January, and the Arctic in July. In zonal averages, these biases that are found in various areas may be effectively cancelled. The NDW6-simulated SWCRF values are closer to the CERES-retrieved results, especially for most of the Northern Hemisphere and at 60-90°S latitudes in January and at 30-90°N latitudes and most of the Southern Hemisphere in July. In

contrast, the NSW6-simulated SWCRF values are closer to the CERES results, especially at 45°S-10°N latitudes in January and at 0-45°N latitudes in July.

Figure E2 shows the spatiotemporal distribution of the simulated OSR. As shown in Figure E1, the good performance of the NDW6-simulated clouds and SWCRF generally produces closer results to those of CERES in the Southern Ocean, Northern Pacific, and Atlantic Ocean in January. Especially over the ocean, the larger magnitude of the SWCRF yields a larger

magnitude of the OSR. Therefore, the biases of the simulated SWCRF directly reflect the results of the simulated OSR. This is also indicated by the comparisons of the zonal distribution of both SWCRF and OSR. Over land, however, due to limited clouds and large aerosols, radiative impacts due to aerosols can be found. In January, at 30-45°N latitudes, including industrial areas such as eastern China and dusty areas such as central Asia and western China, the NDW6-simulated SWCRF values are comparable to the CERES results, but the NDW6-simulated OSR values are larger than the CERES results. This suggests the

overestimation of scattering aerosols, underestimation of light-absorbing aerosols, or overestimation of the surface albedo in dusty areas. Figure 8 does not suggest overestimation of simulated aerosols compared to the satellite results. Figure C1 shows an underestimation tendency for absorption AOT among the AeroCom models. Therefore, overestimation of the simulated OSR may be caused by the underestimation of simulated light-absorbing aerosols. In July, at 60-90°N latitudes, even though the magnitudes of the NDW6-simulated SWCRF are lower than those of the CERES results, the NDW6-simulated OSR values

are comparable to the CERES results. This may imply the underestimation of the simulated scattering aerosols, which is consistent with the results over the Arctic shown in Figure 12 of Goto et al. (2020), even though the seasonal variation in the simulated aerosols is comparable to the ground-based observations. Other possibilities include the overestimation of light-absorbing aerosols over the Arctic and/or this effect on the decrease in water vapor, but this possibility is inconsistent with the results of Figure C1. Globally, the NDW6-simulated OSR is more comparable to the CERES result than the NSW6 result in

terms of annual, January, and July averages, as shown in Table 1. The global and annual averages are calculated to be 98.6 Wm$^{-2}$ (NDW6), 102.0 Wm$^{-2}$ (NSW6), and 99.0 Wm$^{-2}$ (CERES).

For longwave radiation, the impacts of ARI are relatively small. Because the NICAM does not explicitly address the interaction between aerosols and ice clouds, the impacts of the ACI on longwave radiation are also small. Here, the results of OLR and LWCRF are briefly discussed using the global averages shown in Table 1. The global and annual averages of LWCRF are

calculated to be 21.5 Wm$^{-2}$ (NDW6), 26.8 Wm$^{-2}$ (NSW6), and 27.9 Wm$^{-2}$ (CERES). The NSW6-simulated LWCRF appears closer to the CERES results, especially at 30°S-30°N latitudes. When the horizontal distribution of the LWCRF is examined, compensation errors in terms of longitudes are found. In the western Pacific Ocean, the NSW6-simulated clouds at high levels are remarkably overestimated compared to the CERES results, but in other areas, both the NDW6- and NSW6-simulated LWCRF are underestimated compared to the CERES results. However, because the NDW6-simulated clouds at the high level are not as remarkably overestimated compared to the CERES results, the global average of the NDW6-simulated LWCRF is

are not as remarkably overestimated compared to the CERES results, the global average of the NDW6-simulated LWCRF is lower than the CERES results. In the OLR shown in Table 1, the global and annual averages are calculated to be 242.2 Wm$^{-2}$ (NDW6), 236.8 Wm$^{-2}$ (NSW6), and 240.2 Wm$^{-2}$ (CERES). In January and July, the NDW6-simulated OLR appears close to the CERES results, whereas the NSW6-simulated OLR is lower than both the CERES and NDW6 results. As already

mentioned, compensation errors in the regional distribution arise, but such errors cannot be solved by improvements to the aerosols.

## Code and data availability

The source codes of NICAM.19 used in this study can be obtained at DOI:10.5281/zenodo.7731449 upon request under general terms and conditions (http://nicam.jp/hiki/?Research+Collaborations). The relevant model results in this study are archived at DOI:10.5281/zenodo.7731486.

## Author contributions

DG designed and conducted the numerical experiments and analyses. TS and HY configured the model and prepared the external conditions of the experiments. DG wrote the initial draft of the paper, and all the coauthors (TS, KS, HY, and TT) participated in the discussions of the results and commented on the original manuscript.

## Competing interests

The authors declare that they have no conflicts of interest.

## Acknowledgments

We acknowledge the developers and administrators of the NICAM (http://nicam.jp/; last access: 30 August 2023), MODIS (https://modis.gsfc.nasa.gov/; last access: 30 August 2023), MISR (https://misr.jpl.nasa.gov/; last access 30 August 2023), CALIOP (https://www-calipso.larc.nasa.gov/; last access: 30 August 2023), GPGP (https://www.ncei.noaa.gov/access/metadata/landing-page/bin/iso?id=gov.noaa.ncdc:C00933; last access 30 August 2023), MACLWP (https://disc.gsfc.nasa.gov/datasets/MACLWP_diurnal_1/summary?keywords=measures; last access 30 August 2023), ISCCP (https://isccp.giss.nasa.gov/; last access 30 August 2023), 2C-RAIN-PROFILE (https://www.cloudsat.cira.colostate.edu/data-products/2c-rain-profile; last access 30 August 2023), and the relevant PIs of the IMPROVE (http://vista.cira.colostate.edu/Improve/; last access: 30 August 2023), EMEP (https://www.emep.int/; last access: 30 August 2023), EANET (https://www.eanet.asia/; last access: 30 August 2023), AERONET (https://aeronet.gsfc.nasa.gov/; last access: 30 August 2023), SKYNET (https://www.skynet-isdc.org/; last access: 30 August 2023) and CARSNET sites. The CERES datasets were obtained from the NASA LaRC Atmospheric Science Data Center (https://asdc.larc.nasa.gov/; last access: 30 August 2023). Some of the authors were supported by the Environment Research and Technology Development Fund S-20 (S-20-1(1): JPMEERF21S12001, S-20-1(3): JPMEERF21S12003, and S-20-1(4): JPMEERF21S12004) of the Environmental Restoration and Conservation Agency that were provided by the Ministry of Environment (MOE) of Japan and by the Japan Society for the Promotion of Science (JSPS) KAKENHI grants (17H04711 and 19H05669) and the Ministry of

Education, Culture, Sports, Science, and Technology (MEXT) (JPMXP1020200305) via the "Program for Promoting Research on the Supercomputer Fugaku" (Large Ensemble Atmospheric and Environmental Prediction for Disaster Prevention and Mitigation). Additionally, we were supported by the Advanced Studies of Climate Change Projection (SENTAN) of MEXT (JPMXD0722680395), MOE/GOSAT, Japan Science and Technology (JST), Japan Aerospace Exploration Agency (JAXA)/EarthCARE, JAXA/GCOM-C, and National Institute for Environmental Studies (NIES), Japan. The model simulations were performed using the following supercomputers: the RIKEN/Fugaku computer (hp210166, hp210253, hp220167, and hp220213), Flow at Information Technology Center, Nagoya University, and NEC SX-Aurora TSUBASA at NIES. Global maps in the figures are drawn using the Grid Analysis and Display System (GrADS) (http://cola.gmu.edu./grads/; last access: 30 August 2023). We acknowledge Takuro Michibata (Okayama University, Japan) for preparing the Cloudsat datasets in Figure 6 and discussing the cloud precipitation in NICAM, Eiji Oikawa (Meteorological Research Institute, Japan) for preparing the CALIPSO dataset used in Goto et al. (2020), Takashi M. Nagao (University of Tokyo, Japan) for preparing the MODIS cloud dataset used in Goto et al. (2020), and Haruka Hotta (University of Tokyo, Japan) for helping set up the NICAM to run with NDW6.

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

**Table 1: Annual, January and July mean values of clouds, precipitation, and radiation**

| | Precipitation [mm day$^{-1}$] | LWP[*1] [g m$^{-2}$] | COT[*2] | Low-level CF | CDR[*3] [μm] | OSR [W m$^{-2}$] | SWCRF [W m$^{-2}$] | OLR [W m$^{-2}$] | LWCRF [W m$^{-2}$] |
|---|---|---|---|---|---|---|---|---|---|
| | Annual mean | | | | | | | | |
| NDW6 | 3.01 | 95.8 | 7.1 | 0.19 | 9.8 | 98.6 | -42.5 | 242.2 | 21.5 |
| NSW6 | 2.78 | 104.4 | 8.3 | 0.19 | 8.0 | 102.0 | -45.9 | 236.8 | 26.8 |
| Observation[*4] | 2.68 | 119.6 | 12.9 | 0.26 | 13.8 | 99.0 | -45.7 | 240.2 | 27.9 |
| | January | | | | | | | | |
| NDW6 | 2.99 | 98.0 | 7.2 | 0.18 | 9.8 | 105.9 | -48.4 | 238.5 | 22.0 |
| NSW6 | 2.79 | 99.1 | 8.1 | 0.18 | 8.1 | 106.7 | -49.3 | 233.4 | 26.7 |
| Observation[*4] | 2.73 | 120.2 | 14.5 | 0.25 | 13.2 | 105.9 | -50.4 | 237.6 | 27.6 |
| | July | | | | | | | | |
| NDW6 | 3.09 | 100.7 | 7.3 | 0.21 | 9.6 | 94.4 | -41.8 | 245.5 | 21.7 |
| NSW6 | 2.85 | 119.0 | 9.3 | 0.21 | 8.1 | 101.6 | -48.7 | 240.4 | 26.8 |
| Observation[*4] | 2.71 | 121.8 | 13.2 | 0.27 | 14.3 | 94.1 | -44.5 | 244.0 | 27.7 |

[*1] LWP over oceans (60°S-60°N); [*2] COT (60°S-60°N); [*3] CDR at warm-topped clouds (60°S-60°N); [*4] GPCP (precipitation), MAC (LWP), MODIS (COT), ISCCP (low-level CF), MODIS (CDR), and CERES (OSR, SWCRF, OLR, and LWCRF)

**Table 2: Global and annual mean values of ERFari for anthropogenic aerosols, ERFaci for anthropogenic aerosol-, and the net ERF (sum of ERFari and ERFaci) for shortwave, longwave, and net (sum of shortwave and longwave) radiation under the all-sky and clear-sky conditions. All units are in W m$^{-2}$.**

| | ERFari under the all-sky conditions | | | |
|---|---|---|---|---|
| | NDW6 | NSW6 | HRM | LRM |
| Shortwave | -0.22 | -0.26 | -0.33 | -0.26 |
| Longwave | 0.03 | 0.03 | 0.04 | 0.02 |
| Net | -0.19 | -0.23 | -0.29 | -0.24 |
| | ERFari under the clear-sky conditions | | | |
| Shortwave | -0.52 | -0.60 | -0.63 | -0.51 |
| Longwave | 0.05 | 0.05 | 0.05 | 0.03 |
| Net | -0.47 | -0.55 | -0.57 | -0.48 |
| | ERFaci | | | |
| Shortwave | -1.34 | -0.63 | -0.81 | -1.17 |
| Longwave | 0.06 | -0.10 | -0.12 | 0.07 |
| Net | -1.28 | -0.73 | -0.93 | -1.10 |
| | ERFari+ERFaci | | | |
| Shortwave | -1.56 | -0.89 | -1.15 | -1.43 |
| Longwave | 0.09 | -0.07 | -0.08 | 0.09 |
| Net | -1.47 | -0.96 | -1.23 | -1.34 |

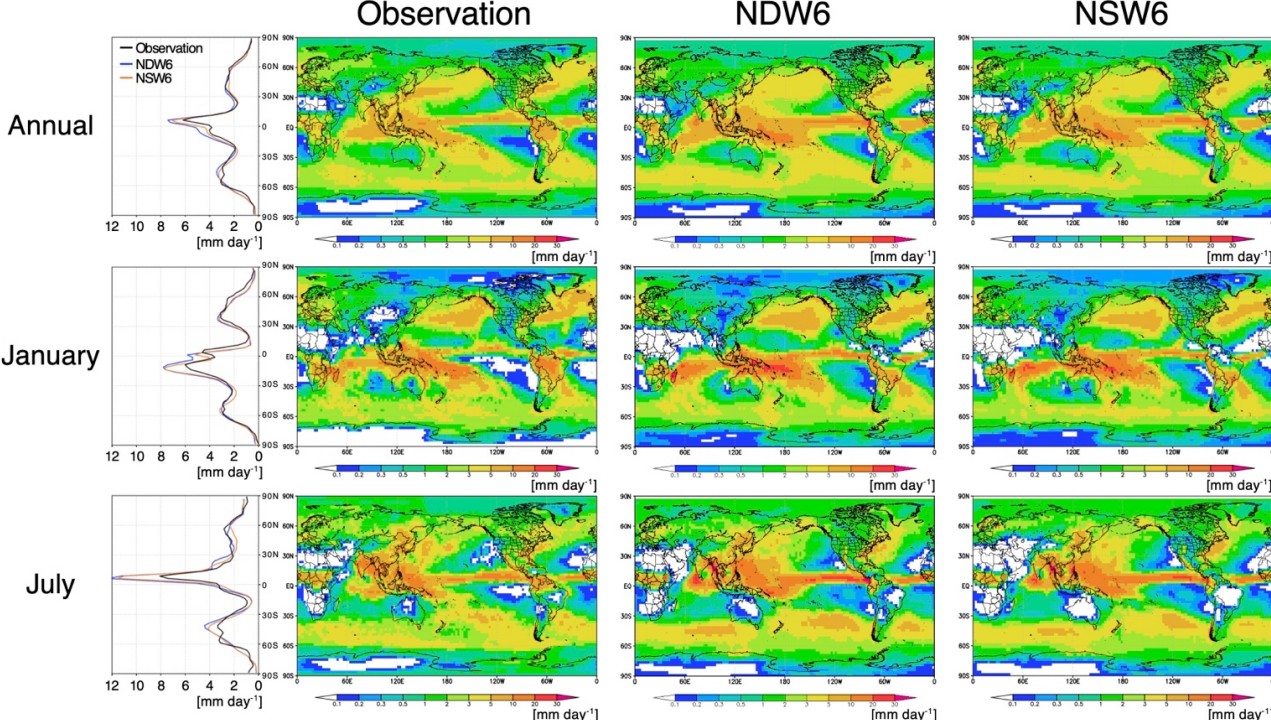

**Figure 1: Zonal and horizontal distribution of precipitation (NDW6 and NSW6 simulations and GPCP as observations) as annual, January, and July averages. All units are in mm day⁻¹.**

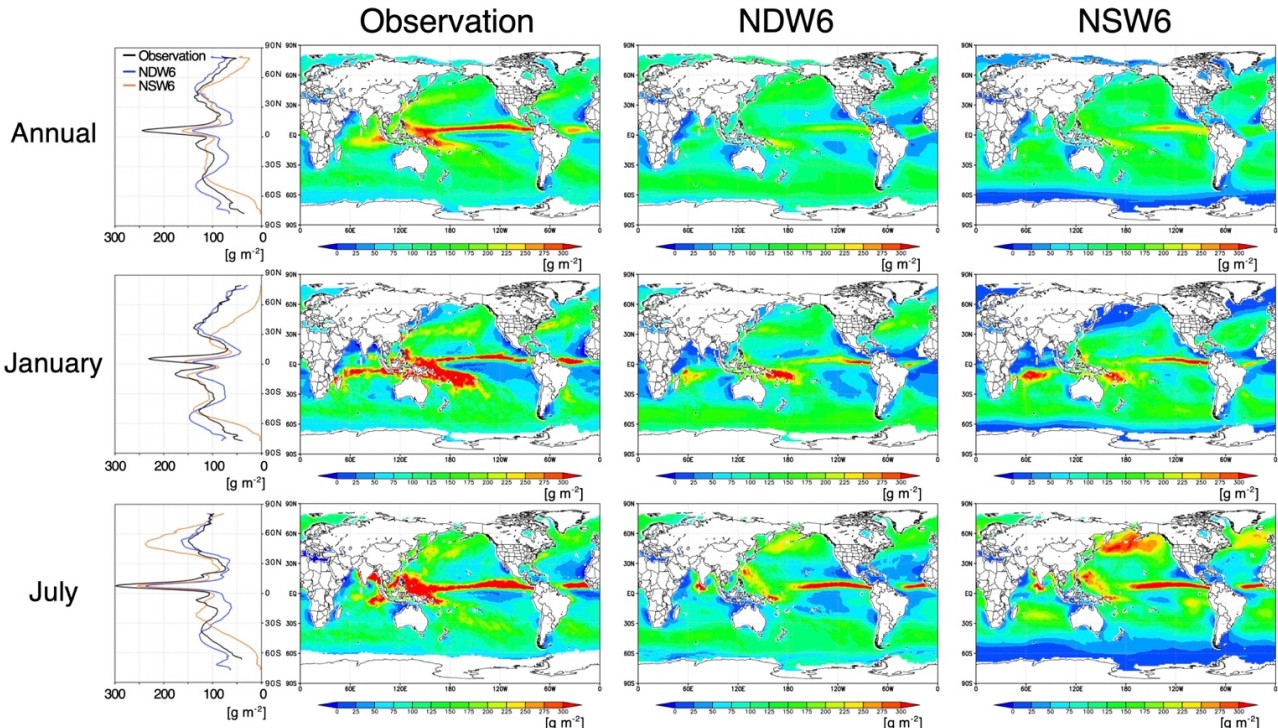

**Figure 2: Zonal and horizontal distributions of LWP (NDW6 and NSW6 simulations and MAC as observations) over only the ocean as annual, January, and July averages. All units are in g m⁻².**

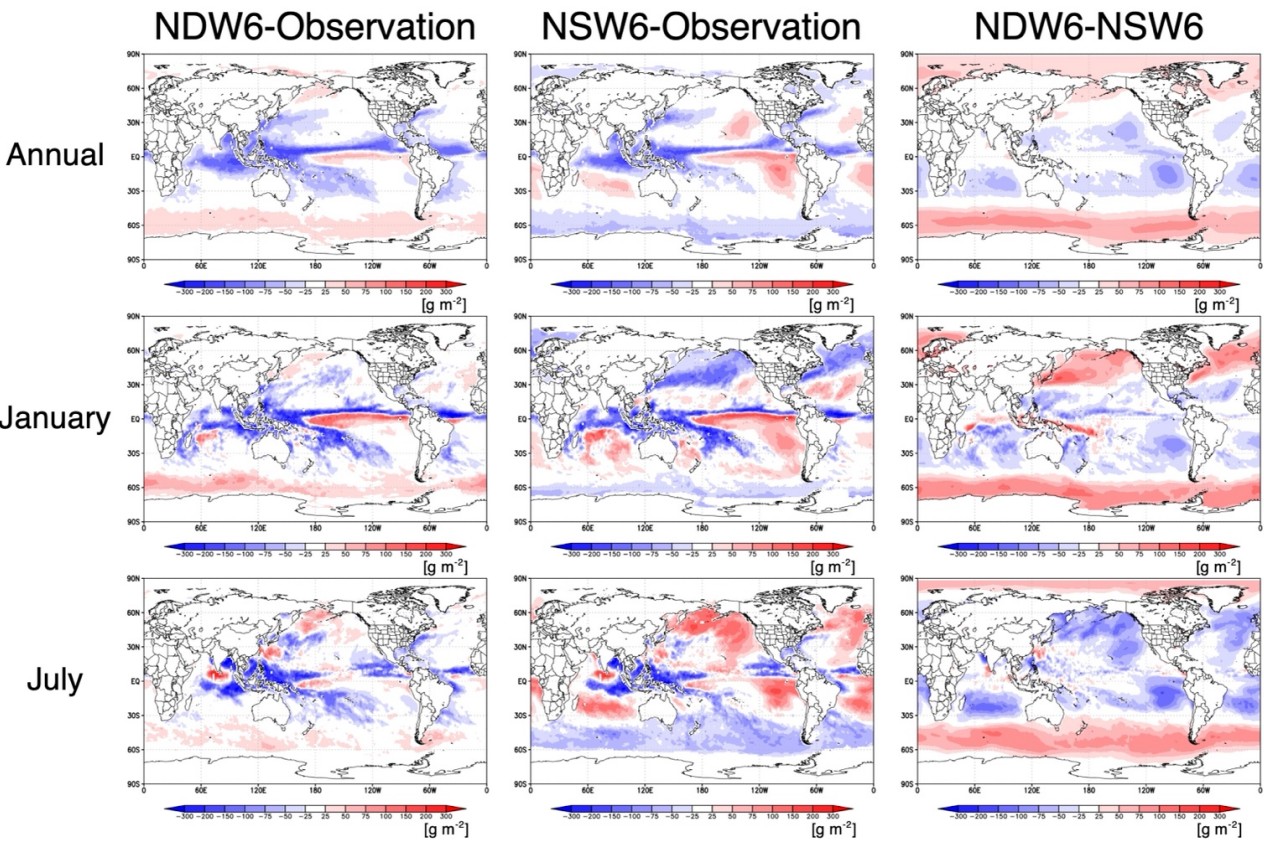

**Figure 3: Horizontal distributions of differences in LWP among NDW6, NSW6 and Observation (MAC) over only the ocean as annual, January, and July averages. All units are in g m⁻².**


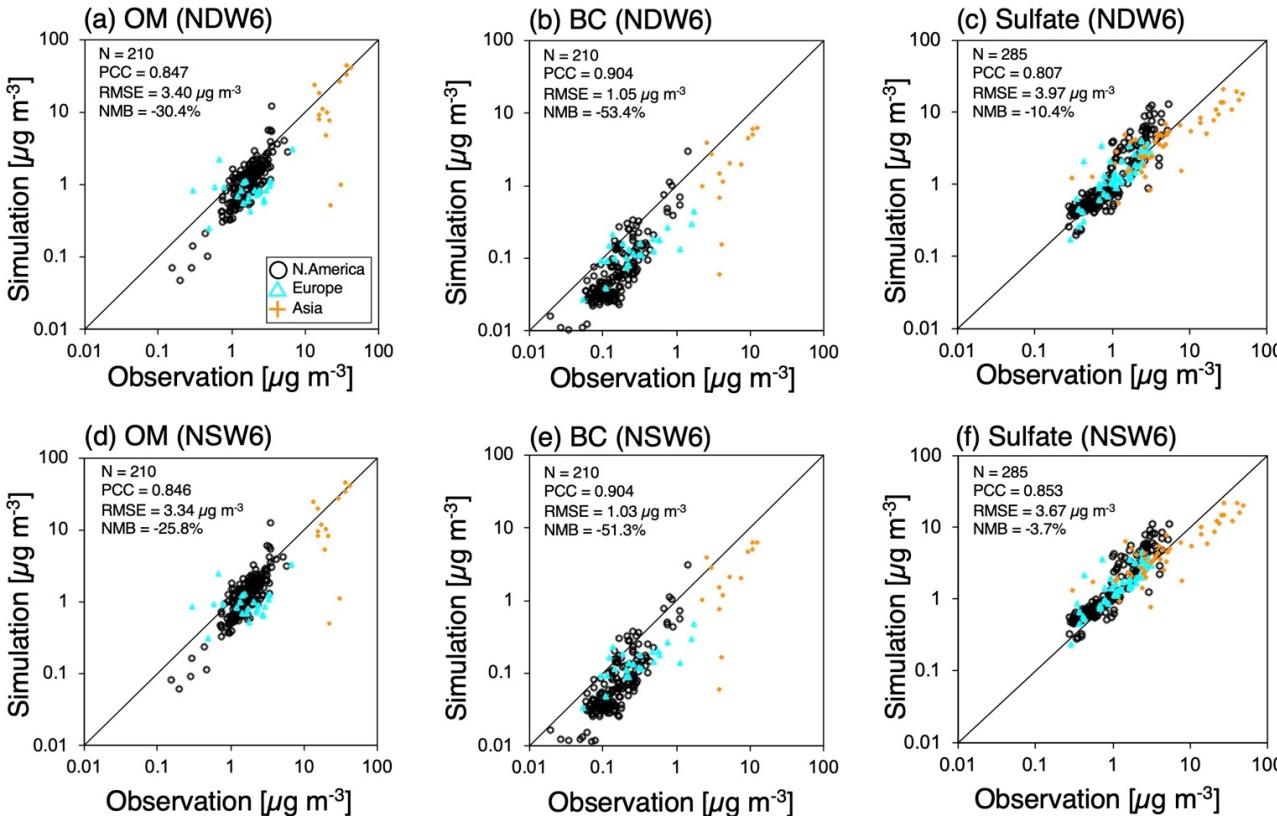

**Figure 4: Scatterplots of the annual averages of surface aerosol mass concentrations (OM, BC, and sulfate) between in situ measurements (IMPROVE, EMEP, EANET and CAWNET) and the NICAM simulations (NDW6 and NSW6). All units are in μg m⁻³. The statistical metrics (N: sampling number, PCC: Pearson correlation coefficient, RMSE: root-mean-square error, and NMB: normalized mean bias), defined as (A1)-(A3) in Appendix A, are also shown in each panel. The values are also listed in Table A1.**

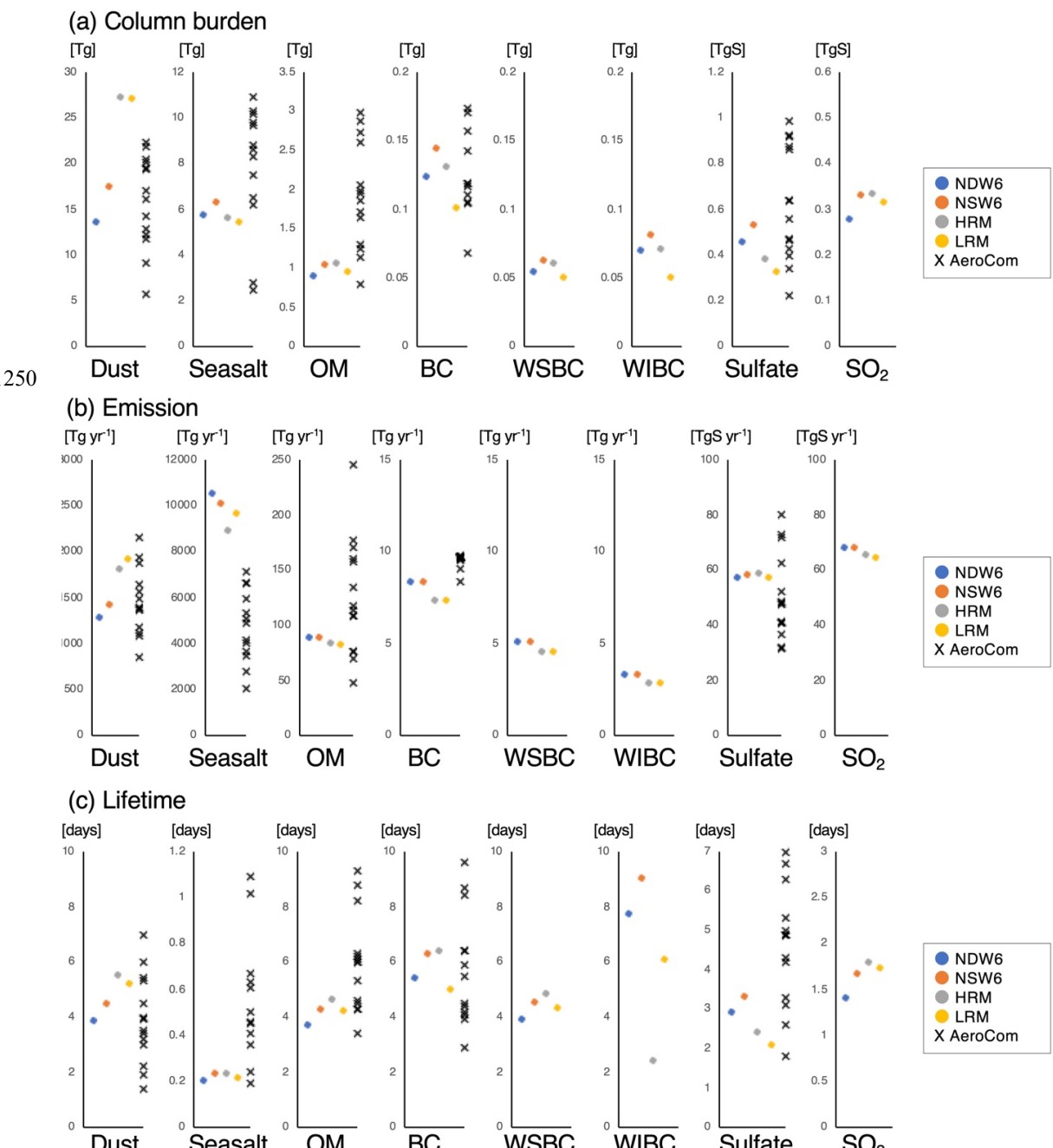

**Figure 5: Global and annual mean values of (a) column burdens [Tg or TgS], (b) emission fluxes [Tg yr⁻¹ or TgS yr⁻¹], and (c) atmospheric lifetimes of the simulated aerosols and SO₂ [days]. The results include NDW6 and NSW6 in this study and references for the HRM and LRM in Goto et al. (2020) and AeroCom (Gliß et al., 2021). The values are also listed in Table A2.**



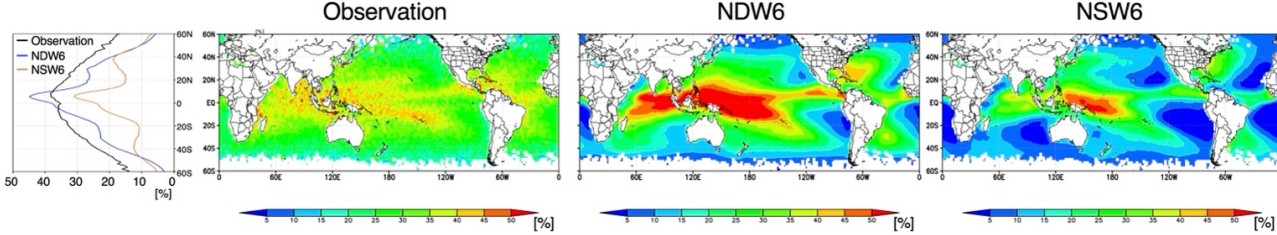

**Figure 6: (a) NDW6-simulated, (b) NSW6-simulated, and (c) Cloudsat-retrieved ratio of column precipitation to the sum of column precipitation and total cloud water (RPCW) above 1 km height as annual averages. All units are in %.**


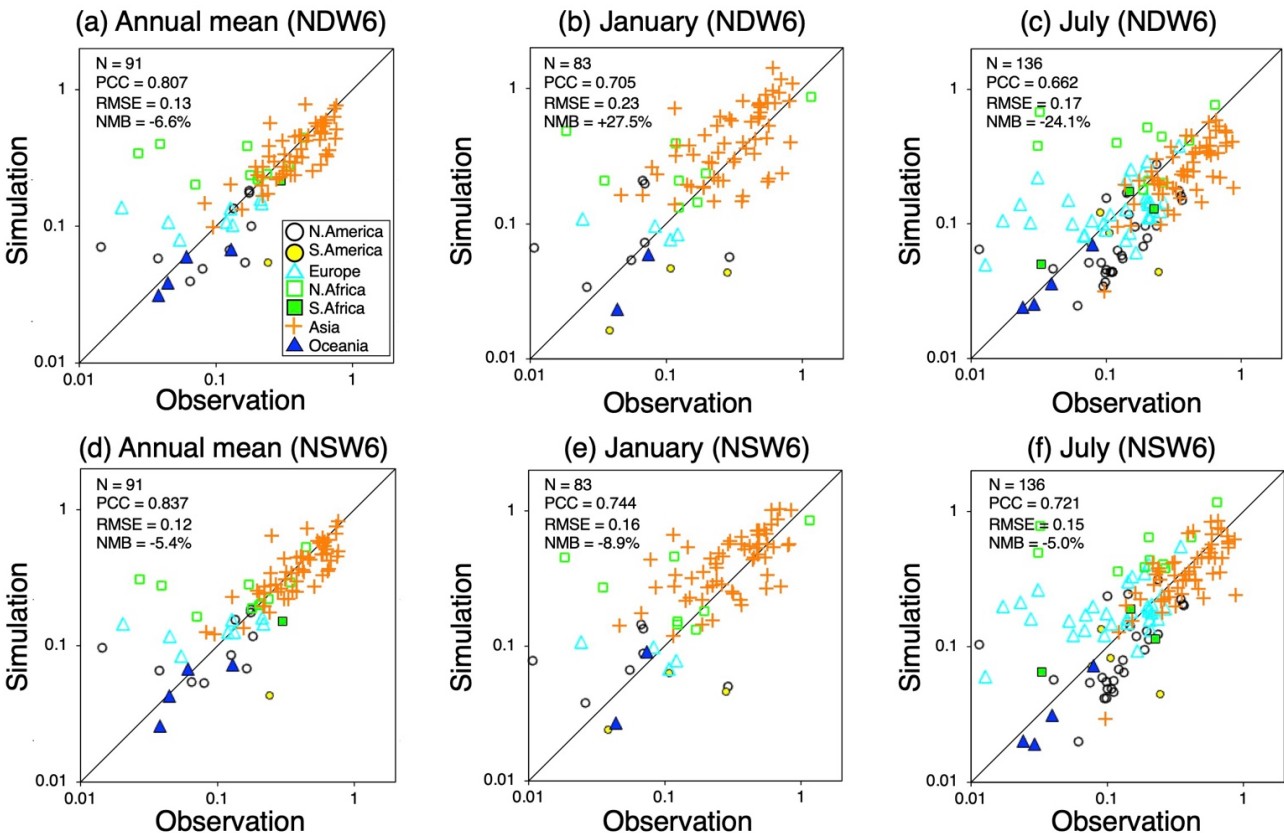

**Figure 7: Scatterplot of the annual, January and July averages of AOT between ground-based measurements (AERONET, SKYNET and CARSNET) and the NICAM (NDW6 and NSW6) simulations. The different colors and symbols reflect the sites in the different regions (North America, South America, Europe, North Africa, South Africa, Asia, and Oceania) as defined in panel (a). The numbers located in the upper-left corner in each panel represent the statistical metrics, N, PCC, RMSE and NMB, which are defined as (A1)-(A3) in Appendix A.**

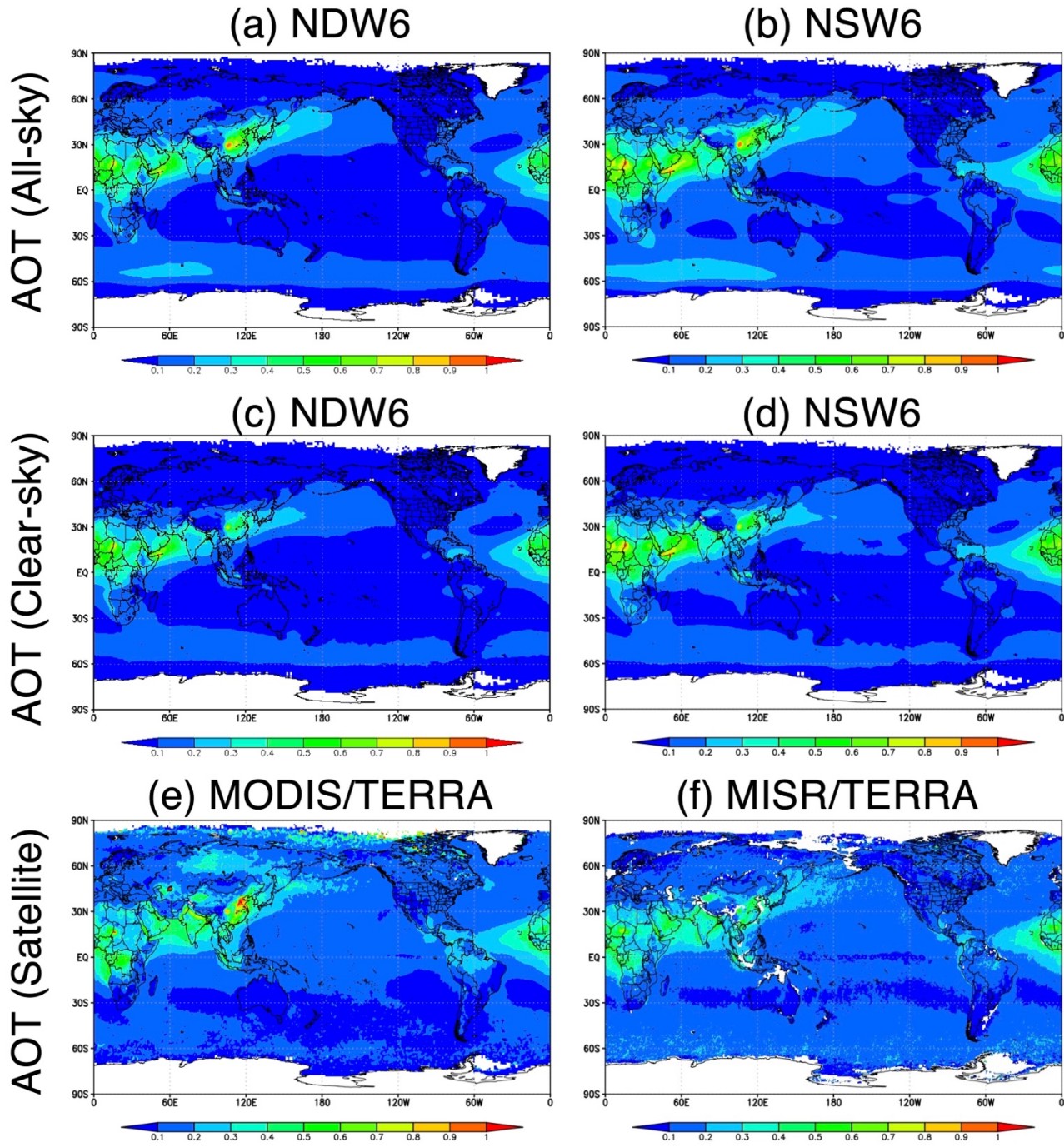

l270

**Figure 8: Global distributions of the annual averages of the NDW6-simulated AOT under (a) all-sky and (c) clear-sky conditions, the NSW6-simulated AOT under (b) all-sky and (d) clear-sky conditions, and (e) the MODIS/TERRA-retrieved and (f) the MISR/TERRA-retrieved AOT under clear-sky conditions.**

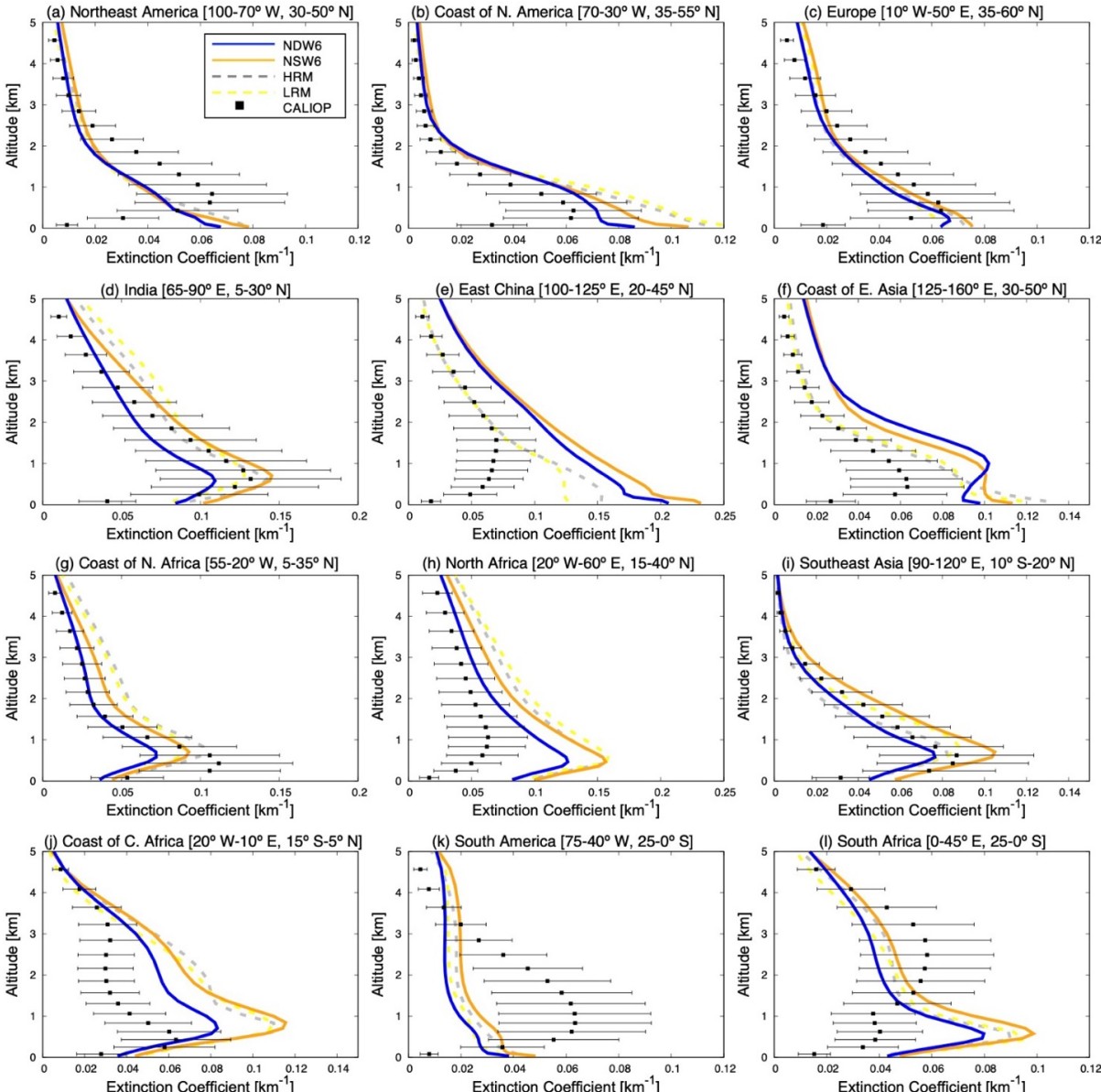

**Figure 9: Vertical profiles of the annually and regionally averaged aerosol extinction coefficients from the NICAM simulations (NDW6, NSW6, HRM and LRM) and from CALIOP/CALIPSO observations in 12 different regions, which are generally defined in Goto et al. (2020) and Koffi et al. (2016). The CALIOP-retrieved values include the standard deviation of the results from 2014-2016 as bars. All units are in km⁻¹.**

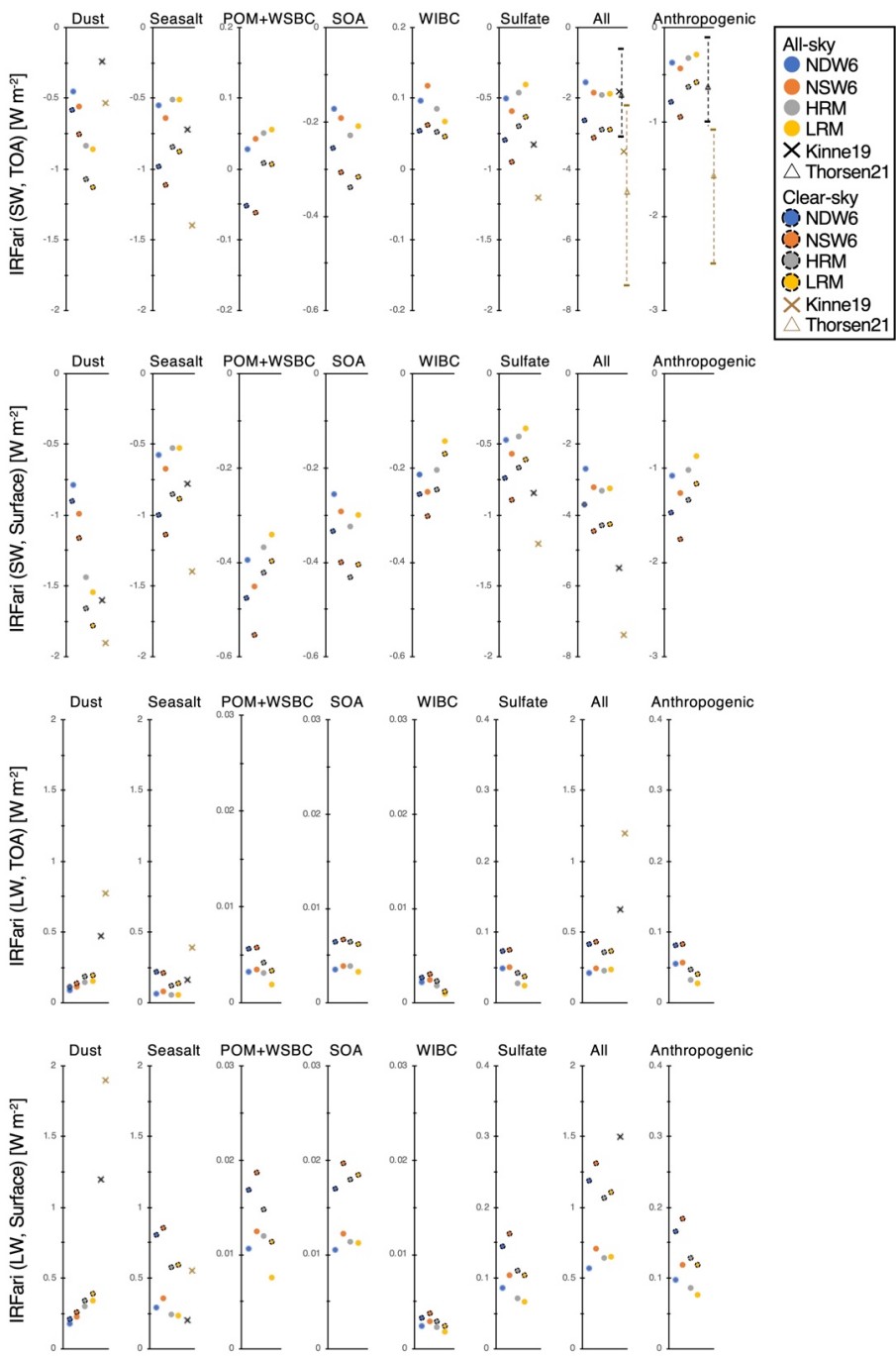

**Figure 10: Instantaneous radiative forcing due to aerosol-radiation interaction (IRFari) for each aerosol (dust, sea salt, POM+WSBC, SOA, WIBC, and sulfate), total aerosols (all), and anthropogenic aerosols only (anthropogenic) for shortwave and longwave radiation at the TOA and the surface. The references are Thorsen et al. (2021) and Kinne (2019). All units are in W m⁻².**

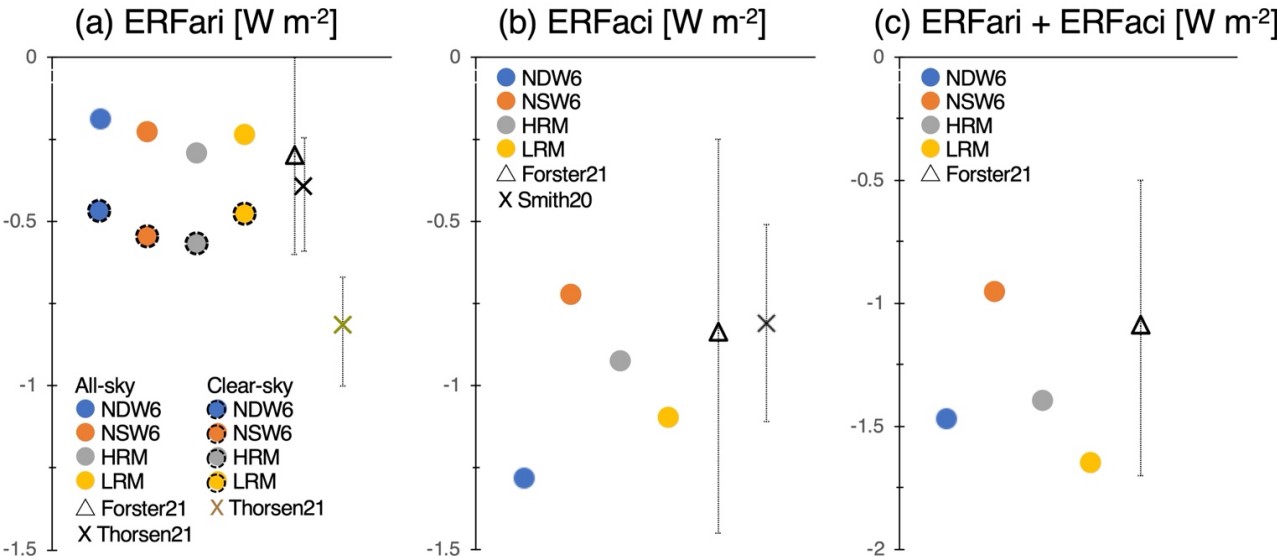

**Figure 11: Global and annual mean values of (a) effective radiative forcing for anthropogenic aerosol-radiation interaction (ERFari) for net (sum of shortwave and longwave) radiation, (b) ERFaci for anthropogenic aerosol-cloud interaction, and (c) the net ERF (sum of ERFari and ERFaci). All units are in W m⁻². In ERFari, the reference of Forster21 is estimated in the net radiation by IPCC-AR6 or Forster et al. (2021), whereas the reference of Thortsen21 is estimated in the shortwave radiation by Thorsen et al. (2021). The reference for Smith20 is Smith et al. (2020). The values are also listed in Table 2.**

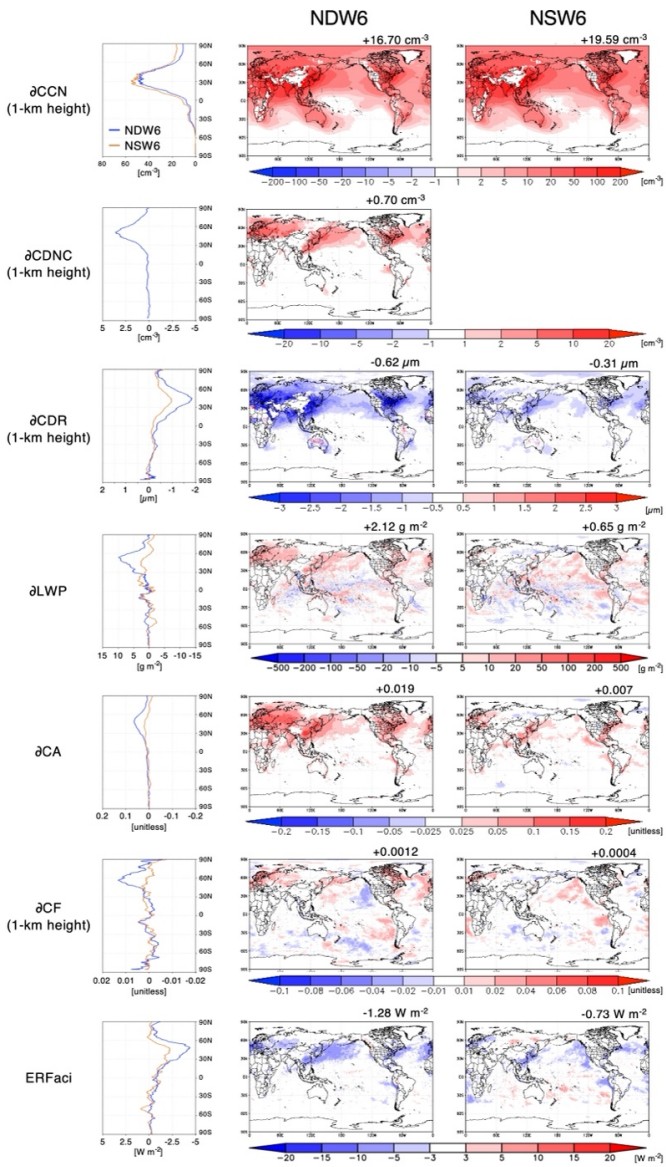

**Figure 12:** Global distributions of the annual averages of the NDW6- and NSW6-simulated CCN change at 1-km height ($\partial$CCN), CDNC (which in NSW6 is equal to the CCN concentrations in NSW6 due to the ignorance of sink processes in the CDNC in NSW6) change at 1-km height ($\partial$CDNC), CDR change for warm clouds at 1-km height ($\partial$CDR), LWP change ($\partial$LWP), CA change ($\partial$CA), CF change at 1-km height ($\partial$CF), and net ERFaci by comparing the results between NDW6 and NSW6 for simulations with aerosol and precursor gas emissions for the present and the preindustrial era. The number located in the upper right in each panel represents the global and annual mean value. The results at 1-km height also include areas with elevations higher than 1-km height in white.

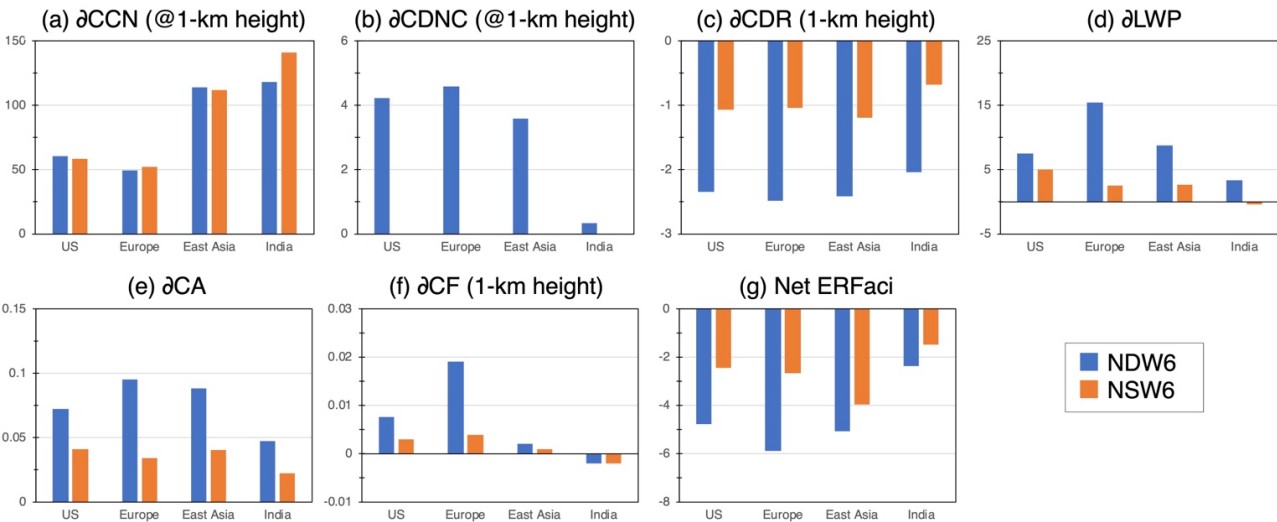

**Figure 13: Regional averages of the differences in CCN at 1-km height, CDNC (only in NDW6), CDR at 1-km height, LWP, CA, CF at a 1-km height, and net ERFaci between the preindustrial and the present days. The regions are defined as US (90°W-60°W, 30°N-50°N), Europe (0°E-30°E, 40°N-60°N), East Asia (110°E-140°E, 20°N-50°N), and India (70°E-90°E, 10°N-35°N).**

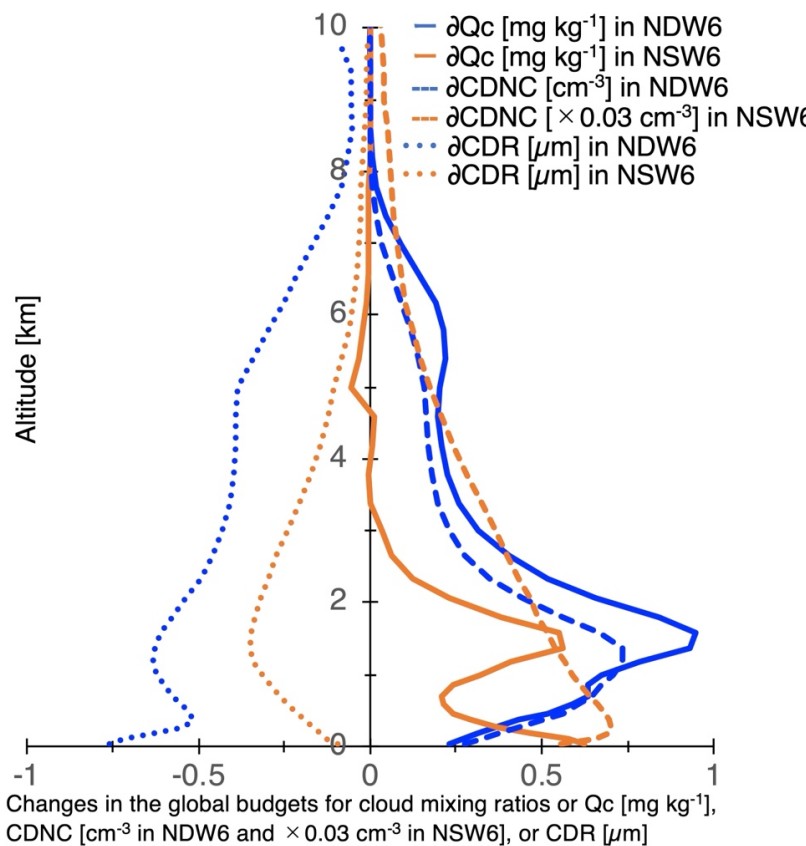

∂Qc [mg kg⁻¹] in NDW6
∂Qc [mg kg⁻¹] in NSW6
∂CDNC [cm⁻³] in NDW6
∂CDNC [×0.03 cm⁻³] in NSW6
∂CDR [μm] in NDW6
∂CDR [μm] in NSW6

Changes in the global budgets for cloud mixing ratios or Qc [mg kg⁻¹], CDNC [cm⁻³ in NDW6 and ×0.03 cm⁻³ in NSW6], or CDR [μm]

**Figure 14: Global budgets of the annual averages of the NDW6- and NSW6-simulated Qc (mixing ratio of cloud droplets), the NDW6-simulated CDNC, the NSW6-simulated CDNC (which is equal to CCN number concentrations), and the NDW6- and NSW6-simulated CDR (cloud droplet effective radius for warm clouds)**

**Table A1: Statistical metrics of PCC, RMSE, and NMB for the annual averages of surface aerosol mass concentrations (OM, BC, and sulfate) between in situ measurements and the NICAM simulations (NDW6 and NSW6 in this study are shown in the panels of Figure 4 and HRM and LRM are shown in Figure 8 in Goto et al., 2020).**

| | NDW6 | NSW6 | HRM | LRM |
|---|---|---|---|---|
| **OM** | | | | |
| **PCC** | 0.847 | 0.846 | 0.819 | 0.794 |
| **RMSE [μg m$^{-3}$]** | 3.40 | 3.34 | 5.03 | 5.21 |
| **NMB [%]** | -30.4 | -25.8 | -54.8 | -56.1 |
| **BC** | | | | |
| **PCC** | 0.904 | 0.904 | 0.890 | 0.869 |
| **RMSE [μg m$^{-3}$]** | 1.05 | 1.03 | 1.16 | 1.28 |
| **NMB [%]** | -53.4 | -51.3 | -46.4 | -52.3 |
| **Sulfate** | | | | |
| **PCC** | 0.807 | 0.853 | 0.815 | 0.768 |
| **RMSE [μg m$^{-3}$]** | 3.97 | 3.67 | 3.94 | 4.34 |
| **NMB [%]** | -10.4 | -3.7 | -14.6 | -23.7 |

**Table A2: Global and annual mean values of the NICAM-simulated aerosol budgets.**

| Species | Parameters [Units] | This study (NDW6) | This study (NSW6) | Goto et al. (2020) (HRM) | Goto et al. (2020) (LRM) | References from model results |
|---|---|---|---|---|---|---|
| **Dust** | Column [Tg] | 13.44 | 17.35 | 27.08 | 27.01 | 16.6 (5.7-22.3)[a] |
| | Emission [Tg yr$^{-1}$] | 1273 | 1414 | 1805 | 1911 | 1440 (848-5659)[a] |
| | Dry Deposition [Tg yr$^{-1}$] | 234 | 289 | 342 | 363 | 396 (37-2791)[b] |
| | Grav. Deposition [Tg yr$^{-1}$] | 380 | 452 | 634 | 663 | 314 (22-2475)[b] |
| | Wet Deposition [Tg yr$^{-1}$] | 669 | 689 | 825 | 880 | 357 (295-1382)[b] |
| | Lifetime [Day] | 3.82 | 4.43 | 5.49 | 5.17 | 3.7 (1.4-7.0)[a] |
| **Seasalt** | Column [Tg] | 5.69 | 6.25 | 5.60 | 5.42 | 8.7 (2.5-26.4)[a] |
| | Emission [Tg yr$^{-1}$] | 10486 | 10048 | 8856 | 9624 | 4980 (2030-50000)[a] |
| | Dry Deposition [Tg yr$^{-1}$] | 2820 | 3006 | 2272 | 2169 | 1313[c] |
| | Grav. Deposition [Tg yr$^{-1}$] | 2169 | 2316 | 1998 | 1951 | 327[c] |
| | Wet Deposition [Tg yr$^{-1}$] | 5498 | 4726 | 4586 | 5504 | 1889[c] |
| | Lifetime [Day] | 0.20 | 0.23 | 0.23 | 0.21 | 0.56 (0.19-1.51)[a] |
| **OM** | Column [Tg] | 0.88 | 1.02 | 1.04 | 0.94 | 1.91 (0.79-2.99)[a] |
| | Emission [Tg yr$^{-1}$][h] | 87.2 | 87.2 | 82.2 | 81.9 | 116.0 (48.0-246.0)[a] |
| | Dry Deposition [Tg yr$^{-1}$] | 7.0 | 7.4 | 6.3 | 6.6 | approximately 15 (0.2-28)[d] |
| | Grav. Deposition [Tg yr$^{-1}$] | 6.1 | 6.8 | 3.7 | 3.9 | |
| | Wet Deposition [Tg yr$^{-1}$] | 74.3 | 73.8 | 72.6 | 71.4 | approximately 90 (approximately 50-140)[d] |
| | Lifetime [Day] | 3.66 | 4.24 | 4.60 | 4.17 | 6.0 (3.4-9.3)[a] |
| **BC** | Column [Tg] | 0.12 | 0.14 | 0.13 | 0.1 | 0.131 (0.068-0.260)[a] |
| | Emission [Tg yr$^{-1}$][h] | 8.3 | 8.3 | 7.3 | 7.3 | 9.7 (8.4-9.7)[a] |
| | Dry Deposition [Tg yr$^{-1}$] | 0.9 | 1.0 | 0.8 | 0.8 | |
| | Grav. Deposition [Tg yr$^{-1}$] | 0.3 | 0.3 | 0.2 | 0.2 | |
| | Wet Deposition [Tg yr$^{-1}$] | 7.1 | 7.0 | 6.3 | 6.3 | |
| | Lifetime [Day] | 5.38 | 6.26 | 6.37 | 4.96 | 5.5 (2.9-8.7)[a] |
| **WSBC** | Column [Tg] | 0.05 | 0.06 | 0.06 | 0.05 | 0.19[f] |
| | Emission [Tg yr$^{-1}$][i] | 5.0 | 5.0 | 4.5 | 4.5 | |
| | Dry Deposition [Tg yr$^{-1}$] | 0.5 | 0.5 | 0.4 | 0.4 | |
| | Grav. Deposition [Tg yr$^{-1}$] | 0.3 | 0.3 | 0.2 | 0.2 | |
| | Wet Deposition [Tg yr$^{-1}$] | 4.3 | 4.3 | 3.9 | 3.9 | |
| | Lifetime [Day] | 3.88 | 4.49 | 4.78 | 4.29 | 6.4[f] |
| **WIBC** | Column [Tg] | 0.07 | 0.08 | 0.07 | 0.05 | 0.03[f] |

| | | | | | | |
|---|---|---|---|---|---|---|
| | Emission [Tg yr$^{-1}$] [i] | 3.3 | 3.3 | 2.8 | 2.8 | |
| | Dry Deposition [Tg yr$^{-1}$] | 0.5 | 0.5 | 0.4 | 0.4 | |
| | Grav. Deposition [Tg yr$^{-1}$] | 0.0 | 0.0 | 0 | 0 | |
| | Wet Deposition [Tg yr$^{-1}$] | 2.8 | 2.8 | 2.4 | 2.4 | |
| | Lifetime [Day] | 7.68 | 9.00 | 8.95 | 6.04 | 1.0[f], 1.0-1.7[g], 9.6 (w/o aging)[g] |
| **Sulfate** | Column [TgS] | 0.45 | 0.52 | 0.38 | 0.32 | 0.60 (0.22-0.98)[a] |
| | Production [TgS yr$^{-1}$] | 57.1 | 57.8 | 58.4 | 56.7 | 37.6-61.1[c] |
| | from the gas phase | 13.3 | 14.6 | 16.8 | 16.1 | 6.2[c] -17.4[e] |
| | from the aqueous phase | 43.7 | 43.2 | 41.7 | 40.6 | 21.1[e]-58.8[c] |
| | Dry Deposition [TgS yr$^{-1}$] | 3.8 | 4.6 | 3.9 | 3.6 | 5.8-7.6[c] |
| | Grav. Deposition [Tg yr$^{-1}$] | 0.1 | 0.1 | 0.5 | 0.4 | 0.0[c] |
| | Wet Deposition [TgS yr$^{-1}$] | 53.4 | 53.4 | 52.0 | 50.4 | 31.8-53.5[c] |
| | Lifetime [Day] | 2.89 | 3.29 | 2.38 | 2.05 | 4.9 (1.8-7.0)[a] |
| **SO$_2$** | Column [TgS] | 0.28 | 0.33 | 0.33 | 0.31 | |
| | Production [TgS yr$^{-1}$][h] | 67.8 | 67.7 | 65.0 | 64.1 | |
| | Chemical loss (gas phase) | 13.3 | 14.6 | 13.0 | 12.5 | |
| | Chemical loss (aqueous phase) | 43.7 | 43.2 | 41.7 | 40.6 | |
| | Dry Deposition [TgS yr$^{-1}$] | 13.0 | 13.2 | 11.7 | 12.0 | |
| | Wet Deposition [TgS yr$^{-1}$] | 2.3 | 1.8 | 1.4 | 1.1 | |
| | Lifetime [Day] | 1.40 | 1.66 | 1.79 | 1.72 | |

[a] Gliß et al. (2021); [b] Huneeus et al. (2011); [c] Takemura et al. (2000); [d] Tsigaridis et al. (2014); [e] Goto et al. (2011); [f] Chung and Seinfeld (2002); [g] Goto et al. (2012); [h] The global and annual mean values in this study (NDW6 and NSW6) are slightly different from those (HRM and LRM) in Goto et al. (2020) because the method of remapping from a latitude-longitude grid emission map to an icosahedral grid in NICAM is modified in this study.

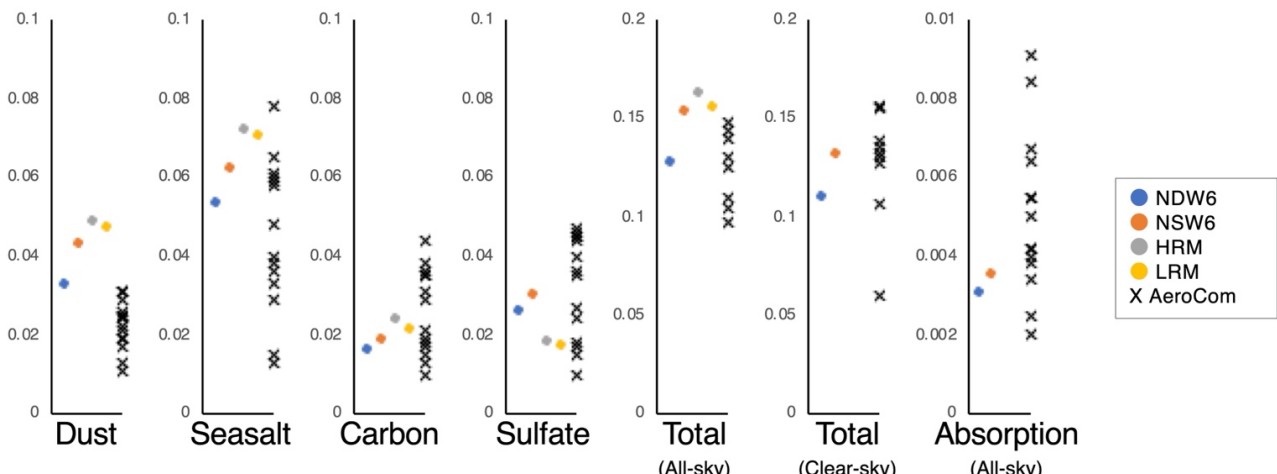

**Figure C1: Global and annual mean values of AOT for chemical components (dust, sea salt, carbonaceous aerosols, and sulfate) under all sky conditions, AOT of total aerosols under both all-sky and clear-sky conditions, and absorption AOT under all-sky conditions. There are no HRM and LRM results for AOT under clear-sky conditions and absorption AOT.**

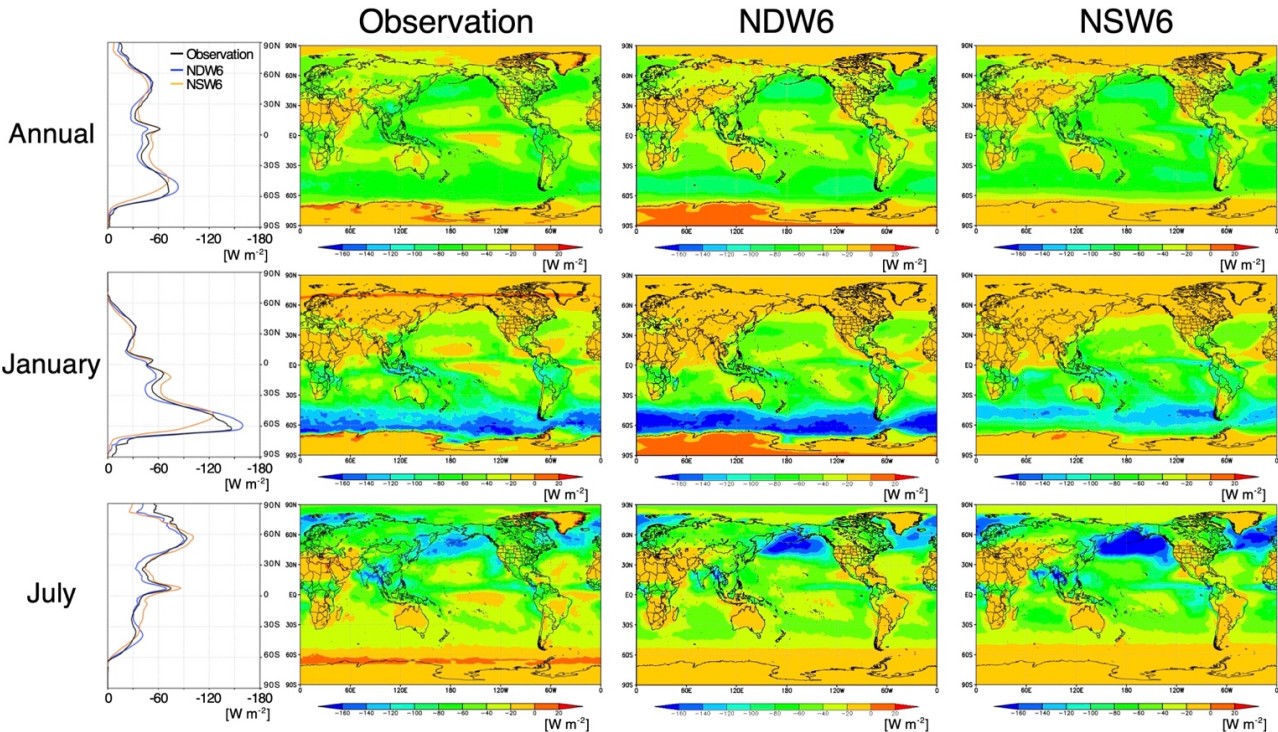

Figure E1: Zonal and horizontal distribution of SWCRF (NDW6 and NSW6 simulations and CERES as observation) as annual, January, and July averages. All units are in W m⁻².

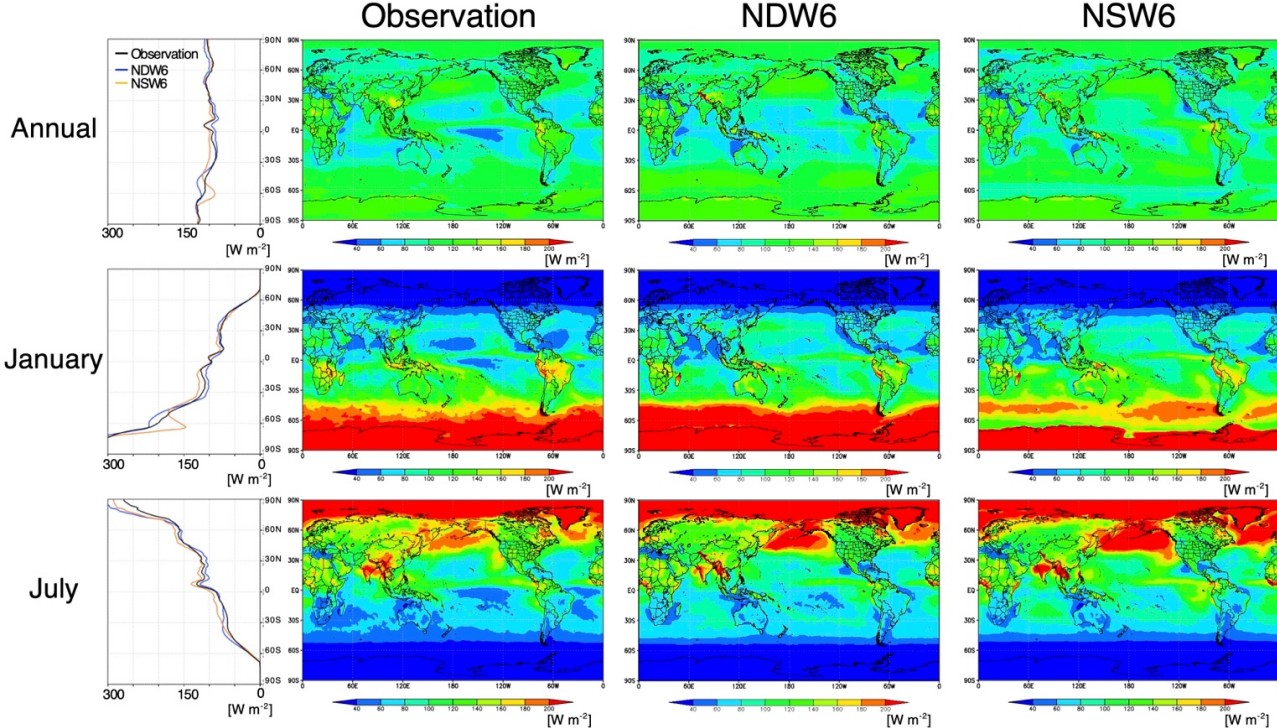

**Figure E2: Same as Figure E1 but for OSR.**