# Peer review of "Impacts of a double-moment bulk cloud microphysics scheme (NDW6-G23) on aerosol fields in NICAM.19 with a global 14-km grid resolution"

_Geoscientific Model Development, 2023_

## Referee Comment (RC1)

**Review of Goto et al, 2023**

This paper compares results from a single moment cloud microphysics scheme (NSW6 simulations) with those from a double moment scheme (NSW6) in global 6-year 14km resolution simulations. The models are also evaluated using observations and other models.

Both versions perform well against observations of LWP and precipitation in a general sense although a deeper analysis of this evaluation would have been useful – e.g., focussing on certain important areas such as stratocumulus regions that are important for the aerosol-cloud interaction radiative forcing.

It is claimed that NDW6 performs better than NSW6, but I don't think that is justified. It is clear that NSW6 performs better in the tropics and NDW6 at high latitudes. It would be better to be upfront about this and to discuss/examine some of the reasons why this is the case.

Hence, the justification for choosing NDW6 over NSW6 is not that strong from the aerosol results. Perhaps this would be better justified by the likely better representation of cloud microphysical processes in NDW6 (e.g, a better representation of fall speeds, hydrometeor separation, droplet evaporation, etc.?). And the large difference in ERFaci are also a good reason to consider NDW6 - although which one is more accurate is unknown... It might be worth mentioning and considering that the LWP response to aerosol could be constrained/evaluated to some degree with a short simulation of a well-observed volcanic eruption, e.g., like the Holuhraun eruption (Malavelle, Nature, 2017; doi:10.1038/nature22974)

There is a large difference in the aerosol-cloud interaction forcing between the NSW6 and NDW6 simulations, but there is less analysis and discussion devoted to that in the paper. For example, there is no analysis about potential differences in the Twomey effect and cloud lifetime adjustments. E.g., how big might these be in NDW6 vs NSW6? Can you compare the PI to PD CDNC differences for NDW6 to the CCN differences in NSW6? Can you compare changes in the cloud fraction (particularly low altitude clouds) – (this should be possible if the simulations were nudged, otherwise it is likely to be noisy.)

It is highly likely that the difference in the autoconversion scheme between NSW6 and NDW6 is having a large impact on the response of the LWP to aerosols (which is very different between the simulations), but this is not mentioned. It should be mentioned and discussed in the results and summary.

In summary, before publication I think this paper needs a bit of work to add some more relevant analysis and discussion – in particular a fairer discussion of the performance of the single moment scheme vs double moment for aerosols and a more thorough analysis of the differences in aerosol-cloud interaction forcing (ERFaci).

**Specific comments**

L126 – "although rain does not directly change cloud water in this case" – It's not clear what you mean here or why this part of the sentence is necessary. Are you saying that there is no scavenging/collection of cloud water by rain? Or that it happens through a calculated microphysical process rate rather than some direct assumption? You'll have to elaborate or decide whether this part of the sentence is needed.

L126 – "Thus, the impact of scavenged aerosols on cloud water is inevitably overestimated in single-moment bulk cloud microphysics schemes" – it's not clear what you mean here either. I can see that there may be an overestimate of the number of droplets present for a given amount of aerosol since droplet numbers cannot evolve and reduce over time in a cloud (partial evaporation, coalescence, etc.) in single moment schemes. What you are saying about scavenging is less clear - are you saying that the impact of aerosol changes is likely to be larger in the single moment scheme due to this direct link between CCN and droplet number? And that this would include a larger effect due to removal of aerosol by scavenging? This would need rewriting to say that if so.

L191 – "sulfate are assumed to have unimodal particle size distributions" – what are the shapes of the size distributions? Lognormal?

L207 onwards – you describe the length of the simulations and some other details here – but this is in the aerosol section, which makes it hard to find. The information on the model run length, spin-up, etc. should be in its own subsection. Also you need some details on whether they were free running or nudged to meteorology?

Plus, the details on the satellite data used for model evaluation should be in its own subsection too and not in the aerosol section.

L209 – "which are obtained from the end of the 1-year aerosol online simulations coupled to NSW6." – you will need to explain what these "online simulations" are? What do you mean by "online"?

L261 – "However, the seasonal and horizontal biases of the NSW6-simulated LWP are effectively cancelled, and the global and annual mean values of the NSW6-simulated LWP appear closer to the MAC results." – I can't see much evidence for the NSW6 LWP values having compensating seasonal and horizontal biases…

L262 – "In terms of the distribution pattern and seasonal cycle, the NDW6-simulated LWP is closer to the MAC result compared to the NSW6 result." – I don't see any evidence for this. Although I would be wary of trusting the MAC LWP values too much in precipitating regions like the tropics since precipitation can cause biases.

L275 – "the results in NDW6 are generally better and closer to those of the real atmosphere." – I'm afraid that I don't think that you can make this statement with the evidence presented. The differences in precipitation (Fig. 1) between NSW6 and NDW6 are very small and it's hard to say which is best when looking at the spatial maps for January and July. Similarly for LWP. Perhaps NDW6 is looking a bit better for July for the higher latitudes, but it's quite hard to tell with the colour range chosen – maybe you need to narrow the colour range on the maps to make the differences clearer? But I think it's clear from the maps and the zonal means that NSW6 does better in the tropics and NDW6 better at higher latitudes even when taking into account the spatial variability and individual months. This would also need addressing at L517.

L288 – "the recent models participating in the AeroCom Phase-III project" – are these what you term "references" in Figure 4? This should be explained in the caption. It would also be better to label this as "Aerocom models" or similar in the figure(s). Similarly, for Fig. 9 – you need separate markers and labels for Kinne (2019) and Thorsen (2021) and you need to explain why there are several crosses for all-sky and clear-sky for "All" and "Anthropogenic", but not the other plots in the caption – e.g., are they for different AeroCom model members? Are there also several crosses for Thorsen and what are they if so?

L310 – "the NDW6-simulated RPCW is much closer to the CloudSat-retrieved RPCW" – again, I have to disagree here. It is true for the subtropical regions and higher latitudes, but not for the tropics.

L320 – "The difference in the column burden of SO2 between NDW6 and NSW6 is caused by the chemical loss in the aqueous phase (0.7 TgS yr-1 or +2%) and gas phase (1.1 TgS yr-1 or -7%) and wet deposition (0.4 TgS yr-1 or +24%), as shown in Table A1." – these values seem different to those quoted in Table A1?

L323 – Need to describe HRM and LRM somewhere in the main paper.

L375 – "example, the IRFari dust values are calculated to be -0.46 Wm-2 (NDW6), -0.57 Wm-2 (NSW6), and -0.24 Wm-2 (Kinne, 2019)." – need to make it clear that these are for all-sky.

L375 – it would be good to add a bit of detail about what kind of data Kinne (2019) and Thorsen (2021) represent. I.e., model aerosol reanalysis, satellite observations, etc.

L376 – "This is partly caused by the weaker absorption of AOT in this study compared to the median value of the AeroCom models" – But it also could be due to the higher dust AOT values?

L379 – "This is inconsistent with the results of the larger column burden and AOT of dust in this study compared to those of the AeroCom models" – you should point out that it is consistent with too little SW absorption, though.

L388 – "For other nonlight-absorbing components, i.e., sea salt and sulfate, the difference in the IRFari values between all-sky and clear sky conditions is very small" – this doesn't look to be the case? The differences between all-sky and clear-sky are largest for sulphate and sea-salt?

L390 – Make it clear that this is for all-sky. Same for L396.

L451 – "The difference in the ERFaci between NDW6 and NSW6 may be partly explained by a nonlinear relationship of the ERFaci to AOT ==under the different LWPs==, as proposed by Carslaw et al. (2013) who argued that even if the aerosol difference between PI and PD is similar, the value of ERFaci can be larger when the aerosol concentration is lower."

- Presumably you mean here "for different baseline AOT fields"?
- Also "even if the aerosol difference between PI and PD is similar" would be clearer as "even if the PI to PD aerosol difference for two simulations are similar".

L455 – "Figure 11 shows that the horizontal distribution of changes in the simulated LWP" – it would be good to introduce the LWP adjustment as another potential factor in causing the difference in the ERFaci between NDW6 and NSW6 (since you have previously suggested that the baseline AOT is a potential cause).

But what about potential differences in the Twomey effect and cloud lifetime adjustments? How big might these be in NDW6 vs NSW6? Can you compare the PI to PD CDNC differences for NDW6 to the CCN differences in NSW6? Can you compare changes in the cloud fraction (particularly low altitude clouds) – this should be possible if the simulations were nudged. Otherwise it is likely to be noisy.

L466 – "Notably, the reason for the differences in the susceptibility between NDW6 and NSW6 should be addressed in future studies."

- Although it is clear that the difference in the autoconversion scheme between NSW6 and NDW6 is likely to have a large impact here and so this should be mentioned and discussed here and in the summary too.

L496 – "The differences in the dust emissions, dust column burden and SO2, AOT, and IRFari values for total aerosols between NDW6 and NSW6 are larger." – at L480 you said that the AOT differences were small, so this is a bit of a contradiction.

L508 – "These differences are mainly caused by the difference in the susceptibility of the LWP to AOT" – in the results section this wasn't stated so clearly. It would be good to mention that in the results too – although do you have evidence that this is the case?

L518 – "and thus, the use of NDW6 is recommended in environmental and climate simulations."

- The justification regarding the LWP and precipitation performance here is not very strong (see above regarding the NSW6 vs NDW6 performance). Perhaps this would be better justified by the likely better representation of cloud microphysical processes in NDW6 (e.g, a better representation of fall speeds, hydrometeor separation, droplet evaporation, etc.?). And the large difference in ERFaci are also a good reason to consider NDW6 - although which one is more accurate is unknown... It might be worth mentioning and considering that the LWP response to aerosol could be constrained/evaluated to some degree with a short simulation of a well-observed volcanic eruption, e.g., like the Holuhraun eruption (Malavelle, Nature, 2017; doi:10.1038/nature22974)

L529-531 – I can't see the PCC, etc. values from Goto (2020) listed anywhere – these should be included or quoted in this paper somewhere – ideally in a table so that the reader can compare the new simulations to the old ones.

Table A1 – it would be useful to say what kind of observations the difference references use (satellite, model reanalysis, models, etc.).

L554 – "whereas the difference in the cloud microphysics module does not affect the chemical budget of SO2 oxidation" – presumably you are talking about the cloud microphysical changes within the NSW6 module here rather than the difference between double and single moment (since there are large differences for SO2 between NSW6 and NDW6)? You should make this clearer if so.

**Typos / grammar**

L18 – "but some aerosol species, especially dust and sulfate, have larger differences among the experiments with NDW6 and NSW6 compared to those among the experiments with different horizontal resolutions, i.e., 14 km and 56 km grid spacing,…"

- Would be better as "but differences between the NDW6 and NSW6 experiments are larger for some aerosol species, especially dust and sulfate, compared to those between experiments with different horizontal resolutions, i.e., 14 km and 56 km grid spacings,…"

L48 – "and hence, an elaboration of both the cloud module and aerosol physics module is required to improve ACI in climate models". "Elaboration" is perhaps not the best choice of words. I recommend "evaluation" instead.

L52 – "It is promising that convective cloud systems are better represented with finer model resolution when cumulus parameterizations are avoided" – I would recommend this instead: "These results suggest that convective cloud systems are better represented with a finer model resolution for which cumulus parameterizations are avoided"

L72 – "solved" -> "resolved"

L88 – "calculcated" -> "run"

L96 - "incorporated to" -> "incorporated into"

L97 – "were reflected to the version in NICAM.19" -> "was incorporated into the version in NICAM.19"

L107 – "to aerosol physics module" -> "to the aerosol physics module"

L157 – "concentration higher" -> "concentration is higher"

L167 – "dependence of leaf area index" – should be "dependence on leaf area index"?

L175 – "OM" has not been defined.

L176 – "Secondary organic aerosols (SOAs) are assumed to be particles by multiplying the emission fluxes of isoprene and terpenes provided by" -> "Secondary organic aerosols (SOAs) are assumed to form particles, which are calculated by multiplying the emission fluxes of isoprene and terpenes provided by" – also, what are they multiplied by? A constant factor?

L178 – "Parts of SO2 are emitted from volcanic eruptions (Diehl et al., 2012) and are formed from DMS," – "SO2 is emitted from volcanic eruptions (Diehl et al., 2012) and is also formed from DMS,"

L217 – "These identical datasets were prepared and used in Goto et al. (2020)" – I assume you mean "The same datasets were prepared and used in Goto et al. (2020)". Otherwise it makes it sound like all of the observational datasets are identical.

L249 – "by satellite results" -> "using satellite data".

L268 – "CF at the low level" -> "low-altitude CF"

L306 – need to define what WSBC and WIBC are somewhere.

L394 – "which can be caused by its lower lifetime among the references" -> "which may be due to its short lifetime relative to the values from Kinne (2019) (and Thorsen?)".

L395 – "sea salt is more scavenged by wet deposition" -> "sea salt is scavenged more by wet deposition"

L418 – "but comparable to" -> "but are comparable to"

L422 – "are smaller than"-> "are smaller in magnitude than"

L427 – "fluxes are compared for model evaluations of radiation budget." -> "fluxes are compared and evaluated."

L427 – "the global and January averages of the SWCRF" -> "the global averages of the SWCRF for January". Similarly, for L428 for July and L436 for global averages.

L442 – "by comparing the results between NDW6 and NSW6 under the preindustrial (PI) and the present day (PD)" – better to be clear that this means PI and PD emissions for aerosols and precursor gases. E.g., "by comparing the results between NDW6 and NSW6 for simulations with preindustrial (PI) and the present day (PD) aerosol and precursor gas emissions". Similarly for the caption of Fig. 11.

L457 – "such as the United States, Europe, and East Asia, increase" -> "such as the United States, Europe, and East Asia, increase in magnitude"

L457 – "the NDW6-estimated ERFaci value is larger negatively than the NSW6-estimated ERFaci" -> "the negative NDW6-estimated ERFaci values are larger in magnitude than the NSW6-estimated ERFaci values"

L460 – "key to understand the difference" -> "key to understanding the difference"

L502 – "whereas those in the sulfate are mainly caused by the wet deposition of SO2." -> "whereas those in the sulfate are mainly caused by the differences in the wet deposition of SO2."

L530 – "or" -> "of"

L538 – "Surely," – wrong choice of phrase here. Would be better as "In support of this,"

Figure E1 – the caption has become jumbled.

---

## Author Comment (AC3)

Reviewer3

[C3-1] In this study, the authors quantify the impact of a double-moment cloud microphysics scheme on aerosols in a high-resolution global model, comparing to its single-moment counterpart in the same model. The description of results and comparison between the two and with published results in the literature are quite comprehensive, but the explanations of differences are often handwaving. Particularly interesting result is the much higher ERFaci in NDW6 than in NSW6. In my opinion, more in-depth analyses in terms of how the two schemes treat aerosol indirect effects on clouds and precipitation in the model are needed. Also, what's the implication of such results for the use of double-moment microphysics in high-resolution climate models?

[A3-1] We appreciate your great contributions to improve our manuscript. Your comments and suggestions are very helpful and motivate us to investigate the results more deeply, especially the analysis of aerosol-cloud interaction in our models in Section 4.2 in the revised manuscript. We believe our revision is acceptable for you.

The comment of "what's the implication of such results for the use of double-moment microphysics in high-resolution climate models?" is also important point. The high-resolution is certainly important to simulate aerosols and clouds more realistically, as shown in this study and our previous study (Goto et al., 2020). The advantage of using double-moment cloud microphysics scheme is that the cloud microphysics representation of double-moment microphysics (NDW6) is more elaborate than that of single-moment microphysics (NSW6). In addition, this study showed that the NSW6-simulated cloud droplet radii for water clouds (CDR) was overestimated due to the inability to predict cloud droplet number concentrations in NSW6. These points were discussed in section 4.2 and mentioned in section 5 in the revised manuscript.

Through the revision, we modified figures and tables as follows:

- Figures 1, 2, 5, E1 and E2: We changed the color of the zonal averages (NDW6 blue, NSW6 orange, as in the other figures).
- Figures 2 and 5: We replotted the model results in white for grids with missing satellite data. We replotted the zonal averages of the model results by eliminating the grids with missing satellite data.
- Figure 4: We changed the caption named "references" to "AeroCom".
- Figure 7: We changed the subtitle named "AOD" to "AOT".
- Figure 9: We provided IRFari of all and each aerosol for shortwave & longwave at the TOA & the surface under all & clear sky conditions. We changed the caption named "references" to "Kinne19" and "Thorsen21".
- Figure 10: We removed the results of IRFari because they were shown in Figure 9. Instead, we newly added the results of ERFari and sum of ERFari and ERFaci. We also modified the ERFaci for shortwave to net ERFaci (for both shortwave and longwave).
- Figure 11: We modified $\partial$AOT to $\partial$CCN. We added new parameters such as $\partial$CDNC, $\partial$CDR, $\partial$CA, $\partial$CF, and net ERFaci to further explore ACI.
- Figure 12: We replaced Table 3 in the original manuscript to Figure 12 in the revised manuscript by adding relevant parameters such as $\partial$CCN, $\partial$CDNC, $\partial$CDR, $\partial$CA, $\partial$CF, and net ERFaci.
- Figure 13: To explain possible overestimations of the NSW6-simulated Twomey effect, we newly plotted global budgets of the annual averages of the NDW6- and NSW6-simulated $Q_c$ (mixing ratio of cloud droplets) and CDNC (cloud droplet number

concentrations).
- Table 1 in the original manuscript: We removed it and added a paragraph to explain the HRM and LRM as references in section 2.5 in the revised manuscript.
- Table 1 in the revised manuscript: We simply moved Table 2 in the original manuscript to Table 1 in the revised manuscript.
- Table 2 in the revised manuscript: We showed global and annual mean values of ERFari, ERFaci, and the sum of ERFari and ERFaci for shortwave, longwave, and net radiation under both all-sky and clear-sky conditions.
- Table A1: We newly added the statistical metric to compare results in this study with the references by Goto et al. (2020).
- Table A2: We simply moved Table A1 in the original manuscript to Table A2 in the revised manuscript, with two exceptions. One, we changed "References" to "References from model results". Second, we changed the $SO_2$ production value from 67.5 to 67.7.

Please note that some English was corrected in the revised manuscript.

[C3-2] It's hard to find specific information regarding the setup of experiments (NDW6, NSW6, HRM, LRM), except for the resolutions and microphysics schemes in Table 1. Which specific simulation years? What are the aerosol emissions used for the simulation years, mean or year-specific? Such information is important to determine whether the comparison of simulations results with observations and other models in the literature is valid. Please include the details in Table 1.

[A3-2] Thank you for your comment on the setup of experiments. As you pointed, some of explanation in the original manuscript was missed. So, we added section 2.3 named "Experimental conditions" including the information of the setup and emission inventories to the revised manuscript (Lines 201-216) as follows:

"All experiments with both NDW6 and NSW6 are carried out for 6-years after the 1-month spin-up calculation. The simulation results are climatological runs, because the model does not nudge meteorological fields such as wind and temperatures but nudges the sea surface temperature (SST) and sea ice by the results of the NICAM from Kodama et al. (2015). The initial conditions for the model spin-up are obtained from the end of the 1-year aerosol simulations coupled to NSW6 without nudging the meteorological fields under the present era.

The emission fluxes used in this study are the Hemispheric Transport of Air Pollution (HTAP)-v2.2 (Janssen-Maenhout et al., 2015) for BC, organic carbon (OC) and $SO_2$ from anthropogenic sources in 2010 and the Global Fire Emission Database (GFED) version 4 (van der Werf et al., 2017) for BC, OC and $SO_2$ from biomass burning in climatological average from 2005 to 2014. The ratio of OC to OM is set at 1.6 for anthropogenic activities and 2.6 for biomass burning (Tsigaridis et al., 2014). Secondary organic aerosols (SOAs) are assumed to form particles, which are calculated by multiplying the emission fluxes of isoprene and terpenes provided by the Global Emissions Initiative (GEIA) (Guenther et al., 1990) using constant factors. $SO_2$ is emitted from volcanic eruptions (Diehl et al., 2012) and is also formed from DMS, which is interactively emitted in the aerosol module (Bates et al., 1987). Sulfate is formed from $SO_2$ oxidation with a 3-dimensional distribution of monthly oxidants (ozone, $H_2O_2$ and OH) provided by a chemical transport model (CHASER) coupled to MIROC (Sudo et al., 2002). Emission

fluxes for dust (Takemura et al., 2009) and sea salt (Monahan et al., 1986) are interactively calculated in the model using mainly the wind speed at a height of 10 m."

In addition, we added a section 2.5 in the revised manuscript to explain the information of the models, i.e., HRM and LRM in Goto et al (2020) as references. Because we added this section to the revised manuscript, we removed Table 1 in the original manuscript. The section 2.5 in the revised manuscript (Lines 281-295) is as follows:

"Our previous model results provided in Goto et al. (2020) using NICAM.16 at a global 14-km high resolution (hereafter referred to as the HRM) and a global 56-km low resolution (hereafter referred to as the LRM) are used as references to compare the NICAM results. As mentioned in section 2.1, the number of vertical layers is set at 38, and the timestep is 1 minute in both the HRM and LRM. The integration periods in both the HRM and LRM are 3 years as climatological runs. The emission inventories, i.e., 2010 for anthropogenic sources, climatological average in 2005-2014 for biomass burning, and natural sources in the present era, and the nudged SST and sea ice in this study are identical to those in both the HRM and LRM, but the initial conditions in this study are different from those in both the HRM and LRM, which use the model results at the end of December after a 1.5-month spin-up. The initial conditions for the model spin-up are prepared by the reanalysis datasets of the National Centers for Environmental Prediction (NCEP) Final (FNL) (Kalnay et al., 1996) in November 2011. In the cloud microphysics and autoconversion modules, NDW6 coupled to Seifert and Beheng (2006) and NSW6 coupled to Khairoutdinov and Kogan (2000) are used in this study, whereas NSW6 coupled to Berry (1967) is used in both the HRM and LRM. The improvement in the aerosol module described in section 2.2 is also different from that in the HRM and LRM. The results of the HRM and LRM are useful for evaluating the current model results because the observations are limited in some parameters, such as aerosol global budgets and radiative forcings."

[C3-3] At many places, IRFari and ERFaci are referred to as "shortwave" aerosol forcing. If the longwave component is not considered at all, I don't think they are comparable to the cited values in the literature, which mostly include both shortwave and longwave components and are referred to as net radiative forcing. Please confirm and clarify.

[A3-3] Thank you for your suggestion. Surely, we didn't show the results of IRF and ERF for longwave as well as net. We put these estimates in the revised Figure 10 and Table 2. We modified Figure 10 and added net ERFaci values to the revised manuscript.

[Figure]

Figure 10: Global and annual mean values of (a) effective radiative forcing for anthropogenic aerosol-radiation interaction (ERFari) for shortwave and net (sum of shortwave and longwave) radiation, (b) ERFaci for anthropogenic aerosol-cloud interaction, and (c) the net ERF (sum of ERFari and ERFaci). All units are in W m⁻². In ERFari, the reference of Forster21 is estimated in the net radiation by IPCC-AR6 or Forster et al. (2021), whereas the reference of Thortsen21 is estimated in the shortwave radiation by Thorsen et al. (2021). The reference for Smith20 is Smith et al. (2020). The values are also listed in Table 2.

Table 2: Global and annual mean values of ERFari for anthropogenic aerosol, ERFaci for anthropogenic aerosol-, and the net ERF (sum of ERFari and ERFaci) for shortwave, longwave, and net (sum of shortwave and longwave) radiation under the all-sky abnd clear-sky conditions. All units are in W m⁻².

|  | ERFari under the all-sky conditions | | | |
|  | NDW6 | NSW6 | HRM | LRM |
|---|---|---|---|---|
| Shortwave | -0.22 | -0.26 | -0.33 | -0.26 |
| Longwave | 0.03 | 0.03 | 0.04 | 0.02 |
| Net | -0.19 | -0.23 | -0.29 | -0.24 |
|  | ERFari under the clear-sky conditions | | | |
| Shortwave | -0.52 | -0.60 | -0.63 | -0.51 |
| Longwave | 0.05 | 0.05 | 0.05 | 0.03 |
| Net | -0.47 | -0.55 | -0.57 | -0.48 |
|  | ERFaci | | | |
| Shortwave | -1.34 | -0.63 | -0.81 | -1.17 |
| Longwave | 0.06 | -0.10 | -0.12 | 0.07 |
| Net | -1.28 | -0.73 | -0.93 | -1.10 |
|  | ERFari+ERFaci | | | |
| Shortwave | -1.56 | -0.89 | -1.15 | -1.43 |
| Longwave | 0.09 | -0.07 | -0.08 | 0.09 |
| Net | -1.47 | -0.96 | -1.23 | -1.34 |

[C3-4] Both NDW6 and NSW6 appear to simulate a too weak SWCRF and LWCRF (shown in Table 2), as compared to observations and historical mean of CMIP models. Is this due to interannual variability or model bias? How does this affect the evaluation of aerosols and their associated forcings in NDW6? Please include a discussion on this issue.

[A3-4] Thank you for your comment on CRF. We checked the results of CRF in all six years and found that the standard deviations were smaller than the difference between NDW6 and CERES. The standard deviations were calculated to be 0.2 Wm$^{-2}$ (NDW6, Annual), 0.6 Wm$^{-2}$ (NDW6, January), and 0.2 Wm$^{-2}$ (NDW6, July). Therefore, the difference between NICAM and CERES is not caused by the internal variability.
In the beginning of Section 4.2, we roughly mentioned the results of SWCRF and the bias, and discussed the spatial and zonal distributions in Appendix E. As mentioned in Appendix A, the SWCRF results are generally consistent to the LWP results. As mentioned there and in section 3.1 related to LWP, the global averages in NDW6 are not closer to the observation compared to those in NSW6, but this can be caused by compensation errors in space (In the original manuscript, we mentioned the error in space and time, but after checking again, we found that it was caused by the error only in space. We modified this in the revised manuscript (Lines 328-329, 340, 515, 621, and 659)). The NICAM results of both SWCRF and LWP are underestimated compared to the observation in most regions, but the NSW6 has some positive bias of SWCRF and LWP in some regions such as the western Pacific Ocean. As a result, the zonal and global averages of SWCRF and LWP in NDW6 are underestimated relative to NSW6. Therefore, we cannot conclude that the bias of SWCRF and LWP in NDW6 is larger than that in NSW6.
The large underestimation of SWCRF in the western Pacific Ocean is related to the underestimation of the clouds. The underestimation of the low-level clouds in NICAM were realized in NICAM studies (e.g., Kodama et al., 2021). Since the NICAM-simulated precipitation is close to the observation, the NICAM-simulated cloud-to-precipitation conversion is overestimated, as shown in Figure 5. This causes the overestimation of the wet deposition for aerosols as shown in Figure 7; therefore, the underestimation of the NICAM-simulated SWCRF can cause the underestimation of the simulated aerosols in such tropic regions. This is a possible bias to be caused by the difference in the SWCRF between the model and observation.
Over land where the observed LWP information is very limited, the observed SWCRF may be useful to evaluate the model. In high aerosol loading areas such as China, Europe, and the United State, the simulated SWCRF tends to be underestimated compared to the observation. This indicates the underestimation of the simulated LWP and/or the overestimation of the simulated cloud droplet effective radius (CDR). When the simulated LWP is underestimated, the simulated aerosols are underestimated, according to the above discussion. When the simulated CDR is overestimated, the simulated CCN is underestimated. This is consistent to the underestimation of the simulated aerosol. Therefore, if negative biases in the simulated SWCRF are eliminated, the simulated aerosols will increase. In the revised manuscript (Lines 516-522), we modified this as follows:
"The NDW6-estimated SWCRF values are concluded to be better than the NSW6 results, but the underestimation of the simulated SWCRF can have an impact on the aerosol simulations. The underestimation of the simulated SWCRF indicates the underestimation of the simulated LWP and/or the overestimation of the simulated CDR. When the

simulated LWP is underestimated, the simulated aerosols are also underestimated because the simulated precipitation is generally comparable to the observations in this study, as shown in Figure 1. When the simulated CDR is overestimated, the simulated CCN must be underestimated. This is consistent with the underestimation of the simulated aerosol. Therefore, if the negative biases in the simulated SWCRF are eliminated, the simulated aerosols will increase."

As for longwave CRF, the impacts of the longwave CRF (LWCRF) on aerosols are small in this study, because the LWCRF is mainly determined from the high-level clouds and convective clouds and our model ignored the direct interaction between aerosols and ice crystal (as an ice nuclei). Therefore, if negative biases in the simulated LWCRF in this study are eliminated, the simulated high-level clouds increase but the simulated aerosols may not be changed. Because the direct interaction between aerosols and ice clouds is not considered in this model, the impacts of the bias of the simulated LWCRF are unclear. In the revised manuscript (Lines 523-525), we modified this as follows:

"The underestimation of the simulated LWCRF is caused by the underestimation of the simulated high-level clouds, but the impacts of this negative biases in the simulated LWCRF on the aerosol simulations are unclear due to ignorance of the interaction between aerosols and ice crystals (as ice nuclei) in this model."

**Minor comments and technical corrections:**

[C3-5] L118-119: If CDNC is fully prognostic in the NDW6 double-moment scheme, I assume it's always updated with source and sink tendencies. Why is there an additional constraint by CCN that depends on supersaturation as well? please clarify.

[A3-5] We apologize for the lack of explanation about this part "In addition, a CDNC value is assumed to be updated to a CCN value only when the CCN value exceeds the CDNC value in a grid box". The CCN value only when the CCN value exceeds the CDNC value in a grid box is an aerosol activation process and a source term. The sink tendencies are accretion, autoconversion, and evaporation for water clouds. As you mentioned, the CDNC is determine by source and sink tendencies.

The additional constraint by CCN that depends on supersaturation is needed to nucleate water clouds (not to nucleate ice clouds). One reason of this is an aerosol activation process parameterized by Abdul-Razzak and Ghan (2000) is applicable only for water clouds but calculate in any clouds even below supersaturation in the aerosol physic model.

Therefore, we modified this in the revised manuscript (Lines 122-126) as follows:

"In addition, a source term of CDNC is assumed to be updated to a CCN value only when the CCN value exceeds the CDNC value in a grid box. The CDNC is updated with source (aerosol activation) and sink (autoconversion, accretion, and evaporation for water clouds) in NDW6 (Seiki and Nakajima, 2014). The balance of source and sink tendencies determines the CDNC in NDW6."

[C3-6] L123: Please clarify on "which" is updated in this study. NDW6 or NSW6?

[A3-6] Thanks you for your comment. This is NSW6. To avoid unclear expression, we removed the part 'which is updated in this study (cf.., Seiki and Roh, 2020)" from the revised manuscript.

[C3-7] L132: Is there no shallow convection parameterization at 14-km grid spacing? Please justify.

[A3-7] We also do not use shallow convection schemes, like other studies using NICAM with 14-km grid spacing (e.g., Satoh et al., 2010; Kodama et al., 2021). We modified this in the revised manuscript (Lines 143-144) as follows:
"As in previous studies using the NICAM (e.g., Satoh et al., 2010; Kodama et al., 2021), no parameterization schemes for deep and shallow convection are used in this study."
We realized possible biases caused by not including a shallow convection parameterization. For example, the simulated low-level clouds and shortwave radiation such as OSR and SWCRF in NICAM tend to be underestimated compared to the observation (Kodama et al., 2021). This can also influence SST and is very important for atmosphere-ocean coupling models to predict SST. For example, Masunaga et al. (2023) introduced a flux adjustment to the atmosphere-ocean coupled NICAM without shallow convection parameterization to avoid SST drift. However, we don't think the exclusion of the shallow convection parameterization is so important for atmospheric circulation models with fixed SST. Therefore, we don't use any shallow convection parameterization like previous studies using the NICAM as an atmospheric circulation model (e.g., Satoh et al., 2010; Kodama et al., 2021).

Reference:
Masunaga, R., Miyakawa, T., Kawasaki, T., Yashiro, H.: Flux adjustment on seasonal-scale sea surface temperature drift in NICOCO, Journal of the Meteorological Society of Japan. Ser. II, 101(3), 175-189, doi:10.2151/jmsj.2023-010, 2023

[C3-8] L157-159: This might be relevant to the comment for L118-119. This treatment needs more clarification and justification. If the aerosol scheme and the NDW6 are fully coupled, I don't see why this constraint is justified for certain conditions only. CCN number, which is only meaningful with supersaturation specified, can be larger than CDNC before the activation tendency is updated to CDNC. Otherwise, CCN diagnosed from interstitial aerosols should be mostly smaller than CDNC in clouds. I wonder whether partial-grid clouds matter here.

[A3-8]
We apologize for confusing you due to the lack of explanation about this. Like our answer in [A3-5], we would like to mention the source tendency of the CDNC. Therefore, we modified this in the revised manuscript (Lines 154-156) as follows:
"First, when the CCN number concentration is higher than the CDNC calculated online in the aerosol module, the value of water supersaturation is positive, and the atmospheric pressure is above 300 hPa, the CCN number concentration becomes an input of source tendency for CDNC."

[C3-9] L174: Change anthropogenic "materials" to "sources" or "activities".

[A3-9] Thanks. Changed.

[C3-10] L207: Does that mean there is no BC, OC and SO2 emission in the extra experiment except for fire and volcanos? I wonder how this experiment is different from the preindustrial-condition experiment, which should also have emissions in the anthropogenic sectors (e.g., residential, agricultural waste burning, etc.)

[A3-10] Thank you for your comment on the experimental setup in preindustrial condition in this study. Yes, as you said, our condition is slightly different from the preindustrial condition in the other studies, because the other studies use the 1850 (or 1750) aerosol emissions. As you mentioned, the residential sector in the preindustrial time has a large contribution to the total amount (Hoesly et al., 2018). This point is important but was not discussed in the original manuscript. We wanted to recalculate the simulations using the 1850 aerosol emissions, but unfortunately unable to do them due to the limitation of our computer resources. So, we would like to speculate the uncertainty of the ERF caused by the different assumption of the preindustrial era. As a reference, the results in Hoesly et al. (2018) are shown in Table. We added the following discussions to the revised manuscript (Lines 217-227):

"In the preindustrial experiments, the anthropogenic emission fluxes of BC, OC and $SO_2$ are assumed to be zero in this study. Hoesly et al. (2018) estimated global averages of the differences in the emission amounts of anthropogenic sources between 1850 and 2010 to be 2.1% (sulfate), 12.0% (BC), and 22.7% (OC). The residential sector has the largest contribution to the total anthropogenic emissions in the preindustrial era. Takemura (2020) calculated the IRFari due to anthropogenic sulfate under the conditions of 0% and 30% of the present emissions and found that the difference in the IRFari was within 0.03 $Wm^{-2}$. Therefore, differences in the assumptions for the preindustrial era between this study and other studies, such as IPCC-AR6 (Szopa et al., 2021), will result in a difference in the IRFari due to anthropogenic sources of at most 0.05 $Wm^{-2}$. Takemura (2020) also calculated ERFari and ERFaci due to anthropogenic sulfate under the conditions of 0% and 30% of the present emissions and found that the difference in ERFari plus ERFaci was within 0.2 $Wm^{-2}$. These are possible uncertainties in the estimated radiative forcings due to anthropogenic sources in this study, but these magnitudes are smaller than the difference between NDW6 and NSW6 in this study, as shown in section 4."

We also added the following sentence to the summary (Lines 651-657) in the revised manuscript:

"As mentioned in section 2.3, the assumption of the preindustrial conditions for aerosols can cause possible differences in the aerosol radiative forcing due to the anthropogenic sources between this study and other studies, such as IPCC-AR6 (Szopa et al., 2021). This study assumes that the anthropogenic emission fluxes of BC, OC and $SO_2$ are zero in the preindustrial conditions, whereas other studies often use them in 1750 or 1850 provided by Hoesly et al. (2018). Using the results of MIROC by Takemura (2020), the possible difference in the aerosol radiative forcing due to the anthropogenic source will be at most 0.05 $Wm^{-2}$ (IRFari) and 0.2 $Wm^{-2}$ (ERFari plus ERFaci)."

Table Emission amounts in 1850 and 2010 years by Hoesly et al. (2018)

| Anthropogenic Source | 1850 | 2010 | 2010-1850 [%] |
|---|---|---|---|
| $SO_2$ [$ktSO_2$] | 2,481 | 115,487 | 2.1 |
| BC [ktC] | 934 | 7,755 | 12.0 |
| OC [ktC] | 4,262 | 18,755 | 22.7 |

Reference:
Hoesly, R. M., Smith, S. J., Feng, L., Klimont, Z., Janssens-Maenhout, G., Pitkanen, T., Seibert, J. J., Vu, L., Andres, R. J., Bolt, R. M., Bond, T. C., Dawidowski, L., Kholod, N., Kurokawa, J., Li, M., Liu, L., Lu, Z., Moura, M. C. P., O'Rourke, P. R., and Zhang, Q.: Historical (1750-2014) anthropogenic emissions of reactive gases and aerosols from the Community Emissions Data System (CEDS), Geosci. Model Dev., 11, 369-408, doi:10.5194/gmd-11-369-2018, 2018.

[C3-11] L247-249: the use of "products" and "results" for model simulations and satellite, respectively, should be the other way around, i.e., referred to as simulation results and satellite products.

[A3-11] Thanks. Changed.

[C3-12] L261: What does the "horizontal biases" mean? Spatial or regional biases?

[A3-12] Thanks for your comment. Yes, this "horizontal" means "spatial" or "regional". We removed "horizontal biases" from the revised manuscript. Related to this comment, we found that the compensation errors were found only in the horizontal distribution (not found in seasonal distribution). Therefore, we modified the analysis of the differences in the LWP between NDW6 and NSW6 to the revised manuscript (Lines 318-329) as follows:
"In the tropics where the LWP is larger than the other areas, the NDW6-simulated LWP is lower and not closer to the MAC results than the NSW6-simulated LWP. Notably, the MAC results contain regional biases of up to 25%, especially in the tropics (Elsaesser et al., 2017), but even with the largest errors, the NDW6- and NSW6-simulated LWPs in the tropics are still underestimated compared to the MAC results. In the horizontal distribution over the eastern Pacific Ocean and Southern Atlantic Ocean at lower latitudes (30°S-0), the NDW6-simulated LWP is lower than the NSW6 results but comparable to the MAC results. However, over the western Pacific Ocean and Indian Ocean at the lower latitudes, both NDW6- and NSW6-simulated LWPs are lower than the MAC results. Therefore, the overestimation of the NSW6-simulated LWP in the eastern Pacific Ocean and Southern Atlantic Ocean effectively balanced the underestimation of the zonal averages of the simulated LWP and unexpectedly led to zonal LWP values closer to the MAC results. This situation also occurs in the northern hemisphere at lower latitudes (30°N-0). Therefore, in the lower latitudes (30°S-30°N), the zonal averages of the NSW6-simulated LWP look closer to the MAC results, but this is attributed to the compensation errors in the regional distribution. As a result, the global and annual mean values of the NSW6-simulated LWP appear closer to the MAC results."

[C3-13] Figure 7: "AOD" is used in the figure labels, but "AOT" is used in the figure caption, tables (e.g., Table 3), and the main text. Please make them all consistent.

[A3-13] We use 'AOT' in this manuscript, so modified Figure 7.

---

## Author Response (AR1)

Reviewer 1

[C-1] Review of Goto et al, 2023

This paper compares results from a single moment cloud microphysics scheme (NSW6 simulations) with those from a double moment scheme (NSW6) in global 6-year 14km resolution simulations. The models are also evaluated using observations and other models.

Both versions perform well against observations of LWP and precipitation in a general sense although a deeper analysis of this evaluation would have been useful – e.g., focussing on certain important areas such as stratocumulus regions that are important for the aerosol-cloud interaction radiative forcing.

It is claimed that NDW6 performs better than NSW6, but I don't think that is justified. It is clear that NSW6 performs better in the tropics and NDW6 at high latitudes. It would be better to be upfront about this and to discuss/examine some of the reasons why this is the case.

Hence, the justification for choosing NDW6 over NSW6 is not that strong from the aerosol results. Perhaps this would be better justified by the likely better representation of cloud microphysical processes in NDW6 (e.g, a better representation of fall speeds, hydrometeor separation, droplet evaporation, etc.?). And the large difference in ERFaci are also a good reason to consider NDW6 - although which one is more accurate is unknown... It might be worth mentioning and considering that the LWP response to aerosol could be constrained/evaluated to some degree with a short simulation of a well-observed volcanic eruption, e.g., like the Holuhraun eruption (Malavelle, Nature, 2017; doi:10.1038/nature22974)

There is a large difference in the aerosol-cloud interaction forcing between the NSW6 and NDW6 simulations, but there is less analysis and discussion devoted to that in the paper. For example, there is no analysis about potential differences in the Twomey effect and cloud lifetime adjustments. E.g., how big might these be in NDW6 vs NSW6? Can you compare the PI to PD CDNC differences for NDW6 to the CCN differences in NSW6? Can you compare changes in the cloud fraction (particularly low altitude clouds) – (this should be possible if the simulations were nudged, otherwise it is likely to be noisy.)

It is highly likely that the difference in the autoconversion scheme between NSW6 and NDW6 is having a large impact on the response of the LWP to aerosols (which is very different between the simulations), but this is not mentioned. It should be mentioned and discussed in the results and summary.

In summary, before publication I think this paper needs a bit of work to add some more relevant analysis and discussion – in particular a fairer discussion of the performance of the single moment scheme vs double moment for aerosols and a more thorough analysis of the differences in aerosol- cloud interaction forcing (ERFaci).

[A-1] We appreciate your great contributions to improve our manuscript. Your comments and suggestions are very helpful and motivate us to investigate the results more deeply. We think these points are included in the specific points, so we would like to answer each point below.
Through the revision, we modified figures and tables as follows:
- Figures 1, 2, 5, E1 and E2: We changed the color of the zonal averages (NDW6 blue, NSW6 orange, as in the other figures).
- Figures 2 and 5: We replotted the model results in white for grids with missing satellite data. We replotted the zonal averages of the model results by eliminating the grids

- with missing satellite data.
- Figure 4: We changed the caption named "references" to "AeroCom".
- Figure 7: We changed the subtitle named "AOD" to "AOT".
- Figure 9: We provided IRFari of all and each aerosol for shortwave & longwave at the TOA & the surface under all & clear sky conditions. We changed the caption named "references" to "Kinne19" and "Thorsen21".
- Figure 10: We removed the results of IRFari because they were shown in Figure 9. Instead, we newly added the results of ERFari and sum of ERFari and ERFaci. We also modified the ERFaci for shortwave to net ERFaci (for both shortwave and longwave).
- Figure 11: We modified ∂AOT to ∂CCN. We added new parameters such as ∂CDNC, ∂CDR, ∂CA, ∂CF, and net ERFaci to further explore ACI.
- Figure 12: We replaced Table 3 in the original manuscript to Figure 12 in the revised manuscript by adding relevant parameters such as ∂CCN, ∂CDNC, ∂CDR, ∂CA, ∂CF, and net ERFaci.
- Figure 13: To explain possible overestimations of the NSW6-simulated Twomey effect, we newly plotted global budgets of the annual averages of the NDW6- and NSW6-simulated Qc (mixing ratio of cloud droplets) and CDNC (cloud droplet number concentrations).
- Table 1 in the original manuscript: We removed it and added a paragraph to explain the HRM and LRM as references in section 2.5 in the revised manuscript.
- Table 1 in the revised manuscript: We simply moved Table 2 in the original manuscript to Table 1 in the revised manuscript.
- Table 2 in the revised manuscript: We showed global and annual mean values of ERFari, ERFaci, and the sum of ERFari and ERFaci for shortwave, longwave, and net radiation under both all-sky and clear-sky conditions.
- Table A1: We newly added the statistical metric to compare results in this study with the references by Goto et al. (2020).
- Table A2: We simply moved Table A1 in the original manuscript to Table A2 in the revised manuscript, with two exceptions. One, we changed "References" to "References from model results". Second, we changed the $SO_2$ production value from 67.5 to 67.7.

Please note that some English was corrected in the revised manuscript.

Specific comments

[C1-2] L126 – "although rain does not directly change cloud water in this case" – It's not clear what you mean here or why this part of the sentence is necessary. Are you saying that there is no scavenging/collection of cloud water by rain? Or that it happens through a calculated microphysical process rate rather than some direct assumption? You'll have to elaborate or decide whether this part of the sentence is needed.

[C1-3] L126 – "Thus, the impact of scavenged aerosols on cloud water is inevitably overestimated in single- moment bulk cloud microphysics schemes" – it's not clear what you mean here either. I can see that there may be an overestimate of the number of droplets present for a given amount of aerosol since droplet numbers cannot evolve and reduce over time in a cloud (partial evaporation, coalescence, etc.) in single moment schemes. What you are saying about scavenging is less clear - are you saying that the impact of aerosol changes is likely to be larger in the single moment scheme due to this direct link between CCN and droplet number? And that this would include a

larger effect due to removal of aerosol by scavenging? This would need rewriting to say that if so.

[A1-2 & A1-3] Thank you for your comment on this expression. Sorry for confusing you. After we deeply considered this point, we decided to remove this sentence from the revised manuscript.

[C1-4] L191 – "sulfate are assumed to have unimodal particle size distributions" – what are the shapes of the size distributions? Lognormal?

[A1-4] Thanks for your comment. To clarify this, we changed the "unimodal particle size distributions" into "lognormal particle size distributions" (Line 181 in the revised manuscript).

[C1-5] L207 onwards – you describe the length of the simulations and some other details here – but this is in the aerosol section, which makes it hard to find. The information on the model run length, spin-up, etc. should be in its own subsection. Also you need some details on whether they were free running or nudged to meteorology?

[A1-5] Thank you for your suggestion. We agree. We newly add section 2.3 "Experimental conditions" to the revised manuscript to explain the experimental design. It includes the information on the model run length (6 years), spin-up (1 month using 1 year test simulation), external datasets (emission, SST, and sea ice) to run the models. We did not nudge the meteorological fields. This is also reflected on section 2.3. Please see section 2.3 in the revised manuscript.

[C1-6] Plus, the details on the satellite data used for model evaluation should be in its own subsection too and not in the aerosol section.

[A1-6] Thank you for your suggestion. Yes, we agree it. We move this part to new section 2.4 "Observations" in the revised manuscript.

[C1-7] L209 – "which are obtained from the end of the 1-year aerosol online simulations coupled to NSW6." – you will need to explain what these "online simulations" are? What do you mean by "online"?

[A1-7] Thank for your comment. To escape this unclear expression, we removed the term "online" from the revised manuscript. It just means that the aerosol results are simulated by the NICAM coupled to NSW6.

[C1-8] L261 – "However, the seasonal and horizontal biases of the NSW6-simulated LWP are effectively cancelled, and the global and annual mean values of the NSW6-simulated LWP appear closer to the MAC results." – I can't see much evidence for the NSW6 LWP values having compensating seasonal and horizontal biases...

[A1-8] Thank you for your comment on the difference in the simulated LWP between NDW6 and NSW6. We had to explain this more detail in the original manuscript.

In the tropics where the LWP is larger than the other areas, the NDW6-simulated LWP is lower and not closer to the MAC results than the NSW6-simulated LWP. However, as you mentioned in C1-9 and Elsaesser et al. (2017), it should be noted that the MAC results contain regional biases of up to 25%, especially in the tropics. Even with the largest errors, both NDW6- and NSW6-simulated LWP in the tropics are still underestimated compared to the MAC results.

In the zonal distribution in the lower latitudes (30S-0), the NDW6-simulated LWP is lower than the NSW6 results but comparable to the MAC results. In the latitudes over the western Pacific Ocean and Indian Ocean, both NDW6- and NSW6-simulated LWP are lower than the MAC results. However, in the eastern Pacific Ocean and Southern Atlantic Ocean, NDW6-simulated LWP are lower than the NSW6 results but comparable to the MAC results. Therefore, the overestimation of the NSW6-simulated LWP in the eastern Pacific Ocean and Southern Atlantic Ocean effectively cancel the underestimation of the zonal averages in the simulated LWP and unexpectedly provides closer LWP values to the MAC results in terms of the zonal averages. This situation also occurs in the northern hemisphere in the lower latitudes (30N-0). Therefore, in the lower latitudes (30S-30N), the zonal averages of the NSW6-simulated LWP looks closer to the MAC results, but this is attributed from the compensation errors in the regional distribution (we removed the compensation errors in the seasonality from the original manuscript).

In summary, we added these explanations to the revised manuscript to support our statement of the compensation errors in the regional distribution in the NSW6-simulated LWP as follows (Lines 318-329 in the revised manuscript):

"In the tropics where the LWP is larger than the other areas, the NDW6-simulated LWP is lower and not closer to the MAC results than the NSW6-simulated LWP. Notably, the MAC results contain regional biases of up to 25%, especially in the tropics (Elsaesser et al., 2017), but even with the largest errors, the NDW6- and NSW6-simulated LWPs in the tropics are still underestimated compared to the MAC results. In the horizontal distribution over the eastern Pacific Ocean and Southern Atlantic Ocean at lower latitudes (30°S-0), the NDW6-simulated LWP is lower than the NSW6 results but comparable to the MAC results. However, over the western Pacific Ocean and Indian Ocean at the lower latitudes, both NDW6- and NSW6-simulated LWPs are lower than the MAC results. Therefore, the overestimation of the NSW6-simulated LWP in the eastern Pacific Ocean and Southern Atlantic Ocean effectively balanced the underestimation of the zonal averages of the simulated LWP and unexpectedly led to zonal LWP values closer to the MAC results. This situation also occurs in the northern hemisphere at lower latitudes (30°N-0). Therefore, in the lower latitudes (30°S-30°N), the zonal averages of the NSW6-simulated LWP look closer to the MAC results, but this is attributed to the compensation errors in the regional distribution. As a result, the global and annual mean values of the NSW6-simulated LWP appear closer to the MAC results."

[C1-9] L262 – "In terms of the distribution pattern and seasonal cycle, the NDW6-simulated LWP is closer to the MAC result compared to the NSW6 result." – I don't see any evidence for this. Although I would be wary of trusting the MAC LWP values too much in precipitating regions like the tropics since precipitation can cause biases.

[A1-9] Thank you for the useful information about possible bias in the MAC LWP. We checked the reference of Elsaesser et al. (2017) again and found that the statistical error of LWP is large in the tropics by at most 25% regionally. But even when we consider this error, the NICAM-simulated LWP over the tropic is underestimated compared to the MAC result. This point is added to the revised version, as we mentioned in A1-8.

[C1-10] L275 – "the results in NDW6 are generally better and closer to those of the real atmosphere." – I'm afraid that I don't think that you can make this statement with the evidence presented. The differences in precipitation (Fig. 1) between NSW6 and NDW6 are very small and it's hard to say which is best when looking at the spatial maps for January and July. Similarly for LWP. Perhaps NDW6 is looking a bit better for July for the higher latitudes, but it's quite hard to tell with the colour range chosen – maybe you need to narrow the colour range on the maps to make the differences clearer? But I think it's clear from the maps and the zonal means that NSW6 does better in the tropics and NDW6 better at higher latitudes even when taking into account the spatial variability and individual months. This would also need addressing at L517.

[A1-10] Thank you for your comment. As you mentioned, the difference in the simulated precipitation between NDW6 and NSW6 is very small. We think the NDW6-simulated LWP is much better than the NSW6-simulated results, but this may be not so clear for readers. Especially, the term "real atmosphere" was removed from our conclusion in the original manuscript because the observation also includes some uncertainty. Therefore, we changed this expression (Lines 340-342 in the revised manuscript) as follows:
"Therefore, the results of precipitation in both NDW6 and NSW6 are comparable to the observations, but those of LWP in NDW6 are different from those in NSW6. The NDW6-simulated LWPs are generally closer to the observations, except for the tropics."
Thank you for your suggestions to change the color range in Figure 2. As you suggested, we changed them in the revised manuscript.

[C1-11] L288 – "the recent models participating in the AeroCom Phase-III project" – are these what you term "references" in Figure 4? This should be explained in the caption. It would also be better to label this as "Aerocom models" or similar in the figure(s). Similarly, for Fig. 9 – you need separate markers and labels for Kinne (2019) and Thorsen (2021) and you need to explain why there are several crosses for all-sky and clear-sky for "All" and "Anthropogenic", but not the other plots in the caption – e.g., are they for different AeroCom model members? Are there also several crosses for Thorsen and what are they if so?

[A1-11] Thank you for your suggestions to clarify the references. We modified these labels and captions. In addition, we newly explained these references in section 2.5 named "reference datasets" of the revised manuscript (Lines 296-302) as follows:
"In addition to the results in Goto et al. (2020) as references for a comparison of global aerosol budgets and aerosol optical properties, results obtained from the AeroCom Phase-III project (Gliß et al., 2021) are used in this study. AeroCom Phase-III includes 14 global models and can be the best reference to evaluate global aerosol simulations. For references of the IRFari, the Max Planck Aerosol Climatology version 2 (MACv2 by Kinne, 2019) provides global maps for aerosol optical and radiative properties by

calculating an offline radiative transfer model with the ensemble mean among the AeroCom global models and the in-situ measurements of AERONET. Another reference for IRFari is the mean value from more than 10 studies based on the observations in Thorsen et al. (2021). The IRFari in Thorsen et al. (2021) is only estimated in the shortwave at the TOA."

[C1-12] L310 – "the NDW6-simulated RPCW is much closer to the CloudSat-retrieved RPCW" – again, I have to disagree here. It is true for the subtropical regions and higher latitudes, but not for the tropics.

[A1-12] Thanks for your comment. We modified this in the revised manuscript (Lines 366-369) as follow:
"Because the NSW6-simulated clouds are larger in most regions except for in the tropics, the NDW6-simulated RPCW is much closer to the CloudSat-retrieved RPCW. In the western Pacific Ocean over the tropics where the simulated aerosols are low, the NSW6 results are closer to the CloudSat results.

[C1-13] L320 – "The difference in the column burden of SO2 between NDW6 and NSW6 is caused by the chemical loss in the aqueous phase (0.7 TgS yr-1 or +2%) and gas phase (1.1 TgS yr-1 or -7%) and wet deposition (0.4 TgS yr-1 or +24%), as shown in Table A1." – these values seem different to those quoted in Table A1?

[A1-13] Thank you for your suggestion. We had some mistakes and corrected them in the revised manuscript (Lines 387-389) as follows:
"The difference in the column burden of $SO_2$ between NDW6 and NSW6 is caused by the chemical loss in the aqueous phase (0.5 TgS yr$^{-1}$ or +1%) and gas phase (-1.3 TgS yr$^{-1}$ or -10%) and wet deposition (0.5 TgS yr$^{-1}$ or +23%), as shown in Table A2."
Because we added a new table (Table A1 in the revised manuscript), Table A1 in the original manuscript was changed to Table A2 in the revised manuscript.

[C1-14] L323 – Need to describe HRM and LRM somewhere in the main paper.

[A1-14] Thank you for your comment on the explanation of HRM and LRM. Yes, we agree. So, we newly added section 2.5 named "reference datasets" to the revised manuscript as follows:
"Our previous model results provided in Goto et al. (2020) using NICAM.16 at a global 14-km high resolution (hereafter referred to as the HRM) and a global 56-km low resolution (hereafter referred to as the LRM) are used as references to compare the NICAM results. As mentioned in section 2.1, the number of vertical layers is set at 38, and the timestep is 1 minute in both the HRM and LRM. The integration periods in both the HRM and LRM are 3 years as climatological runs. The emission inventories, i.e., 2010 for anthropogenic sources, climatological average in 2005-2014 for biomass burning, and natural sources in the present era, and the nudged SST and sea ice in this study are identical to those in both the HRM and LRM, but the initial conditions in this study are different from those in both the HRM and LRM, which use the model results at the end of December after a 1.5-month spin-up. The initial conditions for the model spinup are prepared by the reanalysis datasets of the National Centers for Environmental Prediction (NCEP) Final (FNL) (Kalnay et al., 1996) in November 2011. In the cloud microphysics and autoconversion modules, NDW6 coupled to Seifert and Beheng (2006) and NSW6 coupled to Khairoutdinov and Kogan (2000) are used in this study, whereas NSW6 coupled to Berry (1967) is used in both the HRM and LRM. The improvement in the aerosol module described in section 2.2 is also different from that in the HRM and LRM. The results of the HRM and LRM are useful for evaluating the current model results because the observations are limited in some parameters, such as aerosol global budgets and radiative forcings."

Because we created this section, we removed Table 1 in the original manuscript.

[C1-15] L375 – "example, the IRFari dust values are calculated to be -0.46 Wm-2 (NDW6), -0.57 Wm-2 (NSW6), and -0.24 Wm-2 (Kinne, 2019)." – need to make it clear that these are for all-sky.

[A1-15] Thanks. We added "at the TOA under all-sky conditions" in the revised manuscript.

[C1-16] L375 – it would be good to add a bit of detail about what kind of data Kinne (2019) and Thorsen (2021) represent. I.e., model aerosol reanalysis, satellite observations, etc.

[A1-16] Thank you for your comment on the datasets of Kinne (2019) and Thorsen et al. (2021). According to your suggestion, we add the explanation about these references to section 2.5 named "reference datasets" in the revised manuscript. Kinne (2019) provided global maps for aerosol optical and radiative properties by calculating an offline radiative transfer model with the ensemble mean among the AeroCom global models and the in-situ measurements of AERONET. Thorsen et al. (2021) provided the mean value for the shortwave at the TOA from more than 10 studies based on the observations.

[C1-17] L376 – "This is partly caused by the weaker absorption of AOT in this study compared to the median value of the AeroCom models" – But it also could be due to the higher dust AOT values?

[A1-17] Thank you for your suggestion. Yes, we agree. We added the phrase "and the higher dust AOT" in the sentence (Line 445) in the revised manuscript.

[C1-18] L379 – "This is inconsistent with the results of the larger column burden and AOT of dust in this study compared to those of the AeroCom models" – you should point out that it is consistent with too little SW absorption, though.

[A1-18] Thanks. Yes, we agree. We added the sentence "This is consistent with too little shortwave absorption, but" to the revised manuscript (Line 448).

[C1-19] L388 – "For other nonlight-absorbing components, i.e., sea salt and sulfate, the difference in the IRFari values between all-sky and clear sky conditions is very small" – this doesn't look to be the case? The differences between all-sky and clear-sky are largest for sulphate and sea-salt?

[A1-19] Thanks for your comment on this. We had a mistake and wanted to mention the difference between TOA and surface is the smallest. In the revised manuscript, we modified this in the revised manuscript (Lines 458-459) as follows:
"For other nonlight-absorbing components, i.e., sea salt and sulfate, the difference in the IRFari values between the TOA and the surface is very small."

[C1-20] L390 – Make it clear that this is for all-sky. Same for L396.

[A1-20] Thanks. We added the word "under all-sky conditions".

[C1-21] L451 – "The difference in the ERFaci between NDW6 and NSW6 may be partly explained by a nonlinear relationship of the ERFaci to AOT under the different LWPs, as proposed by Carslaw et al. (2013) who argued that even if the aerosol difference between PI and PD is similar, the value of ERFaci can be larger when the aerosol concentration is lower."
- Presumably you mean here "for different baseline AOT fields"?
- Also "even if the aerosol difference between PI and PD is similar" would be clearer as "even if the PI to PD aerosol difference for two simulations are similar".

[A1-21] Yes. We wanted to point out that the different baseline of AOT fields can provide the difference in the ERFaci between two experiments, even if the difference in the AOT between two experiments is small. We think your second comment is a clear statement. However, after we carefully analyzed this, we concluded that the difference in the baseline of aerosols between NDW6 and NSW6 did not cause the difference in the ERFaci between NDW6 and NSW6. Surely, Carslaw et al. (2013) and Wilcox et al. (2015) pointed out this possibility of the differences in the ERFaci among the experiments. In this study, the different baseline of AOT between NDW6 and NSW6 under the present days can be found at most 20%, so we thought this can be a reason of the difference in the ERFaci between NDW6 and NSW6 in the original manuscript. However, when we looked at CCN, which is more sensitive to the ERFaci, the different baseline of CCN at 1-km height between NDW6 and NSW6 at the preindustrial days was smaller even in Europe where the difference in the ERFaci between NDW6 and NSW6 was the largest among the regions. Therefore, we modified this in the revised manuscript (Lines 583-588) as follows:
"Carslaw et al. (2013) and Wilcox et al. (2015) pointed out that the different baselines of aerosol fields can provide small differences in ERFaci between two simulations. As mentioned in the previous sections for aerosols, the NDW6-simulated aerosols are generally lower than the NSW6 results, for example IRFari is approximately 15% lower. However, the baseline of CCN at 1-km height between NDW6 and NSW6 under the PI conditions is not very different, so the difference in the baseline of aerosols between NDW6 and NSW6 does not cause the difference in ERFaci between the two simulations."
We also added the following sentence to the summary (Lines 645-647) in the revised

manuscript:

"Other possible reason for the differences in the ERFaci between NDW6 and NSW6 is the different baselines of aerosol fields, as suggested by Carslaw et al. (2013) and Wilcox et al. (2015), but this is minor because the baseline of CCN at 1-km height between NDW6 and NSW6 under the PI conditions is not very different."

Based on the above discussion, we removed the related statements from the abstract from the original manuscript.

Reference:

Wilcox, L. J., Highwood, E. J., Booth, B. B. B., and Carslaw, K. S.: Quantifying sources of inter-model diversity in the cloud albedo effect, Geophys. Res. Lett., 42, doi:10.1002/2015GL063301, 2015.

[C1-22] L455 – "Figure 11 shows that the horizontal distribution of changes in the simulated LWP" – it would be good to introduce the LWP adjustment as another potential factor in causing the difference in the ERFaci between NDW6 and NSW6 (since you have previously suggested that the baseline AOT is a potential cause).

But what about potential differences in the Twomey effect and cloud lifetime adjustments? How big might these be in NDW6 vs NSW6? Can you compare the PI to PD CDNC differences for NDW6 to the CCN differences in NSW6? Can you compare changes in the cloud fraction (particularly low altitude clouds) – this should be possible if the simulations were nudged. Otherwise it is likely to be noisy.

[A1-22] Thank you very much for your comments on the difference in the ERFaci between NDW6 and NSW6. According to many suggestions of you and other reviewers, we checked other parameters such as CCN, CDNC, CDR (effective radius of clouds), CF (cloud fraction), CA (cloud albedo), and net ERFaci and largely modified our analysis in the revised manuscript (Lines 526-582) as follows:

"Given the verification of the NICAM-simulated CRF above, the simulated ACI due to anthropogenic aerosols is discussed by comparing the results between NDW6 and NSW6 for simulations with aerosol and precursor gas emissions for the preindustrial (PI), mentioned in section 2.3, and the present day (PD). Figure 11 shows the global maps of changes in the simulated CCN at 1-km heights, cloud droplet number concentrations (CDNC) at 1-km heights only for NDW6, cloud droplet effective radius (CDR) at 1-km heights, LWP, cloud albedo (CA), cloud fraction (
[revised manuscript text omitted]
 at 1-km height) change (∂CDR), LWP change (∂LWP), CA (cloud albedo) change (∂CA), CF (cloud fraction at 1-km height) change (∂CF), and net ERFaci by comparing the results between NDW6 and NSW6 for simulations with aerosol and precursor gas emissions for the present and the preindustrial era. The number located in the upper right in each panel represents the global and annual mean value. The results at 1-km height also include areas with elevations higher than 1-km height in white.

[Figure]

Figure 12: Regional averages of the differences in CCN at 1-km height, CDNC (cloud droplet number concentration only in NDW6), CDR (cloud droplet effective radius at 1-km height), LWP, CA (cloud albedo), CF (cloud fraction at a 1-km height), and net ERFaci between the preindustrial and the present days. The regions are defined as US (90°W-60°W, 30°N-50°N), Europe (0°E-30°E, 40°N-60°N), East Asia (110°E-140°E, 20°N-50°N), and India (70°E-90°E, 10°N-35°N).

[Figure]

Figure 13: Global budgets of the annual averages of the NDW6- and NSW6-simulated Qc (mixing ratio of cloud droplets), the NDW6-simulated CDNC (cloud droplet number concentration), and the NSW6-simulated CDNC (cloud droplet number concentration, which is equal to CCN number concentrations)

[C1-23] L466 – "Notably, the reason for the differences in the susceptibility between NDW6 and NSW6 should be addressed in future studies."

- Although it is clear that the difference in the autoconversion scheme between NSW6 and NDW6 is likely to have a large impact here and so this should be mentioned and discussed here and in the summary too.

[A1-23] Thank you for your comment on the autoconversion scheme. Theoretically, the dependence of the autoconversion on CDNC in Seifert and Beheng (2006) (hereafter referred to as SB06) is larger than in Khairoutdinov and Kogan (2000) (hereafter referred to as KK00), so for the same ∂CDNC, SB06 has a lower cloud-to-rain conversion efficiency (clouds in SB06 tend to remain). Because NDW6 uses SB06 and NSW6 uses KK00, if we look only at the autoconversion, NDW6 has a larger ∂LWP than NSW6. This tendency is consistent with the results of the difference in ∂LWP between NDW6 and NSW6, as shown in Figure 11 in this study, but the magnitude of the dependence on CDNC is not very large between SB06 and KK00. Therefore, we think that the difference in the autoconversion between SB06 and KK00 alone cannot explain the difference in the ∂LWP between NDW6 and NSW6 in this study.

Ideally, we should do sensitivity experiments for different autoconversion schemes, but we cannot do them due to the limitation of available computer resources in our environment. So, we would like to discuss possible differences in the ERFaci among different autoconversion schemes by using a reference. Michibata and Suzuki (2020) shows the difference in the ERFaci between various autoconversion schemes on the MIROC climate model. This version of MIROC adopts a double-moment cloud microphysics module including prognostic precipitation. The sensitivity experiments in the MIROC used various autoconversion including KK00 and SB06. Therefore, Michibata and Suzuki (2020) can be a suitable reference here. Even though Michibata and Suzuki (2020) did not deeply discuss the difference in the ERFaci among the different autoconversion schemes, the difference in the global and annual mean values of ERFaci between KK00 and SB06 can be estimated to be 0.15 Wm$^{-2}$. The magnitude of the ERFaci with SB06 is smaller than that with KK00, although the dependence of the autoconversion on CDNC in SB06 is larger than in KK00. This suggests that the difference in the LWP response to aerosol between SB06 and KK00 is not large and that the difference in the ERFaci may have opposite signs through processes other than autoconversion. Therefore, we added the following sentences to the revised manuscript (Lines 589-594):

"The difference in the autoconversion from clouds to precipitation between NDW6 and NSW6 can be a reason for the difference in ERFaci between NDW6 and NSW6. Using a global aerosol model, MIROC, coupled to a double-moment bulk cloud microphysics scheme with coarse resolution of 1.4° × 1.4°, the difference in ERFaci between Khairoutdinov and Kogan (2000) and Seifert and Beheng (2006) is estimated to be 0.15 Wm$^{-2}$ (Michibata and Suzuki, 2020). This magnitude of ERFaci difference potentially caused by the two different autoconversion schemes cannot explain the difference in ERFaci between NDW6 and NSW6 of this study."

We also added the following comments to the summary (Lines 647-652) in the revised manuscript:

"Another possible reason is the difference in the autoconversion between NDW6 and NSW6, and the difference in ERFaci between Khairoutdinov and Kogan (2000) and Seifert and Beheng (2006) is estimated to be 0.15 Wm$^{-2}$ by using a general circulation

model MIROC (Michibata and Suzuki, 2020). However, this magnitude of ERFaci difference potentially caused by the two different autoconversion schemes cannot explain the difference in ERFaci between NDW6 and NSW6 of this study."

Reference:
Michibata, T., and Suzuki, K.: Reconciling compensating errors between precipitation constraints and the energy budget in a climate model, Geophys Res. Lett., 47, e2020GL088340, doi:10.1029/2020GL088340, 2020.

[C1-24] L496 – "The differences in the dust emissions, dust column burden and SO2, AOT, and IRFari values for total aerosols between NDW6 and NSW6 are larger." – at L480 you said that the AOT differences were small, so this is a bit of a contradiction.

[A1-24] Thanks for your comment on this remark. We added the term "than those in the other aerosol budgets and components" in the revised manuscript (Line 623).

[C1-25] L508 – "These differences are mainly caused by the difference in the susceptibility of the LWP to AOT" – in the results section this wasn't stated so clearly. It would be good to mention that in the results too – although do you have evidence that this is the case?

[A1-25] Thank you for your comment. In A1-22, we drastically modified the explanation about the ERFaci by showing the relevant parameters, such as CCN, CDNC, CDR and LWP. One of the important conclusions is that the ERFaci due to the cloud lifetime effect in NDW6 was larger than that in NSW6 due to the Twomey effect. The difference in the cloud lifetime effect between NDW6 and NSW6 is possibly caused by the difference in the treatment of cloud droplet and raindrop number concentrations. Unlike NSW6, NDW6 predicts both CDNC and raindrop number concentration (RDNC). This generates different variabilities of CDNC and RDNC between the two schemes, possibly leading to the different susceptibilities. Nevertheless, more detailed analysis will be required in future studies to explore microphysical processes responsible for these different behaviors between the two schemes.

Another important conclusion is that the NSW6-simulated ERFaci certainly includes some bias due to the overestimation of the Twomey effect. This is clearly shown in Figure 13, which shows that the vertical profile of $\partial$CDNC in NSW6 is different from that of $\partial$Qc because NSW6 cannot predict CDNC and adopts $\partial$CCN. Therefore, we modified this sentence in the summary (Lines 637-645) in the revised manuscript as follows:

"These differences are mainly caused by the difference in the susceptibility of the $\partial$LWP to $\partial$CCN. As discussed in section 4.2, it was suggested the increase in changes in ERFaci due to the cloud lifetime effect in NDW6 is larger than that in NSW6 due to the Twomey effect, although the NSW6-simulated ERFaci certainly includes some bias due to the overestimation of the Twomey effect. The different susceptibility between NDW6 and NSW6 could be interpreted in terms of different complexities of hydrometeors interactions between NSW6 and NDW6, particularly whether the CDNC and RDNC are predicted. This generates different variabilities of CDNC and RDNC between the two schemes, possibly leading to the different susceptibilities. Nevertheless, more detailed analysis will be required in future studies to explore microphysical processes responsible for these different behaviors between the two schemes.

[C1-26] L518 – "and thus, the use of NDW6 is recommended in environmental and climate simulations." - The justification regarding the LWP and precipitation performance here is not very strong (see above regarding the NSW6 vs NDW6 performance). Perhaps this would be better justified by the likely better representation of cloud microphysical processes in NDW6 (e.g, a better representation of fall speeds, hydrometeor separation, droplet evaporation, etc.?). And the large difference in ERFaci are also a good reason to consider NDW6 - although which one is more accurate is unknown... It might be worth mentioning and considering that the LWP response to aerosol could be constrained/evaluated to some degree with a short simulation of a well-observed volcanic eruption, e.g., like the Holuhraun eruption (Malavelle, Nature, 2017; doi:10.1038/nature22974)

[A1-26] Thank you for your comment on the recommendation to use the NDW6 scheme. As you said that "this would be better justified by the likely better representation of cloud microphysical processes in NDW6", we think this should be reflected in the revised manuscript. Our answer using Figures 11,12, and 13 in the revised manuscript showed that the differences in the $\partial$CCN and $\partial$CDNC between NDW6 and NSW6 were large, and thus the differences in the $\partial$CDR between NDW6 and NSW6 were large. Therefore, from this point as well, it is considered that the reliability of the calculation results using NDW6 is high, and in the revised manuscript (Lines 667-669), we have corrected this as follows:
"The cloud microphysics representation of NDW6 is more elaborate than that of NSW6, and it found that the NSW6-simulated CDR is overestimated due to the inability to predict CDNC in NSW6. Therefore, the use of NDW6 is recommended in environmental and climate simulations".
And because we found a clear bias of the results in NSW6, we also removed the following part from the end of the paragraph in the revised manuscript: "However, because simulations using NDW6 require 1.5 times more calculation resources, the use of NSW6 is still useful for long-period climate simulations at high resolutions."
Thank you for your suggestion to check the literature on numerical experiments for the Holuhraun eruption. It can be a reference to justify the response of aerosols to clouds. According to Malavelle et al. (2017), the CDR decreases but the LWP doesn't clearly increase or decrease when the CCNs increases. In the revised Figure 11, it clearly shows that the NDW6- and NSW6-simulated CDRs decrease and the NDW6- and NSW6-simulated CDRs increase when the CCN increase. The CDR response to aerosol seems to be consistent to the results in Malavelle et al. (2017), but the LWP response to aerosol seems to be different from the results in Malavelle et al. (2017). Even if we perform the volcanic eruption experiment, we can only evaluate a specific case. In addition, this area (60-70N) and this period (September-October) may be affected not only by interactions between aerosols and clouds, but also by interactions between cloud water and cloud ice. As you mentioned "to some degree", we agree this point and think that it is still difficult to evaluate the LWP response to aerosols in this study. Therefore, although we did not perform this extra experiment in this study, we would like to do so in future studies to investigate the LWP response to aerosol in one case. Therefore, we added this following sentence to the end of conclusion in the revised manuscript (Lines 669-671):
"At the same time, because the ERFaci in NDW6 needs validation, in the future it will be necessary to perform additional experiments targeting specific cases of volcano in

Iceland shown in Malavelle et al. (2017) to deeply evaluate the model results in NDW6."

[C1-27] L529-531 – I can't see the PCC, etc. values from Goto (2020) listed anywhere – these should be included or quoted in this paper somewhere – ideally in a table so that the reader can compare the new simulations to the old ones.

[A1-27] Thank you for your comment on these values and suggestion. To clarify them, we added a new table (Table A1 in the revised manuscript) to summarize the statistical metrics. At least, these values are our mistake, so they are modified in the revised manuscript (Line 678) as follows:
"(e.g., PCC of 0.819, RMSE of 5.03, and NMB of -54.8 $\mu$g m$^{-3}$ from Goto et al., 2020)".

Table A1 Statistical metrics of PCC, RMSE, and NMB for the annual averages of surface aerosol mass concentrations (OM, BC, and sulfate) between in situ measurements and the NICAM simulations (NDW6 and NSW6 in this study shown in the panels of Figure 3 and HRM and LRM in Figure 8 in Goto et al., 2020).

| | NDW6 | NSW6 | HRM | LRM |
|---|---|---|---|---|
| OM | | | | |
| PCC | 0.847 | 0.846 | 0.819 | 0.794 |
| RMSE [$\mu$g m$^{-3}$] | 3.40 | 3.34 | 5.03 | 5.21 |
| NMB [%] | -30.4 | -25.8 | -54.8 | -56.1 |
| BC | | | | |
| PCC | 0.904 | 0.904 | 0.890 | 0.869 |
| RMSE [$\mu$g m$^{-3}$] | 1.05 | 1.03 | 1.16 | 1.28 |
| NMB [%] | -53.4 | -51.3 | -46.4 | -52.3 |
| Sulfate | | | | |
| PCC | 0.807 | 0.853 | 0.815 | 0.768 |
| RMSE [$\mu$g m$^{-3}$] | 3.97 | 3.67 | 3.94 | 4.34 |
| NMB [%] | -10.4 | -3.7 | -14.6 | -23.7 |

[C1-28] Table A1 – it would be useful to say what kind of observations the difference references use (satellite, model reanalysis, models, etc.).

[A1-28] Thanks for your comment. They are all model results. We modified "References from model results" in Table A2 of the revised manuscript.

[C1-29] L554 – "whereas the difference in the cloud microphysics module does not affect the chemical budget of SO2 oxidation" – presumably you are talking about the cloud microphysical changes within the NSW6 module here rather than the difference between double and single moment (since there are large differences for SO2 between NSW6 and NDW6)? You should make this clearer if so.

[A1-29] Thank you very much for reading the appendix carefully. We had a mistake. This part (comment on the difference among the different cloud microphysics modules) should not be mentioned in this section, because this part focused on NSW6, HRM, and LRM using NSW6. We removed this in the revised manuscript.

[C1-30]

L18 – "but some aerosol species, especially dust and sulfate, have larger differences among the experiments with NDW6 and NSW6 compared to those among the experiments with different horizontal resolutions, i.e., 14 km and 56 km grid spacing,..."
- Would be better as "but differences between the NDW6 and NSW6 experiments are larger for some aerosol species, especially dust and sulfate, compared to those between experiments with different horizontal resolutions, i.e., 14 km and 56 km grid spacings,..."

L48 – "and hence, an elaboration of both the cloud module and aerosol physics module is required to improve ACI in climate models". "Elaboration" is perhaps not the best choice of words. I recommend "evaluation" instead.

L52 – "It is promising that convective cloud systems are better represented with finer model resolution when cumulus parameterizations are avoided" – I would recommend this instead: "These results suggest that convective cloud systems are better represented with a finer model resolution for which cumulus parameterizations are avoided"

L72 – "solved" -> "resolved"

L88 – "calculcated" -> "run"

L96 - "incorporated to" -> "incorporated into"

L97 – "were reflected to the version in NICAM.19" -> "was incorporated into the version in NICAM.19"

L107 – "to aerosol physics module" -> "to the aerosol physics module"

L157 – "concentration higher" -> "concentration is higher"

L167 – "dependence of leaf area index" – should be "dependence on leaf area index"?

L175 – "OM" has not been defined.

L176 – "Secondary organic aerosols (SOAs) are assumed to be particles by multiplying the emission fluxes of isoprene and terpenes provided by" -> "Secondary organic aerosols (SOAs) are assumed to form particles, which are calculated by multiplying the emission fluxes of isoprene and terpenes provided by" – also, what are they multiplied by? A constant factor?

L178 – "Parts of SO2 are emitted from volcanic eruptions (Diehl et al., 2012) and are formed from DMS," – "SO2 is emitted from volcanic eruptions (Diehl et al., 2012) and is also formed from DMS,"

L217 – "These identical datasets were prepared and used in Goto et al. (2020)" – I assume you mean "The same datasets were prepared and used in Goto et al. (2020)". Otherwise it makes it sound like all of the observational datasets are identical.

L249 – "by satellite results" -> "using satellite data".

L268 – "CF at the low level" -> "low-altitude CF"

L306 – need to define what WSBC and WIBC are somewhere.

L394 – "which can be caused by its lower lifetime among the references" -> "which may be due to its short lifetime relative to the values from Kinne (2019) (and Thorsen?)".

L395 – "sea salt is more scavenged by wet deposition" -> "sea salt is scavenged more by wet deposition"

L418 – "but comparable to" -> "but are comparable to"

L422 – "are smaller than"-> "are smaller in magnitude than"

L427 – "fluxes are compared for model evaluations of radiation budget." -> "fluxes are compared and evaluated."

L427 – "the global and January averages of the SWCRF" -> "the global averages of the SWCRF for January". Similarly, for L428 for July and L436 for global averages.

L442 – "by comparing the results between NDW6 and NSW6 under the preindustrial (PI) and the present day (PD)" – better to be clear that this means PI and PD emissions for aerosols and precursor gases. E.g., "by comparing the results between NDW6 and NSW6 for simulations with preindustrial (PI) and the present day (PD) aerosol and precursor gas emissions". Similarly for the caption of Fig. 11.

L457 – "such as the United States, Europe, and East Asia, increase" -> "such as the United States, Europe, and East Asia, increase in magnitude"

L457 – "the NDW6-estimated ERFaci value is larger negatively than the NSW6-estimated ERFaci" -> "the negative NDW6-estimated ERFaci values are larger in magnitude than the NSW6-estimated ERFaci values"

L460 – "key to understand the difference" -> "key to understanding the difference"

L502 – "whereas those in the sulfate are mainly caused by the wet deposition of SO2." -> "whereas those in the sulfate are mainly caused by the differences in the wet deposition of SO2."

L530 – "or" -> "of"

L538 – "Surely," – wrong choice of phrase here. Would be better as "In support of this,"

Figure E1 – the caption has become jumbled.

[A1-30] Thank you very much for reading and checking the details in our manuscript and giving your corrections. We reflected them in the revised manuscript.

Reviewer 2

[C2-1] This study investigates the impact of newly implemented 2-moment cloud microphysics scheme on the simulated aerosols and their interactions with radiation and clouds in a global model at 14-km resolution. The authors find that with the new scheme the simulated aerosol burden is overall decreased, which is (said) mainly due to a faster cloud to precipitation conversion (as suggested by the increased RPCW ratio). Consequently, the direct effects of all aerosols (natural + anthropogenic) are reduced. On the other hand, the indirect effect (forcing caused by aerosol-cloud interactions) of anthropogenic aerosols is greatly increased (about doubled). The authors state that there are two possible reasons: 1) the cloud water adjustment changes; 2) the lower background aerosol AOT (burden, CCN).
Evaluating the impact of cloud microphysics change on the aerosol lifecycle and aerosol-cloud interactions is important for global aerosol-climate model development, especially for high-resolution applications. Results from this study will serve as a reference for model development and help the modeling community to better understand the behavior of this model. Therefore, I think this study fits the scope of GMD well and it could be a useful reference. However, I think the current manuscript needs to be significantly improved, especially in evaluating the simulated cloud microphysics responses to aerosol perturbations and in explaining the differences in the simulated aerosol indirect effects.

[A2-1] We appreciate your great contributions to improve our manuscript. Your comments and suggestions are very helpful and motivate us to investigate the results more deeply. The evaluation about the simulated cloud microphysics responses to aerosol perturbations and explanation about the differences in the simulated AIE were greatly modified to the revised manuscript.
Through the revision, we modified figures and tables as follows:
- Figures 1, 2, 5, E1 and E2: We changed the color of the zonal averages (NDW6 blue, NSW6 orange, as in the other figures).
- Figures 2 and 5: We replotted the model results in white for grids with missing satellite data. We replotted the zonal averages of the model results by eliminating the grids with missing satellite data.
- Figure 4: We changed the caption named "references" to "AeroCom".
- Figure 7: We changed the subtitle named "AOD" to "AOT".
- Figure 9: We provided IRFari of all and each aerosol for shortwave & longwave at the TOA & the surface under all & clear sky conditions. We changed the caption named "references" to "Kinne19" and "Thorsen21".
- Figure 10: We removed the results of IRFari because they were shown in Figure 9. Instead, we newly added the results of ERFari and sum of ERFari and ERFaci. We also modified the ERFaci for shortwave to net ERFaci (for both shortwave and longwave).
- Figure 11: We modified $\partial AOT$ to $\partial CCN$. We added new parameters such as $\partial CDNC$, $\partial CDR$, $\partial CA$, $\partial CF$, and net ERFaci to further explore ACI.
- Figure 12: We replaced Table 3 in the original manuscript to Figure 12 in the revised manuscript by adding relevant parameters such as $\partial CCN$, $\partial CDNC$, $\partial CDR$, $\partial CA$, $\partial CF$, and net ERFaci.
- Figure 13: To explain possible overestimations of the NSW6-simulated Twomey effect, we newly plotted global budgets of the annual averages of the NDW6- and NSW6-

simulated Qc (mixing ratio of cloud droplets) and CDNC (cloud droplet number concentrations).

- Table 1 in the original manuscript: We removed it and added a paragraph to explain the HRM and LRM as references in section 2.5 in the revised manuscript.
- Table 1 in the revised manuscript: We simply moved Table 2 in the original manuscript to Table 1 in the revised manuscript.
- Table 2 in the revised manuscript: We showed global and annual mean values of ERFari, ERFaci, and the sum of ERFari and ERFaci for shortwave, longwave, and net radiation under both all-sky and clear-sky conditions.
- Table A1: We newly added the statistical metric to compare results in this study with the references by Goto et al. (2020).
- Table A2: We simply moved Table A1 in the original manuscript to Table A2 in the revised manuscript, with two exceptions. One, we changed "References" to "References from model results". Second, we changed the $SO_2$ production value from 67.5 to 67.7.

Please note that some English was corrected in the revised manuscript.

**Major comments:**
[C2-2] 1. Since the focus of this study is on the impact of cloud microphysics on aerosol simulation. It's vital to show the cloud microphysics property changes in the simulation. I would recommend the authors to check the cloud water mass and number budgets in the simulations and evaluate the impact of aerosol perturbation on the budget changes. A good example is shown in Salzmann et al. (2010) in ACP.

[A2-2] Thank you for your comments and providing a good example for comprehensive analysis of the budget changes. But unfortunately, we didn't output the budgets of cloud water mass and number like Salzmann et al. (2010), and due to our computer resource limitations, we are unable to recalculate them. We would like to calculate these budgets and analyze them carefully in the future study. Instead, we calculated differences in the cloud water mass (mixing ratio) and number budgets between the PD and PI aerosol conditions to see the impacts of aerosol perturbation on cloud budget changes. Their global averages in the model heights were plotted in Figure 13 in the revised manuscript. We think this figure clearly indicates a possible overestimation of the cloud water number concentrations (CDNC) in NSW6 above 3 km, because the CDNC sink processes such as accretion, auto-conversion, and evaporation are not considered in NSW6. The following discussions were added to the revised manuscript (Lines 543-556):
"To evaluate the Twomey effect in NDW6 and NSW6, the global averages of differences in the mixing ratios and number concentrations for clouds between the PD and PI aerosol conditions are plotted in Figure 13. The changes in $\partial Qc$ in both NDW6 and NSW6 are positive at most heights, so Qc increases as aerosols increase. This is consistent with the results of $\partial LWP$ shown in Figures 11 and 12(e). The largest value of $\partial Qc$ in both NDW6 and NSW6 occurs at a height of approximately 1.5 km, but the largest values in NDW6 are distributed up to a height of 2 km. Above a height of 3 km, $\partial Qc$ in NDW6 is positive, whereas $\partial Qc$ in NSQ6 is close to zero or negative. This difference in $\partial Qc$ between NDW6 and NSW6 is possibly caused by the differences in the simulated supercooled liquid water in mixed-phase clouds, as mentioned in section 3.1. For $\partial CDNC$, the largest values in NDW6 occur at a height of 1.2 km, which is slightly lower than the height where the largest value of $\partial Qc$ occurs. This reflects the vertical structure of typical

clouds in NDW6. In contrast, the vertical profile of ∂CDNC in NSW6 is different from that of ∂Qc because NSW6 cannot predict CDNC and adopts ∂CCN. This implies that ∂CDR is anti-proportional to ∂Qc from the surface to the 4-km height and has a low value below the 1.5-km height and the largest value at a height of approximately 1.5 km. Specifically, above a height of 3 km, where ∂Qc is close to zero and ∂CDNC has a positive value, ∂CDR should be small. The possible overestimation of ∂CDR in NSW6 represents possible overestimation of the Twomey effect in NSW6."

[Figure]

Figure (Figure 13 in the revised manuscript): Global budgets of the annual averages of the NDW6- and NSW6-simulated Qc (mixing ratio of cloud droplets), the NDW6-simulated CDNC (cloud droplet number concentration), and the NSW6-simulated CDNC (cloud droplet number concentration, which is equal to CCN number concentrations)

[C2-3] 2. The authors emphasized the liquid water adjustment (2nd indirect effect) in the abstract and summary. What is the impact of the Twomey effect in the model? The cloud droplet number changes (PD vs PD and PD-PI vs. PD-PI between the simulations) and the impact on effective radius and cloud albedo should be evaluated.

[A2-3] Thank you very much for your comments on the difference in the ERFaci between NDW6 and NSW6. According to many suggestions of you and other reviewers, we checked other parameters, such as CCN, CDNC, CDR (effective radius of clouds), CF (cloud fraction), CA (cloud albedo), and net ERFaci and largely modified our analysis in the revised manuscript (Lines 526-542) as follows:
"Given the verification of the NICAM-simulated CRF above, the simulated ACI due to anthropogenic aerosols is discussed by comparing the results between NDW6 and NSW6 for simulations with aerosol and precursor gas emissions for the preindustrial (PI), mentioned in section 2.3, and the present day (PD). Figure 11 shows the global maps of changes in the simulated CCN at 1-km heights, cloud droplet number concentrations (CDNC) at 1-km heights only for NDW6, cloud droplet effective radius (CDR) at 1-km heights, LWP, cloud albedo (CA), cloud fraction (CF) at 1-km height and net ERFaci between PD and PI. Figure 12 also shows the average values of the selected regions. These figures show that the global average of the NDW6-calculated ∂CCN at a 1-km

height is estimated to be 16.70 cm$^{-3}$ ($\partial$CCN), whereas that in NSW6 is estimated to be 19.59 cm$^{-3}$ ($\partial$CCN). The NDW6-calculated $\partial$CCN values are lower than the NDW6 results. In $\partial$CDNC, the NDW6-estimated values are +0.70 cm$^{-3}$ (global), +4.22 cm$^{-3}$ (the United States), +4.58 cm$^{-3}$ (Europe), +3.57 cm$^{-3}$ (East Asia), and +0.34 cm$^{-3}$ (India). However, the CDNC used in NSW6 is equal to the CCN concentrations due to the ignorance of sink process in the CDNC in NSW6, as mentioned in section 2.1, so the difference in $\partial$CDNC between NDW6 and NSW6 is very large. As a result, the NSW6-simulated $\partial$CDR values at the 1-km height are much larger than the NDW6 results. The NDW6-estimated $\partial$CDR is -0.17 $\mu$m (global), -0.64 $\mu$m (the United States), -0.55 $\mu$m (Europe), -0.91 $\mu$m (East Asia), and -0.33 $\mu$m (India), whereas the NSW6-estimated $\partial$CDR is -0.34 $\mu$m (global), -0.93 $\mu$m (the United States), -0.91 $\mu$m (Europe), -1.20 $\mu$m (East Asia), and -0.81 $\mu$m (India). As shown in Figure 11, the NDW6- and NSW6-estimated $\partial$CDR values are negative near the industrial regions where the $\partial$CCN is large. Therefore, the approximately 15% difference in $\partial$CCN between NDW6 and NSW6 causes the approximately 50% difference in $\partial$CDR. This indicates that the Twomey effect, i.e., the response of $\partial$CDR to $\partial$CCN, in NSW6 is larger than that in NDW6."

In addition to the above discussion about $\partial$CCN, $\partial$CDNC, and $\partial$CDR, we discussed $\partial$LWP, $\partial$CA, $\partial$CF, and ERFaci in the revised manuscript (Lines 557-582) as follows:

[revised manuscript text omitted]

[C2-4] 3. It is unclear to me why the authors include the comparison against HRM and LRM (from the other study) and why the discussions are only for some of the fields, but not the others. If the authors want to include this part, the title should be revised (to reflect the impact of resolution and time stepping changes).

[A2-4] Thank you for your comment on the use of HRM and LRM in this study. We treated these results (HRM and LRM) as references to evaluate the model results of NDW6 and NSW6 in this study. In fact, the main target in this study is comparisons between NDW6 and NSW6 in the text, and as a supporting information, comparisons between NSW6, HRM and LRM were mentioned in Appendix. To clarify this, we added a section 2.5 "Reference datasets" in the revised manuscript (Lines 281-302) to explain the information about HRM and LRM as references.

"Our previous model results provided in Goto et al. (2020) using NICAM.16 at a global 14-km high resolution (hereafter referred to as the HRM) and a global 56-km low

resolution (hereafter referred to as the LRM) are used as references to compare the NICAM results. As mentioned in section 2.1, the number of vertical layers is set at 38, and the timestep is 1 minute in both the HRM and LRM. The integration periods in both the HRM and LRM are 3 years as climatological runs. The emission inventories, i.e., 2010 for anthropogenic sources, climatological average in 2005-2014 for biomass burning, and natural sources in the present era, and the nudged SST and sea ice in this study are identical to those in both the HRM and LRM, but the initial conditions in this study are different from those in both the HRM and LRM, which use the model results at the end of December after a 1.5-month spin-up. The initial conditions for the model spin-up are prepared by the reanalysis datasets of the National Centers for Environmental Prediction (NCEP) Final (FNL) (Kalnay et al., 1996) in November 2011. In the cloud microphysics and autoconversion modules, NDW6 coupled to Seifert and Beheng (2006) and NSW6 coupled to Khairoutdinov and Kogan (2000) are used in this study, whereas NSW6 coupled to Berry (1967) is used in both the HRM and LRM. The improvement in the aerosol module described in section 2.2 is also different from that in the HRM and LRM. The results of the HRM and LRM are useful for evaluating the current model results because the observations are limited in some parameters, such as aerosol global budgets and radiative forcings."

To evaluate our results in this study, the results of HRM and LRM are very useful, because the observation and other morel references are limited in some parameters. Therefore, we don't think we need to change the title by reflecting the impact of resolution and time stepping changes.

**Detailed comments:**
[C2-5] Page 1, Line 21: It would be useful to report the net effective aerosol forcing and ERFari (Ghan's method) as well.

[A2-5] Thank you for your comment on the ERF. According to your suggestions, we added extra forcings (ERFari, ERFaci, and ERFari plus ERFaci) in Figure 10 in the revised manuscript. We showed net ERFari values in abstract (Lines 21-24) and net ERFaci values (Lines 29-31) in the revised manuscript.

[Figure]

Figure 10 in the revised manuscript: Global and annual mean values of (a) effective radiative forcing for anthropogenic aerosol-radiation interaction (ERFari) for shortwave

and net (sum of shortwave and longwave) radiation, (b) ERFaci for anthropogenic aerosol-cloud interaction, and (c) the net ERF (sum of ERFari and ERFaci). All units are in W m$^{-2}$. In ERFari, the reference of Forster21 is estimated in the net radiation by IPCC-AR6 or Forster et al. (2021), whereas the reference of Thortsen21 is estimated in the shortwave radiation by Thorsen et al. (2021). The reference for Smith20 is Smith et al. (2020). The values are also listed in Table 2.

[C2-6] Page 1, Line 23: e.g., -> i.e.,

[A2-6] Corrected.

[C2-7] Page 1, Line 31: Why is the 2nd indirect effect (LWP adjustment) so important? How about the Twomey effect in this model?

[A2-7] The details are mentioned in A2-3, so here our modification in Abstract is shown. We modified this in the revised manuscript (Lines 31-34) as follows:
"The magnitude of the ERFaci value in the NDW6 experiment is larger than that in the NSW6 result due to the differences in the susceptibility of the simulated cloud water to the simulated aerosols between NDW6 and NSW6 and the overestimation of the Twomey effect in NSW6 caused by ignorance of sink process in the cloud droplet number concentrations."

[C2-8] Page 2, Line 32: It would be better to look at the ERFaci vs. CCN relationship, rather than ERFaci vs. AOT.

[A2-8] Thanks for your suggestion. We replaced AOD to CCN in the analysis of ACI in the revised Figures 11 and 12.

[C2-9] Page 2, Line 37: better change "aerosol nucleation" to "aerosol activation" to avoid confusion.

[A2-9] Thanks for your correction. We agree.

[C2-10] Page 3, Line 68-69: "difference in the simulated aerosol mass concentrations" Do you mean surface concentrations or mass burden?

[A2-10] Surface concentrations. We added "at the surface" to the revised manuscript (L72).

[C2-11] Page 3, Line 72: Please provide the reference.

[A2-11] Thanks for your comment. This past research is Goto et al. (2020) in the revised manuscript (Line 75).

[C2-12] Page 3, Line 93: Are they climatological runs, or AMIP-style simulations with transient prescribed SST? How is the model initialized?

[A2-12] Thank you for the comments. The experimental conditions were a bit unclear in the original manuscript. The simulations were climatological fields. All the experiments with both NDW6 and NSW6 were carried out for 6-years after the 1-month spinup calculation. The initial conditions for the model spinup were obtained from the end of the 1-year aerosol simulations coupled to NSW6 without nudging the meteorological fields under the present era. We added Section 2.3 named "Experimental conditions" to the revised manuscript (Lines 200-227).

[C2-13] Page 4, Line 109-110: Water vapor is not a hydrometeor.

[A2-13] Thank you for your correction. Yes, it is. We changed this term "hydrometeor" to "water substances" in the revised manuscript of two parts (Lines 112 and 119). We also changed "hydrometeors, except for water vapor" to "hydrometeors" by removing "except for water vapor" in the revised manuscript (Line 156).

[C2-14] Page 4, Line 112: How is the updraft velocity calculated in the model?

[A2-14] Even in 14-km simulations, the updraft velocity is needed to be parameterized using Lohmann et al. (1999). In the revised manuscript (Lines 115-116), we added the following sentence:
"This parameterization is a function of the parameterized updraft velocity with turbulent kinetic energy (Lohmann et al., 1999), aerosol sizes, and aerosol chemical composition".


| Anthropogenic Source | 1850 | 2010 | 2010-1850 [%] |
|---|---|---|---|
| $SO_2$ [$ktSO_2$] | 2,481 | 115,487 | 2.1 |
| BC [ktC] | 934 | 7,755 | 12.0 |
| OC [ktC] | 4,262 | 18,755 | 22.7 |


The comment of "what's the implication of such results for the use of double-moment microphysics in high-resolution climate models?" is also important point. The high-resolution is certainly important to simulate aerosols and clouds more realistically, as shown in this study and our previous study (Goto et al., 2020). The advantage of using double-moment cloud microphysics scheme is that the cloud microphysics representation of double-moment microphysics (NDW6) is more elaborate than that of single-moment microphysics (NSW6). In addition, this study showed that the NSW6-simulated cloud droplet radii for water clouds (CDR) was overestimated due to the inability to predict cloud droplet number concentrations in NSW6. These points were discussed in section 4.2 and mentioned in section 5 in the revised manuscript.

Through the revision, we modified figures and tables as follows:

- Figures 1, 2, 5, E1 and E2: We changed the color of the zonal averages (NDW6 blue, NSW6 orange, as in the other figures).
- Figures 2 and 5: We replotted the model results in white for grids with missing satellite data. We replotted the zonal averages of the model results by eliminating the grids with missing satellite data.
- Figure 4: We changed the caption named "references" to "AeroCom".
- Figure 7: We changed the subtitle named "AOD" to "AOT".
- Figure 9: We provided IRFari of all and each aerosol for shortwave & longwave at the TOA & the surface under all & clear sky conditions. We changed the caption named "references" to "Kinne19" and "Thorsen21".
- Figure 10: We removed the results of IRFari because they were shown in Figure 9. Instead, we newly added the results of ERFari and sum of ERFari and ERFaci. We also modified the ERFaci for shortwave to net ERFaci (for both shortwave and longwave).
- Figure 11: We modified $\partial AOT$ to $\partial CCN$. We added new parameters such as $\partial CDNC$, $\partial CDR$, $\partial CA$, $\partial CF$, and net ERFaci to further explore ACI.
- Figure 12: We replaced Table 3 in the original manuscript to Figure 12 in the revised manuscript by adding relevant parameters such as $\partial CCN$, $\partial CDNC$, $\partial CDR$, $\partial CA$, $\partial CF$, and net ERFaci.
- Figure 13: To explain possible overestimations of the NSW6-simulated Twomey effect, we newly plotted global budgets of the annual averages of the NDW6- and NSW6-simulated Qc (mixing ratio of cloud droplets) and CDNC (cloud droplet number

concentrations).

- Table 1 in the original manuscript: We removed it and added a paragraph to explain the HRM and LRM as references in section 2.5 in the revised manuscript.
- Table 1 in the revised manuscript: We simply moved Table 2 in the original manuscript to Table 1 in the revised manuscript.
- Table 2 in the revised manuscript: We showed global and annual mean values of ERFari, ERFaci, and the sum of ERFari and ERFaci for shortwave, longwave, and net radiation under both all-sky and clear-sky conditions.
- Table A1: We newly added the statistical metric to compare results in this study with the references by Goto et al. (2020).
- Table A2: We simply moved Table A1 in the original manuscript to Table A2 in the revised manuscript, with two exceptions. One, we changed "References" to "References from model results". Second, we changed the $SO_2$ production value from 67.5 to 67.7.

Please note that some English was corrected in the revised manuscript.

[C3-2] It's hard to find specific information regarding the setup of experiments (NDW6, NSW6, HRM, LRM), except for the resolutions and microphysics schemes in Table 1. Which specific simulation years? What are the aerosol emissions used for the simulation years, mean or year-specific? Such information is important to determine whether the comparison of simulations results with observations and other models in the literature is valid. Please include the details in Table 1.

[A3-2] Thank you for your comment on the setup of experiments. As you pointed, some of explanation in the original manuscript was missed. So, we added section 2.3 named "Experimental conditions" including the information of the setup and emission inventories to the revised manuscript (Lines 201-216) as follows:

"All experiments with both NDW6 and NSW6 are carried out for 6-years after the 1-month spin-up calculation. The simulation results are climatological runs, because the model does not nudge meteorological fields such as wind and temperatures but nudges the sea surface temperature (SST) and sea ice by the results of the NICAM from Kodama et al. (2015). The initial conditions for the model spin-up are obtained from the end of the 1-year aerosol simulations coupled to NSW6 without nudging the meteorological fields under the present era.

The emission fluxes used in this study are the Hemispheric Transport of Air Pollution (HTAP)-v2.2 (Janssen-Maenhout et al., 2015) for BC, organic carbon (OC) and $SO_2$ from anthropogenic sources in 2010 and the Global Fire Emission Database (GFED) version 4 (van der Werf et al., 2017) for BC, OC and $SO_2$ from biomass burning in climatological average from 2005 to 2014. The ratio of OC to OM is set at 1.6 for anthropogenic activities and 2.6 for biomass burning (Tsigaridis et al., 2014). Secondary organic aerosols (SOAs) are assumed to form particles, which are calculated by multiplying the emission fluxes of isoprene and terpenes provided by the Global Emissions Initiative (GEIA) (Guenther et al., 1990) using constant factors. $SO_2$ is emitted from volcanic eruptions (Diehl et al., 2012) and is also formed from DMS, which is interactively emitted in the aerosol module (Bates et al., 1987). Sulfate is formed from $SO_2$ oxidation with a 3-dimensional distribution of monthly oxidants (ozone, $H_2O_2$ and OH) provided by a chemical transport model (CHASER) coupled to MIROC (Sudo et al., 2002). Emission

fluxes for dust (Takemura et al., 2009) and sea salt (Monahan et al., 1986) are interactively calculated in the model using mainly the wind speed at a height of 10 m."

In addition, we added a section 2.5 in the revised manuscript to explain the information of the models, i.e., HRM and LRM in Goto et al (2020) as references. Because we added this section to the revised manuscript, we removed Table 1 in the original manuscript. The section 2.5 in the revised manuscript (Lines 281-295) is as follows:

"Our previous model results provided in Goto et al. (2020) using NICAM.16 at a global 14-km high resolution (hereafter referred to as the HRM) and a global 56-km low resolution (hereafter referred to as the LRM) are used as references to compare the NICAM results. As mentioned in section 2.1, the number of vertical layers is set at 38, and the timestep is 1 minute in both the HRM and LRM. The integration periods in both the HRM and LRM are 3 years as climatological runs. The emission inventories, i.e., 2010 for anthropogenic sources, climatological average in 2005-2014 for biomass burning, and natural sources in the present era, and the nudged SST and sea ice in this study are identical to those in both the HRM and LRM, but the initial conditions in this study are different from those in both the HRM and LRM, which use the model results at the end of December after a 1.5-month spin-up. The initial conditions for the model spin-up are prepared by the reanalysis datasets of the National Centers for Environmental Prediction (NCEP) Final (FNL) (Kalnay et al., 1996) in November 2011. In the cloud microphysics and autoconversion modules, NDW6 coupled to Seifert and Beheng (2006) and NSW6 coupled to Khairoutdinov and Kogan (2000) are used in this study, whereas NSW6 coupled to Berry (1967) is used in both the HRM and LRM. The improvement in the aerosol module described in section 2.2 is also different from that in the HRM and LRM. The results of the HRM and LRM are useful for evaluating the current model results because the observations are limited in some parameters, such as aerosol global budgets and radiative forcings."

[C3-3] At many places, IRFari and ERFaci are referred to as "shortwave" aerosol forcing. If the longwave component is not considered at all, I don't think they are comparable to the cited values in the literature, which mostly include both shortwave and longwave components and are referred to as net radiative forcing. Please confirm and clarify.

[A3-3] Thank you for your suggestion. Surely, we didn't show the results of IRF and ERF for longwave as well as net. We put these estimates in the revised Figure 10 and Table 2. We modified Figure 10 and added net ERFaci values to the revised manuscript.

[Figure]

Figure 10: Global and annual mean values of (a) effective radiative forcing for anthropogenic aerosol-radiation interaction (ERFari) for shortwave and net (sum of shortwave and longwave) radiation, (b) ERFaci for anthropogenic aerosol-cloud interaction, and (c) the net ERF (sum of ERFari and ERFaci). All units are in W m⁻². In ERFari, the reference of Forster21 is estimated in the net radiation by IPCC-AR6 or Forster et al. (2021), whereas the reference of Thortsen21 is estimated in the shortwave radiation by Thorsen et al. (2021). The reference for Smith20 is Smith et al. (2020). The values are also listed in Table 2.

Table 2: Global and annual mean values of ERFari for anthropogenic aerosol, ERFaci for anthropogenic aerosol-, and the net ERF (sum of ERFari and ERFaci) for shortwave, longwave, and net (sum of shortwave and longwave) radiation under the all-sky abnd clear-sky conditions. All units are in W m⁻².

|  | ERFari under the all-sky conditions | | | |
|  | NDW6 | NSW6 | HRM | LRM |
| Shortwave | -0.22 | -0.26 | -0.33 | -0.26 |
| Longwave | 0.03 | 0.03 | 0.04 | 0.02 |
| Net | -0.19 | -0.23 | -0.29 | -0.24 |
|  | ERFari under the clear-sky conditions | | | |
| Shortwave | -0.52 | -0.60 | -0.63 | -0.51 |
| Longwave | 0.05 | 0.05 | 0.05 | 0.03 |
| Net | -0.47 | -0.55 | -0.57 | -0.48 |
|  | ERFaci | | | |
| Shortwave | -1.34 | -0.63 | -0.81 | -1.17 |
| Longwave | 0.06 | -0.10 | -0.12 | 0.07 |
| Net | -1.28 | -0.73 | -0.93 | -1.10 |
|  | ERFari+ERFaci | | | |
| Shortwave | -1.56 | -0.89 | -1.15 | -1.43 |
| Longwave | 0.09 | -0.07 | -0.08 | 0.09 |
| Net | -1.47 | -0.96 | -1.23 | -1.34 |

[C3-4] Both NDW6 and NSW6 appear to simulate a too weak SWCRF and LWCRF (shown in Table 2), as compared to observations and historical mean of CMIP models. Is this due to interannual variability or model bias? How does this affect the evaluation of aerosols and their associated forcings in NDW6? Please include a discussion on this issue.

[A3-4] Thank you for your comment on CRF. We checked the results of CRF in all six years and found that the standard deviations were smaller than the difference between NDW6 and CERES. The standard deviations were calculated to be 0.2 Wm$^{-2}$ (NDW6, Annual), 0.6 Wm$^{-2}$ (NDW6, January), and 0.2 Wm$^{-2}$ (NDW6, July). Therefore, the difference between NICAM and CERES is not caused by the internal variability.

In the beginning of Section 4.2, we roughly mentioned the results of SWCRF and the bias, and discussed the spatial and zonal distributions in Appendix E. As mentioned in Appendix A, the SWCRF results are generally consistent to the LWP results. As mentioned there and in section 3.1 related to LWP, the global averages in NDW6 are not closer to the observation compared to those in NSW6, but this can be caused by compensation errors in space (In the original manuscript, we mentioned the error in space and time, but after checking again, we found that it was caused by the error only in space. We modified this in the revised manuscript (Lines 328-329, 340, 515, 621, and 659)). The NICAM results of both SWCRF and LWP are underestimated compared to the observation in most regions, but the NSW6 has some positive bias of SWCRF and LWP in some regions such as the western Pacific Ocean. As a result, the zonal and global averages of SWCRF and LWP in NDW6 are underestimated relative to NSW6. Therefore, we cannot conclude that the bias of SWCRF and LWP in NDW6 is larger than that in NSW6.

The large underestimation of SWCRF in the western Pacific Ocean is related to the underestimation of the clouds. The underestimation of the low-level clouds in NICAM were realized in NICAM studies (e.g., Kodama et al., 2021). Since the NICAM-simulated precipitation is close to the observation, the NICAM-simulated cloud-to-precipitation conversion is overestimated, as shown in Figure 5. This causes the overestimation of the wet deposition for aerosols as shown in Figure 7; therefore, the underestimation of the NICAM-simulated SWCRF can cause the underestimation of the simulated aerosols in such tropic regions. This is a possible bias to be caused by the difference in the SWCRF between the model and observation.

Over land where the observed LWP information is very limited, the observed SWCRF may be useful to evaluate the model. In high aerosol loading areas such as China, Europe, and the United State, the simulated SWCRF tends to be underestimated compared to the observation. This indicates the underestimation of the simulated LWP and/or the overestimation of the simulated cloud droplet effective radius (CDR). When the simulated LWP is underestimated, the simulated aerosols are underestimated, according to the above discussion. When the simulated CDR is overestimated, the simulated CCN is underestimated. This is consistent to the underestimation of the simulated aerosol. Therefore, if negative biases in the simulated SWCRF are eliminated, the simulated aerosols will increase. In the revised manuscript (Lines 516-522), we modified this as follows:

"The NDW6-estimated SWCRF values are concluded to be better than the NSW6 results, but the underestimation of the simulated SWCRF can have an impact on the aerosol simulations. The underestimation of the simulated SWCRF indicates the underestimation of the simulated LWP and/or the overestimation of the simulated CDR. When the

simulated LWP is underestimated, the simulated aerosols are also underestimated because the simulated precipitation is generally comparable to the observations in this study, as shown in Figure 1. When the simulated CDR is overestimated, the simulated CCN must be underestimated. This is consistent with the underestimation of the simulated aerosol. Therefore, if the negative biases in the simulated SWCRF are eliminated, the simulated aerosols will increase."

As for longwave CRF, the impacts of the longwave CRF (LWCRF) on aerosols are small in this study, because the LWCRF is mainly determined from the high-level clouds and convective clouds and our model ignored the direct interaction between aerosols and ice crystal (as an ice nuclei). Therefore, if negative biases in the simulated LWCRF in this study are eliminated, the simulated high-level clouds increase but the simulated aerosols may not be changed. Because the direct interaction between aerosols and ice clouds is not considered in this model, the impacts of the bias of the simulated LWCRF are unclear. In the revised manuscript (Lines 523-525), we modified this as follows:

"The underestimation of the simulated LWCRF is caused by the underestimation of the simulated high-level clouds, but the impacts of this negative biases in the simulated LWCRF on the aerosol simulations are unclear due to ignorance of the interaction between aerosols and ice crystals (as ice nuclei) in this model."

**Minor comments and technical corrections:**
[C3-5] L118-119: If CDNC is fully prognostic in the NDW6 double-moment scheme, I assume it's always updated with source and sink tendencies. Why is there an additional constraint by CCN that depends on supersaturation as well? please clarify.

[A3-5] We apologize for the lack of explanation about this part "In addition, a CDNC value is assumed to be updated to a CCN value only when the CCN value exceeds the CDNC value in a grid box". The CCN value only when the CCN value exceeds the CDNC value in a grid box is an aerosol activation process and a source term. The sink tendencies are accretion, autoconversion, and evaporation for water clouds. As you mentioned, the CDNC is determine by source and sink tendencies.
The additional constraint by CCN that depends on supersaturation is needed to nucleate water clouds (not to nucleate ice clouds). One reason of this is an aerosol activation process parameterized by Abdul-Razzak and Ghan (2000) is applicable only for water clouds but calculate in any clouds even below supersaturation in the aerosol physic model.
Therefore, we modified this in the revised manuscript (Lines 122-126) as follows:
"In addition, a source term of CDNC is assumed to be updated to a CCN value only when the CCN value exceeds the CDNC value in a grid box. The CDNC is updated with source (aerosol activation) and sink (autoconversion, accretion, and evaporation for water clouds) in NDW6 (Seiki and Nakajima, 2014). The balance of source and sink tendencies determines the CDNC in NDW6."

[C3-6] L123: Please clarify on "which" is updated in this study. NDW6 or NSW6?

[A3-6] Thanks you for your comment. This is NSW6. To avoid unclear expression, we removed the part 'which is updated in this study (cf.., Seiki and Roh, 2020)" from the revised manuscript.

[C3-7] L132: Is there no shallow convection parameterization at 14-km grid spacing? Please justify.

[A3-7] We also do not use shallow convection schemes, like other studies using NICAM with 14-km grid spacing (e.g., Satoh et al., 2010; Kodama et al., 2021). We modified this in the revised manuscript (Lines 143-144) as follows:
"As in previous studies using the NICAM (e.g., Satoh et al., 2010; Kodama et al., 2021), no parameterization schemes for deep and shallow convection are used in this study."
We realized possible biases caused by not including a shallow convection parameterization. For example, the simulated low-level clouds and shortwave radiation such as OSR and SWCRF in NICAM tend to be underestimated compared to the observation (Kodama et al., 2021). This can also influence SST and is very important for atmosphere-ocean coupling models to predict SST. For example, Masunaga et al. (2023) introduced a flux adjustment to the atmosphere-ocean coupled NICAM without shallow convection parameterization to avoid SST drift. However, we don't think the exclusion of the shallow convection parameterization is so important for atmospheric circulation models with fixed SST. Therefore, we don't use any shallow convection parameterization like previous studies using the NICAM as an atmospheric circulation model (e.g., Satoh et al., 2010; Kodama et al., 2021).


| Anthropogenic Source | 1850 | 2010 | 2010-1850 [%] |
|---|---|---|---|
| $SO_2$ [$ktSO_2$] | 2,481 | 115,487 | 2.1 |
| BC [ktC] | 934 | 7,755 | 12.0 |
| OC [ktC] | 4,262 | 18,755 | 22.7 |

---

## Referee Report (RR1)

**2nd review of Goto et al.**

Thanks for making many of the changes suggested by the referees. I am mostly happy with the paper now. There are just a few places where some additional figures would be useful to show what is written in the text and where the text could be improved to make things clearer.

**Major points**

L413 – "Therefore, the overestimation of the NSW6-simulated LWP in the eastern Pacific Ocean and Southern Atlantic Ocean effectively balanced the underestimation of the zonal averages of the simulated LWP and unexpectedly led to zonal LWP values closer to the MAC results."

- I found this difficult to follow. I suggest rewording to :-
- "Therefore, the overestimation of the NSW6-simulated LWP in the eastern Pacific Ocean and Southern Atlantic Ocean effectively balanced the underestimation in the Western Pacific Ocean and Indian Ocean, which led to zonal LWP values that were closer to the MAC results."
- Also, it is still not easy to make this comparison from Fig. 2. Map plots of the bias of the models vs MAC would be helpful. As would averages (or the bias and/or LWP fields) for 0-30S as a function of the longitude.

L614 – "When the simulated LWP is underestimated, the simulated aerosols are also underestimated, because the simulated precipitation is generally comparable to the observations in this study, as shown in Figure 1. When the simulated CDR is overestimated, the simulated CCN must be underestimated. This is consistent with the underestimation of the simulated aerosol. Therefore, if the negative biases in the simulated SWCRF are eliminated, the simulated aerosols will increase."

- This is quite speculative. It's not clear how well linked the LWP, CDR and precipitation in the clouds in question are. For frontal mid-latitude precipitation the relationships may not be as straightforward as for stratocumulus. Precipitation is likely to be determined by large scale meteorology. Plus, it's not clear how close to the observations the model needs to be in order to allow a constraint on LWP or CDR. The comparison between models and observations is likely highly uncertain for precipitation. I'm not sure whether you need the extra text here (and it is quite difficult to explain what you are trying to say in a clear way!).

L663 – "Specifically, above a height of 3 km, where ∂Qc is close to zero and ∂CDNC has a positive value, ∂CDR should be small. The possible overestimation of ∂CDR in NSW6 represents possible overestimation of the Twomey effect in NSW6."

- You would need to add dCDR to Fig. 13 to show this.
- Generally it seems strange that such large changes in CDNC in NSW6 (assumed to be the same as dCCN) produce fairly small changes in CDR. E.g., Fig. 12 states that for the US there is around a 60 per cm3 increase in CDNC for NSW6. If we assume a baseline CDNC of 200 per cm3 (which is quite high) then the equates to a 260/200 = 1.3x increase. Since CDR is approximately proportional to CDNC^(-1/3) (assuming equal LWP) then this would be expected to lead to a 1.3^(-1/3) = 0.916x change in CDR. For a PI CDR value of 10um this would lead to a new CDR of 9.16um, or a 0.84um decrease. This is similar to what is

seen in Fig. 12 for NSW6. However, for NDW6 Fig. 12 says that there was only a 4 per cm3 increase in CDNC. Using the same numbers, this would lead to only a 0.07um decrease in CDR; but the quoted change is around 0.6um. So, there seems to be an inconsistency there. Using lower PI values for CDNC could allow the NDW6 numbers to work out, but then the NSW6 value would be out. So, something is not quite right here I think – could it be that the radiation scheme (presumably where CDR is calculated?) of NSW6 doesn't actually assume that the change in CCN is equal to the change in CDNC?

-

L680 - "By decreasing the simulated ∂CDR, increasing the simulated ∂LWP from PI to PD, and increasing the simulated ∂CA and ∂CF at 1-km height, the negative values of the simulated ERFaci in the industrial regions, such as the United States, Europe, and East Asia, increase in magnitude."

- Presumably you are referring to NSW6 vs NDW6 here? You need to make that clear. However, dCDR is larger (in magnitude) for NSW6, and dLWP, dCA and dCF are smaller. Maybe you meant CDR is more negative, but better to talk in terms of magnitude I think.

L685 – "by considering the uncertainty caused by the assumption in the PI conditions."

- It's not clear what you mean here?

L691 – "Therefore, it was suggested that the ERFaci due to the cloud lifetime effect in NDW6 was larger than that in NSW6 due to the Twomey effect,"

- This needs a bit of explanation. Perhaps something like "This, combined with the smaller magnitude of decrease in CDR for NDW6 vs NSW6 and the larger magnitude increase in LWP and CF, suggests that the ERFaci due to the cloud lifetime effect in NDW6 is larger than that in NSW6 due to the Twomey effect,…"

-

**Typos/grammar**

L18 – "but the differences between the results of NDW6 and NSW6 experiments were larger for some aerosol species, especially dust and sulfate, compared to those between the experiments with different horizontal resolutions, i.e., 14 km and 56 km grid spacing, as shown in a previous study."

- I suggest that you start a new sentence and slightly rewrite this to make this sentence clearer : "However, for some aerosol species, especially dust and sulfate, the differences between the NDW6 and NSW6 experiments were larger than those between experiments with different horizontal resolutions (14 km and 56 km grid spacing), as shown in a previous study."

L34 – "caused by ignorance of sink process in the cloud droplet number concentrations." -> "caused by the ignorance of sink processes for cloud droplet number concentrations."

L218 - "Hoesly et al. (2018) estimated global averages of the differences in the emission amounts of anthropogenic sources between 1850 and 2010 to be 2.1% (sulfate), 12.0% (BC), and 22.7% (OC)."

- I think this should be "Hoesly et al. (2018) estimated that the globally averaged emissions in 1850 were 2.1% of the 2010 emissions for sulfate, 12.0% for BC and 22.7% for OC."

---

## Author Response (AR2)

Dear Editor and Reviewers,

We appreciate your continued contributions to improving our manuscript. In this revision, we found our mistake in the calculation of the NDW6-simulated cloud droplet effective radius (CDR) used in Figures 11 and 12(c) in the original manuscript. The NDW6-simulated CDR should be calculated using the NDW6-simulated CDNC. However, the NDW6-simulated CDR was calculated using the NDW6-simulated CCN, as was the case for the NSW6-simulated CDR. Therefore, in the revised version, the NDW6-simulated CDR was newly calculated using the NDW6-simulated CDNC. In the original version, this CDR calculation was performed at each time step in the model, but due to the computer resource limitations, the 6-year integration cannot be performed again in the model. Therefore, we recalculated the CDRs for both NDW6 and NSW6 using monthly averages of cloud mass and number concentrations. Using these results, the global mean difference in the NDW6-simulated CDR between PD and PI, ∂CDR, was changed from -0.17 µm to -0.62 µm, whereas the NSW6-simulated ∂CDR was changed from -0.34 µm to -0.31 µm. The regional averages were also changed (Figure 13(c) in the revised manuscript). These were also shown in this reply. Note that this CDR recalculation does not affect other parameters (like LWP, ERFaci, so on), because this CDR was diagnostic and variable for the output.

[Figure]

Figure: Modified ∂CDR used in Figure 11 in the previous version (Figure 12 in the revised manuscript).

[Figure]

Figure: Modified ∂CDR used in Figure 12 in the previous version (Figure 13 in the revised manuscript).

In this revision, we realized that we did not evaluate CDR in the previous manuscript, so we newly added the evaluation to Table 1 and section 3.1 in the revised manuscript (Lines 134-142, 243, 250, 343-344, 346, and 348). The simulated CDRs were evaluated using CDRs at warm-topped clouds requiring specific conditions of cloud mixing ratio and cloud top temperature. To calculate the simulated CDRs, we newly integrated the model for only 1-year (due to limitations in available computer resources, 6-year integration was too heavy to recalculate). As a result, both the NDW6-simulated and NSW6-simulated CDRs at warm-topped clouds were lower than the MODIS-estimated results, but the NDW6-simulated CDR was generally closer to the observation compared to the NSW6-simulated results.

In the end, one of our main conclusions was slightly modified. In the previous version, we did not address the importance of the Twomey effect in NDW6, but we concluded that both the Twomey and the cloud lifetime effects in NDW6 were larger than in NSW6. This point was reflected to the revised version (Lines 32-33, 587-589, and 645-648). Except for this point, the conclusions remain.

In addition, we corrected some typographical and other errors.

- As noted in [A-2], a new figure (Figure 3) was added to the revised manuscript and the figure numbers were changed.
- IPCC(Forster et al., 2021): -0.25 Wm$^{-2}$ (-0.45 Wm$^{-2}$ to -0.05 Wm$^{-2}$) to -0.3±0.3 Wm$^{-2}$
- We added "the lower limit of" to Line 501 of the revised manuscript.
- We corrected the bars used in Thorsen21 in Figure 11(a).
- We corrected a typo in "and" in the caption of Table 2, deleted "shortwave and" from the caption of Figure 11, and corrected "sink processes" in the caption of Figure 12.
- We removed the explanation for the abbreviations, which are explained in the text, from the captions of Figures 12, 13, and 14.

We apologize for our mistake in the calculate of CDR in NDW6 and apologize for the correction made here, which should have been addressed in the first revision.

Sincerely yours,
Daisuke Goto

[C-1] 2nd review of Goto et al.

Thanks for making many of the changes suggested by the referees. I am mostly happy with the paper now. There are just a few places where some additional figures would be useful to show what is written in the text and where the text could be improved to make things clearer.

[A-1] We appreciate your great contributions to our manuscript. Your comments and suggestions are very helpful. We would like to answer each point below.

Major points

[C-2] L413 – "Therefore, the overestimation of the NSW6-simulated LWP in the eastern Pacific Ocean and Southern Atlantic Ocean effectively balanced the underestimation of the zonal averages of the simulated LWP and unexpectedly led to zonal LWP values closer to the MAC results."

[C-2-1] I found this difficult to follow. I suggest rewording to: "Therefore, the overestimation of the NSW6-simulated LWP in the eastern Pacific Ocean and Southern Atlantic Ocean effectively balanced the underestimation in the Western Pacific Ocean and Indian Ocean, which led to zonal LWP values that were closer to the MAC results."

[C-2-2] Also, it is still not easy to make this comparison from Fig. 2. Map plots of the bias of the models vs MAC would be helpful. As would averages (or the bias and/or LWP fields) for 0-30S as a function of the longitude.

[A-2] As for [A-2-1], we agreed your suggestion. We used this expression in the revised manuscript. As for [A-2-2], we newly added Fig 3 to the revised manuscript by calculating differences in the LWP among NDW6, NSW6, and MAC. To reflect this, the numbers of Figures are changed in the revised manuscript. We also prepared a new figure for 0-30S averages as a function of the longitude, but we thought that this information is overlapped with new Figure 3, so we didn't insert the specific figure to the revised manuscript (we showed it only in this reply).

[Figure]

Figure 3 (new figure): Horizontal distributions of differences in LWP among NDW6, NSW6 and MAC over only the ocean as annual, January, and July averages. All units are in g m$^{-2}$.

[Figure]

Figure (not used in the revised manuscript): Bias and LWP fields for 0-30S as a function of the longitude, as annual, January, and July averages.

[C-3] L614 – "When the simulated LWP is underestimated, the simulated aerosols are also underestimated, because the simulated precipitation is generally comparable to the observations in this study, as shown in Figure 1. When the simulated CDR is overestimated, the simulated CCN must be underestimated. This is consistent with the underestimation of the simulated aerosol. Therefore, if the negative biases in the simulated SWCRF are eliminated, the simulated aerosols will increase."

- This is quite speculative. It's not clear how well linked the LWP, CDR and precipitation in the clouds in question are. For frontal mid-latitude precipitation the relationships may not be as straightforward as for stratocumulus. Precipitation is likely to be determined by large scale meteorology. Plus, it's not clear how close to the observations the model

needs to be in order to allow a constraint on LWP or CDR. The comparison between models and observations is likely highly uncertain for precipitation. I'm not sure whether you need the extra text here (and it is quite difficult to explain what you are trying to say in a clear way!).

[A-3] Thank you for your comment. As you pointed out, this part is quite speculative. We largely modified this part as follows: "The NDW6-estimated SWCRF values are concluded to be better than the NSW6 results, but the underestimation of the NDW6-simulated SWCRF is mainly caused by the underestimation of the simulated LWP due to the underestimation of the simulated CDR shown in Table 1. The impacts of this negative biases in the simulated SWCRF and LWP on the aerosol simulations are still unclear due to complex interactions between aerosols, clouds, and precipitation."

[C-4] L663 – "Specifically, above a height of 3 km, where $\partial Qc$ is close to zero and $\partial CDNC$ has a positive value, $\partial CDR$ should be small. The possible overestimation of $\partial CDR$ in NSW6 represents possible overestimation of the Twomey effect in NSW6."
[C-4-1] You would need to add dCDR to Fig. 13 to show this.
[C-4-2] Generally it seems strange that such large changes in CDNC in NSW6 (assumed to be the same as dCCN) produce fairly small changes in CDR. E.g., Fig. 12 states that for the US there is around a 60 per cm3 increase in CDNC for NSW6. If we assume a baseline CDNC of 200 per cm3 (which is quite high) then the equates to a 260/200 = 1.3x increase. Since CDR is approximately proportional to CDNC^(-1/3) (assuming equal LWP) then this would be expected to lead to a 1.3^(-1/3) = 0.916x change in CDR. For a PI CDR value of 10um this would lead to a new CDR of 9.16um, or a 0.84um decrease. This is similar to what is seen in Fig. 12 for NSW6. However, for NDW6 Fig. 12 says that there was only a 4 per cm3 increase in CDNC. Using the same numbers, this would lead to only a 0.07um decrease in CDR; but the quoted change is around 0.6um. So, there seems to be an inconsistency there. Using lower PI values for CDNC could allow the NDW6 numbers to work out, but then the NSW6 value would be out. So, something is not quite right here I think – could it be that the radiation scheme (presumably where CDR is calculated?) of NSW6 doesn't actually assume that the change in CCN is equal to the change in CDNC?

[A-4] Thank you for your comment. As you suggested, we added $\partial CDR$ to Figure 13 (Figure 14 in the revised manuscript). As mentioned in the beginning of this reply, the NDW6-simulated $\partial CDR$ is larger than the NSW6-simulated $\partial CDR$. When plotting the vertical profile of $\partial CDR$ in the global average, we found that we used incorrect weighting factors to estimate the global average for each layer, so we fixed them in Figure 14 in the revised manuscript. We also found typos in units of $\partial CDNC$, so we corrected them. This correction does not affect other parts in the manuscript. After the correction, the conclusions remained unchanged, except for the difference in the $\partial CDR$ between NDW6 and NSW6. In the revised manuscript, we modified this part as follows: "Specifically, above a height of 3 km, $\partial Qc$ is close to zero, but $\partial CDR$ is not zero because $\partial CDNC$ has a positive value. Even though the magnitude of $\partial CDR$ in NSW6 is lower than that in NDW6, this represents possible overestimation of the Twomey effect in NSW6."

[Figure]

Figure 14 in the revised manuscript: Global budgets of the annual averages of the NDW6- and NSW6-simulated Qc (mixing ratio of cloud droplets), the NDW6-simulated CDNC, the NSW6-simulated CDNC (which is equal to CCN number concentrations), and the NDW6- and NSW6-simulated CDR (cloud droplet effective radius for warm clouds)

Regarding [A-4-2], the main point is the difference in the PI value of CDNC between NDW6 and NSW6. The simulated value of CDNC for PI in NSW6 is approximately 40 per cm³ in the US, for example. According to your calculations, CDNC increases by a factor of 2.5 and CDR changes by a factor of 0.737. Since the CDR value for PI in the US is 6 $\mu$m, this leads to a new CDR of 4.42 $\mu$m, or a 1.6 $\mu$m decrease. In contrast, the simulated CDNC value for PI in NDW6 is approximately 2 per cm³ in the US; The CDNC in NDW6 tends to be lower (which we understand is too low, but it is still difficult to resolve this issue within this manuscript). Thus, a 3-fold increase in CDNC results in a 0.693-forld change in CDR. Therefore, we think that the relationship between CDNC and CDR in NDW6 are consistent to the results in NSW6. We added this point to the revised manuscript (Lines 545-550) as follows: "For example, in the United States, the NSW6-simulated ∂CDNC (=∂CCN) is approximately 60 cm⁻³ and the NSW6-simulated ∂CDR is approximately -1.1 $\mu$m, whereas the NDW6-simulated ∂CDNC is approximately 4 cm⁻³ and the NDW6-simulated ∂CDR is approximately -2.3 $\mu$m. The difference in the ∂CDNC-∂CDR relationship between NDW6 and NSW6 is caused by the difference in the baseline of CDNC and CDR. The NDW6-simulated CDNC under both the PD and PI aerosol conditions is much lower than the NSW6-simulated results, whereas NDW6-simulated CDR under both the PD and PI aerosol conditions is larger than the NSW6-simulated results."

[C-5] L680 - "By decreasing the simulated $\partial$CDR, increasing the simulated $\partial$LWP from PI to PD, and increasing the simulated $\partial$CA and $\partial$CF at 1-km height, the negative values of the simulated ERFaci in the industrial regions, such as the United States, Europe, and East Asia, increase in magnitude." Presumably you are referring to NSW6 vs NDW6 here? You need to make that clear. However, dCDR is larger (in magnitude) for NSW6, and dLWP, dCA and dCF are smaller. Maybe you meant CDR is more negative, but better to talk in terms of magnitude I think.

[A-5] Thank you for your comment. This sentence is for both NDW6 and NSW6, so we added "In both NDW6 and NSW6" to this part in the revised manuscript.

[C-6] L685 – "by considering the uncertainty caused by the assumption in the PI conditions." - It's not clear what you mean here?

[A-6] Thank you. We deleted this from the revised manuscript.

[C-7] L691 – "Therefore, it was suggested that the ERFaci due to the cloud lifetime effect in NDW6 was larger than that in NSW6 due to the Twomey effect," - This needs a bit of explanation. Perhaps something like "This, combined with the smaller magnitude of decrease in CDR for NDW6 vs NSW6 and the larger magnitude increase in LWP and CF, suggests that the ERFaci due to the cloud lifetime effect in NDW6 is larger than that in NSW6 due to the Twomey effect,..."

[A-8] Thank you for your suggestion. The correction of CDR in NDW6 suggests that the ERFaci due to both the Twomey and cloud lifetime effects in NDW6 was larger than that in NSW6. However, the comparisons in the vertical profiles of $\partial$Qc, $\partial$CDNC, and $\partial$CDR suggest that the NSW6-simulated ERFaci certainly includes some bias due to the overestimation of the Twomey effect, especially above 3 km height. Therefore, we modified this (Lines 586-588 in the revised manuscript) and sentences in summary (Lines 643-646 in the revised manuscript) as follows: "Therefore, it was suggested that the ERFaci due to both the Twomey and cloud lifetime effects in NDW6 was larger than that in NSW6, although the NSW6-simulated ERFaci certainly includes some bias due to the overestimation of the Twomey effect."

Typos/grammar
[C-8] L18 – "but the differences between the results of NDW6 and NSW6 experiments were larger for some aerosol species, especially dust and sulfate, compared to those between the experiments with different horizontal resolutions, i.e., 14 km and 56 km grid spacing, as shown in a previous study." - I suggest that you start a new sentence and slightly rewrite this to make this sentence clearer : "However, for some aerosol species, especially dust and sulfate, the differences between the NDW6 and NSW6 experiments were larger than those between experiments with different horizontal resolutions (14 km and 56 km grid spacing), as shown in a previous study."

[A-8] Thank you so much for the correction. We used them in the revised manuscript.

[C-9] L34 – "caused by ignorance of sink process in the cloud droplet number concentrations." -> "caused by the ignorance of sink processes for cloud droplet number concentrations."

[A-9] Thank you so much for the correction. We reconsidered this and decided to remove this from the abstract since it is not directly related to why the ERFaci in NDW6 is larger than that in NSW6. We also found places in the manuscript that needed correction related to your comment, so we changed from "sink process" to "sink processes" in Line L540 and Figure 12 in Lines 1295 in the revised manuscript.

[C-10] L218 - "Hoesly et al. (2018) estimated global averages of the differences in the emission amounts of anthropogenic sources between 1850 and 2010 to be 2.1% (sulfate), 12.0% (BC), and 22.7% (OC)." - I think this should be "Hoesly et al. (2018) estimated that the globally averaged emissions in 1850 were 2.1% of the 2010 emissions for sulfate, 12.0% for BC and 22.7% for OC."

[A-10] Thank you so much for the correction. We used them in the revised manuscript.

---

## Author Response (AR3)

**Public justification (visible to the public if the article is accepted and published)**:
The authors have addressed the reviewer comments, and from a 2nd review from reviewer 1, the authors have explained they realised a mistake in the methods they applied for the calculation of NDW6-simulated cloud droplet effective radius (CDR), used in the aerosol-cloud-interactive forcing shown in Figures 11 of the original manuscript, and within panel c of the adapted Figure 12, showing each step of the calculation.

In applying the corrected method, the authors have had to do calculations from monthly-mean fields, rather than online calculations.

I have read through the comments, and from what I can see, the authors have been very diligent in checking their calculations, and in replying to each part of the reviewer comments.

The addition of the extra Figures during the review provides a clear and transparent analysis to understand how the change from single-moment to double-moment cloud microphysics affects the model predictions.

The authors have re-calculated the post-industrial change in cloud effective radius (pre-industrial to present-day) and with the correction, now find larger changes in cloud droplet effective radius, with lower pre-industrial values in some metrics within the double-moment scheme.

Considering all these aspects, and that the manuscript has 5 Appendices (A, B, C, D and E), I find the manuscript to be very clear, and helpful to other model developers referring to this paper.

Reading text added with this latest revised manuscript, I noticed 2 grammatical/typo errors, and am recommending publication once these 2 minor typo-errrors are corrected:

1) Line 533 of Tracked-Changes manuscript (TC4) -- "The impacts of this negative biases in the simulated SWCRF and LWP on the aerosol simulations are still unclear due to complex interactions between aerosols, clouds, and precipitation"

Please change "of this negative biases" instead to "of these negative biases".

2) Line 578 of Tracked-Changes manuscript (TC4) -- the authors have added the sentence "Even though the magnitude of $\partial$CDR in NSW6 is lower than that in NDW6, this represents possible overestimation of the Twomey effect in NSW6."

I realise this statement relates to the 2 preceding sentences,

"In contrast, the vertical profile of $\partial$CDNC in NSW6 is different from that of $\partial$Qc because NSW6 cannot predict CDNC and adopts $\partial$CCN. Specifically, above a height of 3 560 km, $\partial$Qc is close to zero, but $\partial$CDR is not zero because $\partial$CDNC has a positive value"

But the current wording stating "this represents" is not specific enough
That sentence needs to stated exactly what it is, that "represents possible overestimation".

I think the authors mean the fact that the single-moment NSW scheme has to predict CDN changes from changes in CCN.

And my suggestion is to actually state that specifically in the sentence.

So, I suggest to change:

"Even though the magnitude of $\partial$CDR in NSW6 is lower than that in NDW6, this represents possible overestimation of the Twomey effect in NSW6."

instead to

"Even though the magnitude of $\partial$CDR in NSW6 is lower than that in NDW6, the fact that NSW6 bases predicted CDNC changes on CCN changes represents a potential source of overestimation of the Twomey effect in NSW6."

Or some similar wording to this.

Dear Editor,

We appreciate your great contributions as editor of our manuscript. Your very careful review and handling have greatly improved this manuscript. The two points you raised have been incorporated into the revised version. Thank you very much.

Sincerely yours,
Daisuke Goto

[revised manuscript text omitted]